# Geometric Coherence Learning for Structuring Value Functions in Plain MDPs

Zuyuan Zhang [1]  Zeyu Fang [1]  Tian Lan [1]

## Abstract

Geometric properties can be leveraged to stabilize and speed reinforcement learning. Existing examples include encoding symmetry structure, geometry-aware data augmentation, and enforcing structural restrictions. In this paper, we take a novel view of RL through the lens of order theory and recast value function estimates into learning a desired poset (partially ordered set). We propose *GCR-RL* (Geometric Coherence Regularized Reinforcement Learning) that computes a sequence of super-poset refinements – by refining posets in previous steps and learning additional order relationships from temporal difference signals – thus ensuring geometric coherence across the sequence of posets underpinning the learned value functions. Two novel algorithms by Q-learning and by actor–critic are developed to efficiently realize these super-poset refinements. Their theoretical properties and convergence rates are analyzed. We empirically evaluate GCR-RL in a range of tasks and demonstrate significant improvements in sample efficiency and stable performance over strong baselines.

## 1. Introduction

Reinforcement learning (RL) has achieved impressive empirical success in fields from visual control to planning (Mnih et al., 2015; Van Hasselt et al., 2016; Hessel et al., 2018; Bellemare et al., 2017). Due to the combination of bootstrapping and function approximation using limited data, temporal-difference (TD) updates can diverge or oscillate under approximation error and distribution variations (Tsitsiklis & Van Roy, 1996; Baird et al., 1995), leading to unstable learning behavior (Schaul et al., 2015). To this end, existing work have leveraged geometric properties to regularize and speed up RL. Relevant efforts include abstracting bisimulation relations (Dean & Givan, 1997; Li et al., 2006; Ferns et al., 2004; Castro, 2020; Ravindran, 2004), encoding symmetry structures (Cohen & Welling, 2016; Kondor & Trivedi, 2018; Weiler et al., 2018; Finzi et al., 2020; Bronstein et al., 2021; Kawano et al., 2021), geometry-aware data argumentation (Laskin et al., 2020a; Yarats et al., 2021; Laskin et al., 2020b), and enforcing order restrictions (Agrawal et al., 2019; Blondel et al., 2020) like acyclicity (Zheng et al., 2018) and preference (Christiano et al., 2017). However, there is no systematic framework exploiting geometric structures in value functions with a rigorous theoretical support. Recent metric-style approaches are complementary to this objective. Bisimulation metrics, MICo, generalized bisimulation metrics, and quasimetric RL impose metric or quasimetric geometry on states, representations, or goal-reaching values (Ferns et al., 2004; Castro et al., 2021; Tao et al., 2026; Wang et al., 2023). These methods ask whether two states are close, behaviorally similar, or separated by goal-reaching cost. In contrast, GCR-RL learns a TD-induced directed relation and then maintains the poset axioms needed for coherent value ordering. Thus quotienting and DAGification are not metric constraints: quotienting identifies symmetric state–action elements before antisymmetry is imposed, while DAGification removes local feedback loops before isotonic alignment.

We take a novel view of RL through the lens of order theory (Davey & Priestley, 2002), by regarding value function estimates as learning a desired poset (partially ordered set[1]), denoted by $X$, with state-action pairs as its elements and a minimal partial order $\preceq_D$ necessary to determine the optimal actions in any states. We argue that learning such a poset is sufficient to obtain an optimal policy in RL. Toward this goal, we design a new process that computes a sequence of *super-poset refinements* – by refining posets in previous steps (as sub-posets) and learning additional order relationships from TD signals. This process enforces geometric coherence across the sequence of posets (underpinning learned value functions) and can achieve significantly speed-up in RL. By definition, a poset must satisfy *reflexivity, transitivity*, and *antisymmetry* (Davey & Priestley, 2002). Reflexivity and transitivity are natural for multi-step returns in RL, while *antisymmetry* is what prevents value circles

---

[1] Department of Electrical & Computer Engineering, The George Washington University, Washington, DC, USA . Correspondence to: Zuyuan Zhang <zuyuan.zhang@gwu.edu>, Zeyu Fang <joey.fang@gwu.edu>, Tian Lan <tlan@gwu.edu>.

*Proceedings of the 43rd International Conference on Machine Learning*, Seoul, South Korea. PMLR 306, 2026. Copyright 2026 by the author(s).

[1] Or a continuous poset for MDPs with continuous spaces.

and can further improve the convergence of value function, due to the elimination of value update loops and a reduced problem space. To ensure valid posets in our solution with antisymmetry, we identify symmetry by learning algebraic operators of automorphism on $X$ and consider the resulting quotient set $X/_\sim$ in our proposed GCR-RL (Geometric Coherence Regularized RL).

Two different approaches are proposed to enable GCR-RL. First, we develop a *soft* enforcement approach that casts geometric coherence as an explicit regularizer added to standard TD learning. Concretely, at refinement step $k$ we maintain a super-poset (represented by a directed comparison set) $\mathcal{D}_k$ over elements of $X$ (or equivalence classes in $X/_\sim$), and optimize a coherence-regularized objective $\min_{\theta,\phi} \ \mathcal{L}_{\text{TD}}(\theta) + \lambda \mathcal{R}_{\text{ord}}(V_\theta; \mathcal{D}_k) + \mu \mathcal{R}_{\text{sym}}(V_\theta, g_\phi)$, where $\mathcal{R}_{\text{ord}}$ penalizes violations of the learned partial order (i.e., $x \preceq_{\mathcal{D}_k} y$ but $V_\theta(x) > V_\theta(y)$), and $\mathcal{R}_{\text{sym}}$ enforces consistency under the learned automorphism operator $g_\phi$ so that antisymmetry is upheld on the quotient set $X/_\sim$. We provide rigorous theoretical guarantees for this approach, including order-induced residual reduction and monotone refinement (Theorems 3.3 to 3.6), recovery and identifiability of the relevant equivariant structure (Theorems 3.9 and 3.10), variance reduction and bias–variance control (Theorems 3.2 and 3.8), and a convergence rate of $O(\sqrt{\mathfrak{R}(N)/N})$ (Theorem 3.10).

Second, we develop a *hard* enforcement approach that maintains a valid poset at every refinement step by construction. Specifically, the method alternates between (i) updating the automorphism operator and forming the quotient set $X/_\sim$, (ii) updating super-poset refinements by adding new order relations inferred from TD signals, and (iii) applying a constrained projection onto the feasible set induced by the current quotient-poset order (so that the value estimates satisfy the order constraints on $X/_\sim$ exactly). This hard-enforcement variant directly enforces reflexivity/transitivity and guarantees antisymmetry via quotienting, while retaining the refinement mechanism driven by TD comparisons; its formal properties are analyzed through tangent–normal feasibility, restricted descent, and oracle speedup guarantees (Theorems 4.2 to 4.4).

Our key contributions are as follows:

- We view value functions through the lens of order theory as learning a desired poset. It allows us to introduce a new process of learning a sequence of super-poset refinements, thus preserving the geometric coherence underpinning the learned value functions in RL.

- We propose GCR-RL to regularize standard RL by ensuring reflexivity, transitivity, and antisymmetry properties of posets. In particular, antisymmetry is achieved by learning algebraic operators of automorphism and eliminates value update loops for faster learning.

- We develop two methods for enforcing geometric coherence in GCR-RL, by coherence-regularized TD learning as a soft enforcement and by quotient-poset constrained projection as a hard enforcement.

- We empirically evaluate GCR-RL on grid, Minigrid, Atari, and non-transitive chain tasks, showing significant improvements in sample efficiency and performance, as well as reduced Bellman residuals and more stable value learning over strong baselines.

## 2. Preliminaries

Value-based deep RL (e.g., DQN/Double-DQN/Rainbow) remains vulnerable to instability under function approximation and off-policy TD learning (Mnih et al., 2015; Van Hasselt et al., 2016; Hessel et al., 2018; Tsitsiklis & Van Roy, 1996; Baird et al., 1995). We defer broader background and discussion to Appendix A.

Two geometric lines of work provide partial remedies that we leverage as building blocks. (*Symmetry*) Group-equivariant architectures hard-code known groups (Cohen & Welling, 2016; Kondor & Trivedi, 2018; Finzi et al., 2020; Bronstein et al., 2021), while data-augmentation RL (RAD/DrQ/CURL) often transforms states without coupled action relabeling, risking invariance when equivariance is required (Laskin et al., 2020a; Yarats et al., 2021; Laskin et al., 2020b). (*Order*) Isotonic/monotone projections enforce partial orders (Barlow, 1972) and can be embedded via differentiable optimization layers (Amos & Kolter, 2017; Agrawal et al., 2019; Blondel et al., 2020), while DAG learning techniques help avoid cycles (Zheng et al., 2018) that are known to induce oscillatory dynamics in non-transitive games (Shapley, 1953; Balduzzi et al., 2018). Our approach unifies both: we learn near-equivariant transforms with action permutations and a TD-evidenced poset on symmetry-quotiented elements.

We consider a discounted Markov Decision Process (MDP) $\mathcal{M} = (\mathcal{S}, \mathcal{A}, P, r, \gamma)$ with state space $\mathcal{S}$, action set $\mathcal{A}$, transition kernel $P(\cdot \mid s, a)$, immediate reward $r : \mathcal{S} \times \mathcal{A} \to \mathbb{R}$, and discount $\gamma \in (0, 1)$. For a policy $\pi(a \mid s)$, we consider $Q^\pi(s, a) = \mathbb{E}_\pi\left[\sum_{t \geq 0} \gamma^t r(s_t, a_t) \mid s_0 = s, a_0 = a\right]$. The optimal $Q^*$ is obtained as a fixed point of the Bellman operator: $(\mathcal{T}^* Q)(s, a) = r(s, a) + \gamma \mathbb{E}_{s' \sim P(\cdot \mid s, a)}[\max_{a'} Q(s', a')]$. For a function approximator $Q_\theta : \mathcal{S} \to \mathbb{R}^{|\mathcal{A}|}$, we denote

$$V_\theta(s) = \max_{a \in \mathcal{A}} Q_\theta(s, a). \tag{1}$$

**RL as a sequence of super-poset refinements.** Let $X \subseteq \mathcal{S} \times \mathcal{A}$ denote the set of state–action pairs under consideration. Our order-theoretic view models value learning as constructing a *target poset* over the relevant elements:

TD evidence provides comparisons that progressively refine an underlying partial order. Formally, a poset is a pair $(\bar{X}, \preceq)$ where $\bar{X}$ is the ground set (in our case, a symmetry-quotiented version of $X$) and $\preceq$ is a partial order. *Reflexivity* means every element is comparable to itself ($u \preceq u$); *transitivity* means that consistent multi-step comparisons compose ($u \preceq v$ and $v \preceq w \Rightarrow u \preceq w$). *Antisymmetry* prevents value circles: if $u \preceq v$ and $v \preceq u$, then $u$ and $v$ must represent the same element in the poset. Over training we obtain a sequence $\{(\bar{X}, \preceq_k)\}_{k \geq 0}$ where $\preceq_k$ *refines* $\preceq_{k-1}$ (i.e., $\preceq_{k-1} \subseteq \preceq_k$); we refer to this as a sequence of *super-poset refinements*.

**Automorphisms and equivalence classes.** To ensure antisymmetry in the presence of symmetries, we define the poset on *equivalence classes* of state–action pairs: configurations that are the same up to an MDP symmetry are treated as a single element in the poset. We formalize such symmetries via state–action transformations.

Let $\mathcal{G}$ be a family of paired transformations $g = (T_g, \Pi_g)$ with $T_g : \mathcal{S} \to \mathcal{S}$ acting on states and $\Pi_g : \mathcal{A} \to \mathcal{A}$ permuting actions. We say the MDP is compatible with $\mathcal{G}$ as follows.

**Definition 2.1** ($\mathcal{G}$-invariance). An MDP is $\mathcal{G}$-invariant if, for all $g \in \mathcal{G}$ and $(s, a)$,

$$r(T_g s, \Pi_g a) = r(s, a), P(\cdot \mid T_g s, \Pi_g a) = (T_g)_\# P(\cdot \mid s, a), \tag{2}$$

where $(T_g)_\# P$ denotes the pushforward of $P$ under $T_g$.

Each $g \in \mathcal{G}$ can be viewed as an *automorphism* of the MDP: it preserves rewards and transitions and hence preserves optimal values. Under Theorem 2.1,

$$Q^*(T_g s, \Pi_g a) = Q^*(s, a), \qquad \forall g \in \mathcal{G}, \ (s, a). \tag{3}$$

For each $(s, a)$, the orbit is $\mathcal{O}_\mathcal{G}(s, a) = \{(T_g s, \Pi_g a) : g \in \mathcal{G}\}$, inducing an equivalence relation $\sim$. We write $\bar{X} := X/_\sim$ for the quotient set; $\bar{X}$ will be the ground set of our poset.

**TD-induced partial orders.** Given $\bar{X}$, TD learning provides local order information through TD signals. Over a batch, we collect confident pairwise suggestions of the form $u$ should have no smaller value than $v$, remove directed cycles, and obtain a DAG $D$. The edges of $D$ induce a relation $\preceq_D$ on $\bar{X}$ by reachability: $u \preceq_D v$ if there is a directed path from $u$ to $v$. By construction $\preceq_D$ is reflexive and transitive; on the quotient $\bar{X}$, acyclicity together with quotienting supports antisymmetry. As training progresses and more reliable edges are added, this yields the desired sequence of super-poset refinements.

**A monotone cone and the feasible region.** In the embedded Euclidean space, the set of value vectors that respect the order induced by $D$ forms a monotone cone

$$\mathsf{Mono}(D)$$
$$= \Big\{ V : \ V(u) \geq V(v) + \delta \text{ for all } (u \to v) \in D, \ \delta \geq 0 \Big\}. \tag{4}$$

On the symmetry side, the equivariant subspace is

$$\mathsf{Eq}(\mathcal{G}) = \big\{ Q : \ Q(T_g s, \Pi_g a) = Q(s, a) \text{ for all } g, (s, a) \big\}. \tag{5}$$

Together they define the algebraic poset geometry as the intersection

$$\mathcal{F} := \mathsf{Eq}(\mathcal{G}) \cap \mathsf{Mono}(D), \tag{6}$$

i.e., value vectors consistent with both automorphism-induced identifications and TD-induced order constraints.

## 3. Learned Symmetry and Logic-Order Regularization

Section 2 casts value learning as constructing a sequence of super-poset refinements on symmetry-quotiented elements: automorphisms identify equivalence classes, while TD evidence induces an evolving partial order. Hard projections onto the feasible region $\mathcal{F} = \mathsf{Eq}(\mathcal{G}) \cap \mathsf{Mono}(D)$ are brittle early in training because both $\mathcal{G}$ and $D$ are unknown. This section therefore presents a *soft* enforcement of geometric coherence: we augment a standard off-policy TD learner with (i) a learned *symmetry* module that approximates automorphisms (softly encouraging $\mathsf{Eq}(\mathcal{G})$), and (ii) a learned *logic-order* module that extracts an acyclic partial order from TD signals and aligns values to it differentiably (softly encouraging $\mathsf{Mono}(D)$). We also establish the key analytical messages needed later: the symmetry loss recovers genuine automorphisms under invariance and reduces estimation variance (this subsection), the logic-order loss yields a valid acyclic order surrogate (Section 3.2), and together they support the convergence-rate results in Section 3.5.

Given a minibatch $\mathcal{B} = \{(s_i, a_i, r_i, s_i')\}_{i=1}^B$, the target network provides one-step TD targets and the associated target-relative nudges

$$y_i = r_i + \gamma \max_{a'} Q_{\bar{\theta}}(s_i', a'), \qquad \Delta_i := y_i - V_\theta(s_i), \tag{7}$$

where $V_\theta(s) = \max_a Q_\theta(s, a)$. We use $\Delta_i$ (with confidence weights) to construct a batchwise preference DAG for order learning, while the symmetry module learns transform–relabel pairs that approximate latent automorphisms.

We next detail the two components in turn: Section 3.1 develops the learned symmetry module; Section 3.2 constructs differentiable order alignment from TD evidence.

### 3.1. Learnable near-equivariance and action relabeling

We next develop the learned symmetry module, whose goal is to discover near-automorphisms and enforce (approximate) equivariance so that value estimates are nearly constant along symmetry orbits, providing a soft surrogate of $\mathsf{Eq}(\mathcal{G})$ needed to uphold antisymmetry on the quotient.

Let $z = f_\theta(s) \in \mathbb{R}^d$ be the encoder output and let $Q_\theta(s, \cdot) = h_\theta(z) \in \mathbb{R}^{|\mathcal{A}|}$ be the action–value head. We introduce $K$ paired transformations in representation/action space:

$$T_k z = W_k z, \qquad W_k \in \mathbb{R}^{d \times d},$$
$$\Pi_k \in \mathbb{R}^{|\mathcal{A}| \times |\mathcal{A}|} \quad \text{(doubly-stochastic via Sinkhorn).} \tag{8}$$

Each $(W_k, \Pi_k)$ is intended to approximate one latent symmetry: $W_k$ transforms features, and $\Pi_k$ relabels action coordinates so the entire value vector transforms coherently.

**Equivariance consistency loss.** We penalize deviations from the equivariance condition $Q(s, \cdot) \approx \Pi_k^\top Q(T_k s, \cdot)$ via

$$\mathcal{L}_{\text{eq}} = \frac{1}{KB} \sum_{k=1}^{K} \sum_{i=1}^{B} \left\| Q_\theta(s_i, \cdot) - \Pi_k^\top Q_\theta(T_k s_i, \cdot) \right\|_2^2. \tag{9}$$

To reflect that symmetries can be local, we use applicability weights $\alpha_{i,k} \in [0, 1]$ and optimize the normalized local version

$$\mathcal{L}_{\text{eq}}^{\text{local}} = \frac{1}{K} \sum_{k=1}^{K} \frac{\sum_{i=1}^{B} \alpha_{i,k} \left\| Q_\theta(s_i, \cdot) - \Pi_k^\top Q_\theta(T_k s_i, \cdot) \right\|_2^2}{\sum_{i=1}^{B} \alpha_{i,k} + 10^{-8}}. \tag{10}$$

(A concrete construction of $\alpha_{i,k}$ is deferred to Appendix F.1.)

**Group-like regularization.** To keep learned transforms non-degenerate and encourage group-like structure, we add soft penalties:

$$\mathcal{R}_{\text{id}} = \|W_1 - I\|_F^2 + \|\Pi_1 - I\|_F^2 + \frac{1}{K} \sum_{k=1}^{K} \|W_k^\top W_k - I\|_F^2,$$

$$\mathcal{R}_{\text{clo}} = \mathbb{E}_{i,j} \min_l \left( \|W_i W_j - W_l\|_F^2 + \|\Pi_i \Pi_j - \Pi_l\|_F^2 \right),$$

$$\mathcal{R}_{\text{inv}} = \mathbb{E}_i \min_l \left( \|W_i^\top - W_l\|_F^2 + \|\Pi_i^\top - \Pi_l\|_F^2 \right),$$

$$\mathcal{R}_{\text{ord}} = \frac{1}{K} \sum_{i=1}^{K} \sum_{m \in \mathcal{M}} \left( \|W_i^m - I\|_F^2 + \|\Pi_i^m - I\|_F^2 \right), \tag{11}$$

where $\mathcal{M} \subset \{2, 3, 4, \dots\}$ is a small set of desired finite orders. Implementation details (Sinkhorn parametrization, optional orthogonal retractions, and diversity regularization) are standard and omitted here for brevity. These terms have distinct roles. The identity and near-orthogonality terms prevent degenerate feature transforms; closure and inverse penalties make the transform bank behave like a coherent finite family rather than unrelated augmentations; the finite-order term favors simple rotations/reflections; permutation regularization keeps action relabelings near discrete assignments; and diversity regularization prevents all learned transforms from collapsing to the identity. Together they make the symmetry branch usable for quotienting state–action elements before order constraints are imposed.

We define the loss for the learned symmetry module as

$$\mathcal{L}_{\text{sym}} := \mathcal{L}_{\text{eq}}^{\text{local}} + \gamma_{\text{grp}} \left( \mathcal{R}_{\text{id}} + \mathcal{R}_{\text{clo}} + \mathcal{R}_{\text{inv}} + \mathcal{R}_{\text{ord}} \right)$$
$$+ \gamma_{\text{perm}} \mathcal{R}_{\text{perm}} + \gamma_{\text{div}} \mathcal{R}_{\text{div}}. \tag{12}$$

This loss is a soft surrogate of $\mathsf{Eq}(\mathcal{G})$: it encourages orbit-consistent values without requiring explicit quotienting.

**Theorem 3.1** (Equivariance under invariance ). *If the MDP is $\mathcal{G}$-invariant (Theorem 2.1) and the parametrization $\{(W_k, \Pi_k)\}_{k=1}^{K}$ can represent a finitely generated subgroup of $\mathcal{G}$, then driving $\mathcal{L}_{\text{sym}} \to 0$ yields a near-equivariant solution: $Q_\theta$ is (approximately) constant along $\mathcal{G}$-orbits and aligns with $Q^*$ on these orbits up to identifiable isomorphism.*

This result justifies using $\mathcal{L}_{\text{sym}}$ as a surrogate for learning automorphisms. It is later used to support the quotient-based antisymmetry arguments (i.e., identifying elements before imposing order constraints).

**Theorem 3.2** (Effective sample amplification ). *If symmetry covers a fraction $\rho$ of the empirical distribution, then training with $\mathcal{L}_{\text{eq}}^{\text{local}}$ increases the effective sample size on symmetric regions by approximately $1 + (K-1)\rho$, yielding a corresponding reduction in estimation-variance bounds (up to constants).*

This theorem explains why the symmetry module improves stability and sample efficiency: it shares TD information within learned orbits, and the resulting variance reduction is a key ingredient used later in the convergence-rate analysis.

We next turn to the second geometric component: constructing a cycle-free partial order from TD evidence and defining a differentiable order-alignment loss (Section 3.2).

### 3.2. Preference Graph and DAGification

We next construct a *cycle-free* batchwise partial order from TD evidence. Concretely, the logic-order branch (i) assigns confidence-weighted scores to candidate comparisons, and (ii) selects a high-weight acyclic subgraph $D$ (a DAG) that serves as the order skeleton for the monotone-cone alignment in Section 3.3.

**Scoring TD preferences.** Given $\Delta_i$ in (7), we compute a confidence-weighted score

$$w_i = \sigma(\Delta_i/\tau)\,c_i, \qquad c_i = \exp\big(-\mathrm{Var}_{a'}Q_{\bar{\theta}}(s'_i, a')\big), \tag{13}$$

where $\sigma$ is logistic, $\tau > 0$ is a temperature, and $c_i$ down-weights transitions whose next-state bootstraps disagree across actions. We restrict candidate comparisons to $k$-nearest neighbours in representation space (cosine similarity on $f_\theta(s)$ to avoid spurious long-range ties, and retain only a top-$M$ subset using a batch-adaptive percentile cutoff (e.g., the $p$-th percentile of $\{w_i\}$).

Let $E_{\mathrm{cand}}$ be the surviving candidate edges with weights $w(\cdot)$. We sort $E_{\mathrm{cand}}$ by decreasing weight and add edges one-by-one, skipping any edge that would create a directed cycle. This produces a DAG $D$.

The greedy rule can be viewed as approximating the maximum-weight acyclic subgraph problem: $\max_{D \subseteq E_{\mathrm{cand}}} \sum_{e \in D} w(e)$ s.t. $D$ is acyclic. We therefore define the (non-differentiable) DAGification loss as the negative retained weight $\mathcal{L}_{\mathrm{dag}} := -\frac{1}{|D|} \sum_{e \in D} w(e)$, which measures how much high-confidence TD evidence is preserved under the acyclicity constraint.

**Theorem 3.3** (Greedy DAGification). *The greedy add-unless-cycle procedure outputs an acyclic graph $D$. Moreover, $D$ is maximal with respect to the chosen edge order: adding any skipped candidate edge would create a directed cycle.*

This theorem is used to justify that the downstream monotone alignment is always well-posed on a DAG skeleton (Section 3.3). Maximality explains why the procedure preserves as many high-confidence comparisons as possible under acyclicity.

GCR-RL uses the same replay buffer, target network, and environment interaction budget as the base off-policy learner. If the representation dimension is $d$ and the action space has size $|\mathcal{A}|$, the symmetry branch adds $O(Kd^2 + K|\mathcal{A}|^2)$ parameters for $\{W_k, \Pi_k\}_{k=1}^{K}$. On the order side, candidate comparisons are restricted to local $k$-NN neighborhoods and then pruned to a sparse top-$M$ edge set before greedy DAGification, so the added graph cost scales with the retained candidate set rather than all $O(B^2)$ minibatch pairs. The isotonic step uses a small fixed number of inner updates over the retained DAG edges. Thus the additional work is structural and batch-local; in pixel-based settings, the shared encoder remains the dominant cost. With this DAG $D$ in hand, we next align values to the induced partial order in a differentiable way.

## 3.3. Differentiable Isotonic Projection on DAG

We next define a differentiable surrogate to the isotonic projection onto the monotone cone induced by $D$. This yields a soft enforcement of $\mathrm{Mono}(D)$ that can be backpropagated through and combined with standard TD learning.

On a fixed DAG $D$ over batch indices $\{1, \ldots, B\}$, we would like to minimally adjust $V_\theta$ while enforcing monotonicity on edges: $\hat{V}(u) \geq \hat{V}(v) + \delta$ for all $(u \to v) \in D$ with margin $\delta \geq 0$. The ideal operation is the Euclidean projection onto the cone:

$$\min_{\hat{V} \in \mathbb{R}^B} \quad \frac{1}{B} \sum_{i=1}^{B} \big(\hat{V}(i) - V_\theta(i)\big)^2 \tag{14}$$

$$\text{s.t.} \quad \hat{V}(u) \geq \hat{V}(v) + \delta, \qquad \forall (u \to v) \in D.$$

To make the step differentiable, we optimize the quadratic-penalty surrogate (with $[x]_+ = \max(0, x)$):

$$\begin{aligned} \mathcal{L}_{\mathrm{iso}} = \; & \frac{1}{B} \sum_{i=1}^{B} \big(\hat{V}(i) - V_\theta(i)\big)^2 \\ & + \mu \cdot \frac{1}{|E(D)|} \sum_{(u \to v) \in D} \big[\delta + \hat{V}(v) - \hat{V}(u)\big]_+^2, \end{aligned} \tag{15}$$

and optionally a soft ranking stabilizer

$$\mathcal{L}_{\mathrm{rank}} = \frac{1}{|E(D)|} \sum_{(u \to v) \in D} \log\big(1 + e^{\hat{V}(v) - \hat{V}(u)}\big). \tag{16}$$

We treat $\hat{V}$ as an auxiliary variable and run a fixed small number of GD/SGD steps on $\mathcal{L}_{\mathrm{iso}} + \lambda_{\mathrm{rank}}\mathcal{L}_{\mathrm{rank}}$, returning $\hat{V}^*$ as a differentiable approximation to the exact projection.

We define the loss for the isotonic alignment as

$$\mathcal{L}_{\mathrm{ord}} := \mathcal{L}_{\mathrm{iso}} + \lambda_{\mathrm{rank}}\mathcal{L}_{\mathrm{rank}}, \tag{17}$$

which is the soft surrogate we backpropagate through to align $V_\theta$ to the learned partial order.

**Theorem 3.4** (Penalty-method consistency). *For fixed $D$ and finite $B$, as $\mu \to \infty$, any limit point of minimizers of $\mathcal{L}_{\mathrm{iso}}$ satisfies the constraints in (14), and the solution approaches the exact isotonic projection.*

This theorem justifies $\mathcal{L}_{\mathrm{iso}}$ as a faithful surrogate of the monotone-cone projection, which is required by the soft enforcement viewpoint of this section.

**Theorem 3.5** (Order-consistency on a DAG). *For any DAG $D$, the relation induced by $\hat{V}^*$ on edges of $D$ is cycle-free. If $E(D)$ is contained in a ground-truth partial order, then $\hat{V}^*$ is consistent with that order on those edges.*

This result is used to support the logical flow that: once we build a DAG skeleton (Theorem 3.3), the monotone alignment will not re-introduce cyclic preferences and will preserve correct comparisons when the TD evidence matches a true order.

## 3.4. Overall Objective and Optimization

We next summarize how the two regularizers are optimized as a *soft enforcement* of the geometric coherence $\mathcal{F} = \mathsf{Eq}(\mathcal{G}) \cap \mathsf{Mono}(D)$ using loss functions, while keeping the underlying off-policy TD loop unchanged.

Recall that Sections 3.1 to 3.3 define the symmetry loss $\mathcal{L}_{\mathrm{sym}}$, the DAGification loss $\mathcal{L}_{\mathrm{dag}}$, and the order-alignment loss $\mathcal{L}_{\mathrm{ord}}$. We optimize the overall objective

$$\mathcal{L}_{\mathrm{total}} = \mathcal{L}_{\mathrm{TD}} + \lambda_{\mathrm{sym}}\mathcal{L}_{\mathrm{sym}} + \lambda_{\mathrm{dag}}\mathcal{L}_{\mathrm{dag}} + \lambda_{\mathrm{ord}}\mathcal{L}_{\mathrm{ord}}, \quad (18)$$

where $\mathcal{L}_{\mathrm{TD}}$ is the standard TD regression loss of the base learner (e.g., DQN/Double-DQN). The expanded form of all penalty terms and hyper-parameters (including group-like and permutation/diversity regularizers) is provided in Appendix B.

## 3.5. Theoretical Guarantees

We next show that the loss-based route yields (i) a decrease in expected Bellman residual under order alignment, (ii) reduced oscillation/variance in value iterates, and (iii) a controlled bias–variance trade-off. Finally, under standard statistical assumptions, the learned symmetry and partial order are recovered with a convergence rate.

We measure approximation quality by the expected optimal Bellman residual

$$\mathcal{E}_{\mathrm{Bell}}(\theta) := \mathbb{E}_{(s,a)}\big[\big(Q_\theta(s,a) - \mathcal{T}^* Q_\theta(s,a)\big)^2\big]. \quad (19)$$

**Theorem 3.6** (Residual decrease under order alignment). *Assume i.i.d. (or mixing) sampling and stable learning rates. For a batch DAG $D$ and the isotonic surrogate solution $\hat{V}^\star$, one step of training with $\lambda_{\mathrm{ord}} > 0$ satisfies*

$$\mathcal{E}_{\mathrm{Bell}}(\theta^+) \leq \mathcal{E}_{\mathrm{Bell}}(\theta) - C_1\,\mathrm{Viol}(D, \hat{V}^\star) + C_2\,\|\hat{V}^\star - V_\theta\|_2^2, \quad (20)$$

*for constants $C_1, C_2 > 0$, where $\mathrm{Viol}(D, \hat{V}^\star)$ is the average hinge-squared violation on edges of $D$.*

This theorem formalizes the role of $\mathcal{L}_{\mathrm{ord}}$: it reduces Bellman residual by directly penalizing the TD-supported order violations, while the fidelity term $\|\hat{V}^\star - V_\theta\|_2^2$ prevents over-correction.

**Theorem 3.7** (Stability improvement). *Let $\{V_{\theta_t}\}$ be the value iterates under the update in Algorithm 4. After adding the order regularizer ($\lambda_{\mathrm{ord}} > 0$), the within-window variance of $\{V_{\theta_t}\}$ admits an upper bound that decreases with edge coverage of $D$ and increases monotonically with smaller isotonic violations.*

This result captures the main message needed later: the order regularizer damps oscillatory behavior by discouraging inconsistent local comparisons, yielding smoother value dynamics.

**Theorem 3.8** (Bias–variance trade-off). *Let $\hat{V}$ denote the order-aligned values produced by the logic-order branch. Then the test error decomposes as $\mathbb{E}\big[(\hat{V} - V^*)^2\big] = \mathrm{Var}[\hat{V}] + \big(\mathbb{E}[\hat{V}] - V^*\big)^2$, where $\mathcal{L}_{\mathrm{sym}}$ and $\mathcal{L}_{\mathrm{ord}}$ reduce $\mathrm{Var}[\hat{V}]$ (by sample sharing and shape constraints), and the induced bias is controlled by the mismatch between the true environment structure and the enforced symmetry/order.*

This theorem is used to explain why the regularizers help in the regimes we target: when symmetry/order are approximately correct, the variance reduction dominates the added structural bias.

**Theorem 3.9** (Identifiability of near-equivariance). *If a true equivariance $(T^*, \Pi^*)$ exists and the representation class is expressive, then with sufficiently strong group-like regularization, the learned $\{(W_k, \Pi_k)\}$ recovers $(T^*, \Pi^*)$ up to isomorphism.*

This ensures the symmetry module is not merely a training trick: under standard conditions it identifies interpretable transformations consistent with the dynamics.

We finally ask whether the loss-based route recovers the correct symmetry and a cycle-free subset of the true partial order as the sample size grows.

**Theorem 3.10** (Joint structural consistency). *Under $\mathcal{G}$-invariance, expressive $Q_\theta$, and a margin/noise condition ensuring the signs of TD nudges are learnable, there exists a complexity term $\mathfrak{R}(N)$ such that with $N$ samples, both the equivariance residual and the average order-violation on the learned DAG satisfy (equivariance residual) + (order violation) $= O\left(\sqrt{\frac{\mathfrak{R}(N)}{N}}\right)$ with probability $1 - o(1)$.*

This theorem provides the convergence-rate message needed for the paper's logical flow; the full statement (assumptions and proof sketch) is deferred to Appendix C.

## 4. A Lossless Implementation: Manifold Enforcement

We now propose a *hard* (lossless) enforcement of geometric coherence. Instead of encouraging $Q_\theta$ to stay near $\mathcal{F}$ via auxiliary losses, we *constrain* each update to lie on a batch-wise constraint set that encodes (i) symmetry equalities and (ii) order inequalities. The key idea is to view this constraint set as a (batchwise) *manifold-with-corners* in value space: we update along its tangent directions to fit TD targets, and contract along normal directions to remove constraint violations. This yields a single geometric step per iteration without introducing additional regularization losses.

## 4.1. Constraint Set and Feasible Domain Structure

We next define the batchwise constraint manifold used for hard enforcement.

**Batch variables.** Given a minibatch $\mathcal{B} = \{s_i\}_{i=1}^{B}$, let $V \in \mathbb{R}^B$ denote the state-value vector $V(i) = V_\theta(s_i)$, and let $Q_\theta(s_i, \cdot) \in \mathbb{R}^{|\mathcal{A}|}$ be the action–value vector.

**Hard order constraints (from the DAG).** Let $D = ([B], \mathcal{E})$ be the acyclic preference graph produced on this batch. For each edge $e = (u \rightarrow v) \in \mathcal{E}$, we enforce the monotonic inequality

$$g_e(V) := \delta + V(v) - V(u) \leq 0, \qquad \delta \geq 0. \quad (21)$$

To express inequalities in a projection-friendly form, introduce nonnegative slack variables $\mu \in \mathbb{R}_{\geq 0}^{|\mathcal{E}|}$ and write $g(V) + \mu = 0$, where $g(V) \in \mathbb{R}^{|\mathcal{E}|}$ stacks $\{g_e(V)\}_{e \in \mathcal{E}}$.

**Hard symmetry constraints (from learned transforms).** For each learned transform $g = (T_g, \Pi_g)$, we enforce batchwise equivariance as

$$h_{g,i}(Q_\theta) := Q_\theta(s_i, \cdot) - \Pi_g^\top Q_\theta(T_g s_i, \cdot) = 0, \quad i \in [B], \quad (22)$$

and we stack all constraints as $h(Q_\theta) = 0$. (Implementation details for projecting $(T_g, \Pi_g)$ onto a valid near-group parameterization are deferred to Appendix D.1.)

**Feasible domain as a manifold-with-corners.** Collect the constraints into

$$c(V, \mu; \theta) = \begin{bmatrix} g(V) + \mu \\ h(Q_\theta) \end{bmatrix},$$
$$\mathcal{M} = \Big\{ (V, \mu) : c(V, \mu; \theta) = 0, \ \mu \geq 0 \Big\}. \quad (23)$$

We will compute projected updates using the (sparse) Jacobian $J_c = \partial c / \partial(V, \mu)$, so that tangent directions satisfy $J_c \Delta(V, \mu) = 0$ while normal directions reduce $\|c\|$.

**Theorem 4.1** (Feasibility). *If $\mathcal{E}$ is obtained by greedy add-unless-cycle and $\delta \geq 0$, then the order constraints admit a feasible solution (i.e., $\exists V$ such that $g(V) \leq 0$). If, in addition, the function class and learned transforms can realize (22) on the batch, then $\mathcal{M} \neq \emptyset$.*

Theorem 4.1 is the starting point for the hard-enforcement route: it guarantees that the subsequent manifold projection (tangent update + normal contraction) is well-defined on every batch, so we can enforce geometric coherence by construction rather than by auxiliary penalties.

## 4.2. Manifold Step and Tangent–Normal Decomposition in the Value Layer

We next give a *hard* geometric update that (i) improves TD fit and (ii) enforces the batchwise constraints by con-

struction. Let $\mathcal{M} = \{(V, \mu) : c(V, \mu; \theta) = 0, \ \mu \geq 0\}$ be the feasible domain from §4.1, where $c(V, \mu; \theta) = \big[g(V) + \mu; \ h(Q_\theta)\big]$.

Let $g_V = \nabla_V \mathcal{L}_{\mathrm{TD}}(V)$ and let $J_c$ be the Jacobian of $c$ w.r.t. $(V, \mu)$. The orthogonal projector onto the (value-layer) tangent space is

$$P_{\mathrm{tan}} = I - J_V^\top (J_V J_V^\top)^{-1} J_V, \quad (24)$$

where $J_V$ is the Jacobian of the stacked constraints $[g; h]$ w.r.t. $V$. We take the steepest TD descent direction restricted to feasible directions:

$$\Delta_{\mathrm{tan}} = -P_{\mathrm{tan}} \, g_V. \quad (25)$$

To contract constraint violations we take a minimum-norm correction

$$\Delta_{\mathrm{nor}} = -\lambda \, J_c^\dagger \, c(V, \mu; \theta), \quad (26)$$

where $J_c^\dagger$ is the Moore–Penrose pseudoinverse and $\lambda > 0$ is a contraction gain.

Combining (25)–(26), we update

$$V^+ = V + \eta\big(\Delta_{\mathrm{tan}} + \Delta_{\mathrm{nor}}\big), \quad \mu^+ = \big[\mu - \eta_\mu \, (g(V) + \mu)\big]_+. \quad (27)$$

In practice we optionally gate the normal term (enable $\Delta_{\mathrm{nor}}$ only when $\|c\|$ exceeds a small threshold) to reduce overhead when iterates are already near-feasible.

To establish that (27) is both constraint-safe and optimization-effective, we need two properties: (**A**) the update should reduce violations; (**B**) it should still descend on the TD objective, but only along feasible directions. The next two theorems provide exactly (**A**) and (**B**).

**Theorem 4.2** (Tangent feasibility and normal contraction). *For sufficiently small $\eta, \eta_\mu$ and any $\lambda > 0$, the tangent step satisfies $J_V \Delta_{\mathrm{tan}} = 0$. Moreover, letting $V_c(V, \mu) = \frac{1}{2}\|c(V, \mu; \theta)\|_2^2$, there exists $\kappa > 0$ such that $V_c(V^+, \mu^+) \leq V_c(V, \mu) - \kappa \|J_c^\dagger c(V, \mu; \theta)\|_2^2$, whenever $c(V, \mu; \theta) \neq 0$.*

Theorem 4.2 guarantees the *hard-enforcement* mechanism: tangent motion does not change the constraints to first order, while the normal term strictly contracts violations.

**Theorem 4.3** (Restricted TD descent and convergence on $\mathcal{M}$). *Assume that in a neighborhood of $\mathcal{M}$, the TD objective satisfies a PL condition on the tangent space: $\|P_{\mathrm{tan}} g_V\|_2^2 \geq 2\mu_T\big(\mathcal{L}_{\mathrm{TD}}(V) - \mathcal{L}_{\mathrm{TD}}^\star\big)$, and is $L_T$-smooth along tangent directions. Then for sufficiently small $\eta$ and $\lambda$, the update (27) satisfies*

$$\mathcal{L}_{\mathrm{TD}}(V^+) \leq \mathcal{L}_{\mathrm{TD}}(V) - \gamma \|P_{\mathrm{tan}} g_V\|_2^2 \quad (28)$$

*for some $\gamma > 0$. In particular, once $c(V, \mu; \theta) = 0$ (feasibility), (27) reduces to projected gradient descent and converges linearly at rate $1 - \mu_T / L_T$.*

---

**Algorithm 1** Group parameter contraction

---

1: **Projection:** $T_g \leftarrow \mathrm{Polar}(T_g)$, $\Pi_g \leftarrow \mathrm{ProjPerm}(\Pi_g)$.
2: **Closure:** for generators $g, h$, set $T_{gh} \leftarrow \mathrm{Polar}(T_g T_h)$, $\Pi_{gh} \leftarrow \mathrm{ProjPerm}(\Pi_g \Pi_h)$; enforce identity/inverses.
3: **Alignment:** anchor $T_g$ by Procrustes on $\{\phi_\theta(s_i)\}$ and refine $\Pi_g$ by a matching step.

---

Theorem 4.3 provides the optimization backbone: hard constraints do not stall learning; they convert TD learning into a principled descent on the constrained geometry.

### 4.3. Group-Parameter Closure

We next keep the symmetry equalities $h(Q_\theta) = 0$ usable by maintaining a *closed* and *well-conditioned* parameterization $g \mapsto (T_g, \Pi_g)$. We use a periodic three-stage map (projection $\rightarrow$ closure $\rightarrow$ alignment) on $\{T_g, \Pi_g\}$; the full description and proof are deferred to Appendix D.1.

### 4.4. Oracle Speedups

We finally summarize what hard enforcement buys when the constraints match the environment geometry. The full oracle statement (including constants and technical conditions) is given in Appendix D.2; here we only keep the main bound needed for the paper's message.

**Theorem 4.4** (Oracle speedup). *Assume the hard constraints are enforced with the* oracle *symmetry $\mathcal{G}$ and an oracle DAG $D^\star$. Let $\rho \in [0, 1]$ be the symmetry-pocket coverage and $W(D^\star)$ the width (max antichain size) of $D^\star$. Then, to reach expected squared Bellman residual $\leq \varepsilon$, the required samples satisfy*

$$N_{\mathrm{oracle}}(\varepsilon) = \widetilde{O}\left( \frac{W(D^\star)}{1 + (|\mathcal{G}| - 1)\rho} \cdot \frac{1}{\varepsilon} \right), \quad (29)$$

*and the iteration complexity improves by the restricted condition-number factor $\kappa/\kappa_T$ from Theorem 4.3.*

Theorem 4.4 isolates the two hard-enforcement gains: symmetry reduces statistical redundancy (effective sample amplification), while order constraints reduce degrees of freedom (from batch size to DAG width).

### 4.5. Unified 0-Loss Geometric Procedure

We next provide the minimal pseudocode that instantiates the manifold step (27) and the periodic group contraction (Algorithm 1). Implementation details (rank-deficient $J_c$, subsampling equalities, and batching) are deferred to Appendix D.3.

---

**Algorithm 2** 0-Loss geometric enforcement (hard coherence)

---

1: Sample a minibatch $\mathcal{B}$, compute TD targets and $\mathcal{L}_{\mathrm{TD}}$.
2: Build a batch DAG $D = ([B], \mathcal{E})$ and constraints $c(V, \mu; \theta)$.
3: **Value-layer step:** compute $g_V$, form $P_{\mathrm{tan}}$, take $\Delta_{\mathrm{tan}}$ and $\Delta_{\mathrm{nor}}$, update $(V, \mu)$ by (27), and backpropagate through $V \mapsto Q_\theta$.
4: **if** every $r$ steps **then**
5:    **Group step:** apply Algorithm 1.
6: **end if**

---

## 5. Experiments

We empirically evaluate **GCR-RL** on a set of control benchmarks chosen to probe the two geometric components in our analysis: near-equivariance and logic-order. Our goals are to test whether (i) learned near-equivariance improves sample efficiency on symmetry-rich tasks; (ii) logic-order regularization stabilizes learning in the presence of local cycles; and (iii) both modules *back off* when symmetry or order assumptions are weak so that performance never falls far below strong off-policy baselines.

We focus on three theory-aligned metrics that can be computed directly from evaluation logs: (i) sample efficiency (AUC@N and steps-to-threshold); (ii) asymptotic performance (final return); and (iii) across-seed stability. All definitions, environment details, and implementation choices are deferred to Appendix E.

### 5.1. Environments

We evaluate **GCR-RL** on four environment families designed to stress-test the two geometric components in our analysis—*near-equivariance* and *logic-order*—under diverse observation and noise regimes. Concretely, our suite includes: (i) a symmetry-rich gridworld with *controlled symmetry breaking*; (ii) an *action-permutation* task where state symmetry is weak but action relabeling dominates; (iii) *pixel-based* Atari games, spanning both near-symmetric and symmetry-poor titles; and (iv) a *noisy, cycle-injected* tabular MDP where local non-transitive loops are embedded in a global order. We additionally include Cartpole as a low-dimensional sanity check.

Table 1 summarizes the key properties; full specifications (layouts, rewards, and perturbation protocols) are deferred to Appendix E.1.

To keep the main text focused, we show two representative learning curves that correspond directly to our two geometric targets: symmetry-rich learning (E1) and cycle suppression under noise (E4). The complete set of learning curves is provided in Appendix E.6.

| Env. family | Obs. | Structure (knob) |
|---|---|---|
| RotMirror-Grid (E1) | grid | near-equiv. (state+action), $p_{asym}$ |
| BtnPerm-Minigrid (E2) | RGB | action relabeling (Π), episode perm. |
| Atari (E3) | pixels | symmetry-rich vs. symmetry-poor (game) |
| Noisy-RPS Chain (E4) | tabular | local cycles + global order, detour $\eta$ |

*Table 1.* Compact summary of evaluation environments (Appendix E.1).

| Benchmark | GCR-RL final return | Best non-ours final return | Main structure tested |
|---|---|---|---|
| E1 RotMirror-Grid | $-136.7 \pm 7.1$ | $-164.6 \pm 8.1$ | state–action symmetry |
| E2 BtnPerm-Minigrid | $3.70 \pm 0.00$ | $1.60 \pm 2.70$ | action relabeling |
| Atari Pong | $-1.00 \pm 0.10$ | $-4.00 \pm 0.00$ | near symmetry |
| Atari Breakout | $3.20 \pm 0.00$ | $2.60 \pm 0.30$ | near symmetry |
| Atari Asterix | $368.2 \pm 19.0$ | $249.2 \pm 2.9$ | weak symmetry |
| Atari Freeway | $1.60 \pm 1.00$ | $1.10 \pm 0.30$ | weak symmetry |
| Atari Space Inv. | $101.5 \pm 11.0$ | $59.5 \pm 13.5$ | weak symmetry |
| E4 Noisy-RPS Chain | $21.60 \pm 0.20$ | $20.70 \pm 0.10$ | cyclic order |

*Table 2.* Main-text scalar summary of final returns. Values are mean $\pm$ 95% CI over seeds. "Best non-ours" is the strongest baseline reported in Appendix E.

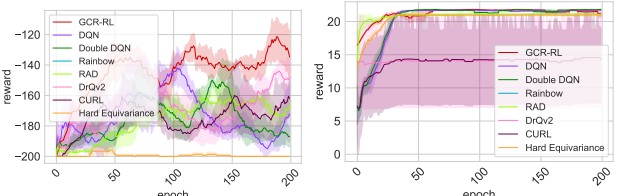

(a) RotMirror-Grid (E1): sym- (b) Noisy-RPS Chain (E4): local metry with controlled breaking non-transitive cycles under noise

*Figure 1.* Representative learning curves for **GCR-RL** and baselines. Full per-environment curves (including Atari and Minigrid) appear in Appendix E.6.

For E4, where ground-truth tabular values are computable, we additionally monitor order-consistency diagnostics during training. Given true values $V^*$, we define the pairwise violation rate and rank correlation by $\text{ViolRate}(V) = \frac{1}{|\mathcal{P}|} \sum_{(i,j) \in \mathcal{P}} \mathbf{1}\{(V_i^* - V_j^*)(V_i - V_j) < 0\}$, $\tau_K(V, V^*) = \text{KendallTau}(V, V^*)$, where $\mathcal{P}$ contains comparable state pairs on the chain. These diagnostics separate structural recovery from reward alone: a method can obtain transient reward while still flipping local order relations, whereas GCR-RL explicitly suppresses cycle-inducing comparisons before isotonic alignment.

Table 2 moves the key scalar results into the main text. The first two rows isolate the two symmetry mechanisms, while E4 isolates the order mechanism by embedding local non-transitive cycles inside a globally ordered chain. The Atari rows show that the same regularizer is not restricted to small synthetic tasks: it remains useful on both near-symmetric titles (Pong/Breakout) and visually heterogeneous titles with weaker symmetry (Asterix, Freeway, Space Invaders). We do not claim that these experiments exhaust large-scale RL; rather, they support the intended claim that learned quotienting and cycle-aware order alignment provide a practical structural bias across tabular, grid, RGB, and pixel-based regimes.

## 6. Conclusion

We presented GCR-RL, an order-theoretic view of value learning that builds a quotient poset over symmetry-aware elements and refines it via super-poset updates, while en-

forcing geometric coherence through near-equivariance with action relabeling and TD-implied partial orders. We developed both a soft regularization route and a hard, lossless enforcement route, and provided guarantees that clarify when the learned order and symmetry structure is consistent and how hard constraints prevent value-update loops. Across symmetry-rich and cycle-injected benchmarks, GCR-RL improves sample efficiency and stabilizes learning compared to strong off-policy baselines. Future work includes scaling online DAG construction and constraint enforcement, extending the approach to continuous-control and actor–critic settings, and leveraging the learned poset structure for exploration and planning.

## Limitations

GCR-RL is most useful when the data contain exploitable approximate symmetries or reliable directed order evidence. When both structures are weak, highly nonstationary, or systematically misidentified, the structural losses may add bias rather than useful regularization. The local applicability weights and confidence-filtered DAG construction are designed to reduce this risk, but they do not remove it completely. The method also adds computational work relative to a plain value-learning baseline. The extra cost comes from the transform/action-relabeling heads, sparse neighbor search, greedy DAGification, and a small fixed number of isotonic inner steps. These costs do not require additional environment interaction or a second learner, and the encoder remains the dominant cost in pixel-based experiments, but large action spaces or very high-dimensional representations may require more aggressive sparsification. Finally, the hard-constraint variant is included to clarify the exact quotient-poset geometry and to separate exact feasibility from the soft regularized implementation. In practice it can be numerically stiffer than the soft variant because equality and inequality constraints must remain feasible under noisy minibatch updates. Extending the approach to continuous-control actor–critic benchmarks and larger Atari suites is a natural next step.

## Acknowledgments

This work was supported in part by the Office of Naval Research (ONR) under Grant No. N00014-23-1-2532 and the Army Research Office (ARO) under Grant No. W911NF2420166.

## Impact Statement

This paper presents work whose goal is to advance the field of Machine Learning. There are many potential societal consequences of our work, none which we feel must be specifically highlighted here.

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

# A. Additional Related Work

Value-based deep RL has evolved from DQN and Double-DQN to stronger variants such as Rainbow, yet instability under function approximation and off-policy TD remains a central issue (Mnih et al., 2015; Van Hasselt et al., 2016; Hessel et al., 2018; Tsitsiklis & Van Roy, 1996; Baird et al., 1995). Pixel-based methods like RAD, DrQ/DrQ-v2, and CURL reduce variance via augmentation consistency or contrastive objectives, but typically transform states without relabeling actions, which can encourage learned *invariance* where true *equivariance* is required (Laskin et al., 2020a; Yarats et al., 2021; Laskin et al., 2020b). Preference-based approaches instead add external comparisons to learn reward models or rankings (Christiano et al., 2017), while structural RL—via abstraction, homomorphisms, and bisimulation—formalizes symmetry and aggregation but usually assumes known maps or metrics that are not learned end-to-end from TD signals (Dean & Givan, 1997; Ravindran, 2004; Li et al., 2006; Castro, 2020). Our method differs by bootstrapping both symmetry and partial order directly from standard TD evidence, without extra supervision or hand-crafted abstractions.

Group-equivariant networks provide strong inductive biases for known symmetry groups in vision and geometry (Cohen & Welling, 2016; Kondor & Trivedi, 2018; Finzi et al., 2020; Bronstein et al., 2021), but real RL environments often exhibit only approximate or context-dependent symmetries. This motivates learning *near-equivariant* transforms with soft group-like regularization, coupled to *action* relabeling via Sinkhorn-based differentiable permutations projected to exact assignments at test time (Sinkhorn, 1967; Cuturi, 2013; Mena et al., 2018). On the representation side, constraining transforms to (near) orthogonal manifolds via Cayley/Householder parameterizations stabilizes training and preserves geometry (Wen & Yin, 2013; Helfrich et al., 2018). Our symmetry module combines these pieces to infer state–action equivariance from interaction data rather than assuming a fixed group.

Monotone structure and isotonic regression provide a classical way to project estimates onto DAG-defined partial orders while staying close to the originals (Barlow, 1972). Monotonic mixing has been used in RL (e.g., QMIX) but typically relies on hand-specified factorizations or known orderings (Rashid et al., 2020). Recent work on differentiable optimization layers and convex surrogates enables embedding such projections into neural networks (Amos & Kolter, 2017; Agrawal et al., 2019; Blondel et al., 2020), while smooth characterizations of acyclicity (e.g., NOTEARS) make it practical to avoid cycles when learning graphs (Zheng et al., 2018). Cyclic and non-transitive interactions are known to induce oscillations in learning dynamics (Shapley, 1953; Balduzzi et al., 2018); our logic-order branch directly targets this issue by constructing a batchwise DAG from TD preferences and applying a differentiable isotonic surrogate, trading small, controllable bias for reduced variance and more stable value propagation.

# B. Full Objective and Hyper-parameters

Let $\mathcal{L}_{\mathrm{RL}}$ be the standard TD loss. The full training objective adds the equivariance and order regularizers to it:

$$
\begin{aligned}
\mathcal{L}_{\mathrm{total}} = {} & \mathcal{L}_{\mathrm{RL}} + \lambda_{\mathrm{eq}}\mathcal{L}_{\mathrm{eq}} \\
& + \gamma_{\mathrm{grp}}(\mathcal{R}_{\mathrm{id}} + \mathcal{R}_{\mathrm{clo}} + \mathcal{R}_{\mathrm{inv}} + \mathcal{R}_{\mathrm{ord}}) \\
& + \lambda_{\mathrm{iso}}(\mathcal{L}_{\mathrm{iso}} + \beta\,\mathcal{L}_{\mathrm{rank}}) \\
& + \lambda_{\mathrm{perm}}\mathcal{R}_{\mathrm{perm}} \\
& + \lambda_{\mathrm{div}}\mathcal{R}_{\mathrm{div}},
\end{aligned}
\tag{30}
$$

where $\lambda_{\mathrm{eq}}$ controls how strongly we pull $Q_\theta$ toward the learned equivariant subspace $\mathsf{Eq}(\mathcal{G})$; $\gamma_{\mathrm{grp}}$ keeps the transforms well-behaved and close to a finite group; $\lambda_{\mathrm{iso}}$ balances order enforcement against fidelity to $V_\theta$, acting as a monotone brake that damps oscillations; $\beta$ sets the strength of the smooth ranking term; $\lambda_{\mathrm{perm}}$ sharpens action relabeling toward true permutations; and $\lambda_{\mathrm{div}}$ prevents all transforms from collapsing to a single symmetry. In practice we use moderate $\lambda_{\mathrm{eq}}$ and small $\lambda_{\mathrm{iso}}$ at the beginning, then anneal $\lambda_{\mathrm{iso}}$ (and the penalty weight $\mu$ in (15)) upward as TD noise decreases.

A single training step proceeds as follows:

1. Sample a minibatch $\mathcal{B} = \{(s_i, a_i, r_i, s_i')\}_{i=1}^B$ and compute TD targets $y_i$ and nudges $\Delta_i$ via (7).

2. Convert $\{\Delta_i\}$ and confidence scores into candidate edges and greedily DAGify to obtain $D$ as in Section 3.2.

3. Run $T_{\mathrm{iso}} \in [1, 5]$ gradient steps on $\hat{V}$ to minimize $\mathcal{L}_{\mathrm{iso}} + \beta\mathcal{L}_{\mathrm{rank}}$ and obtain $\hat{V}^\star$ (Section 3.3).

4. Evaluate $\mathcal{L}_{\mathrm{total}}$ using $Q_\theta$, the equivariance branch, and the isotonic surrogate, and backpropagate through the entire network.

Geometrically, $\mathcal{L}_{\mathrm{eq}}$ keeps updates close to the learned equivariant subspace and $\mathcal{L}_{\mathrm{iso}}$ keeps them inside a batchwise approximation of the monotone cone. The combination nudges $Q_\theta$ toward the structured region $\mathcal{F} = \mathsf{Eq}(\mathcal{G}) \cap \mathsf{Mono}(D)$ while leaving the underlying off-policy loop untouched. We next study how this affects Bellman residuals, stability, and bias–variance trade-offs.

We finally explain why these additions lower the Bellman residual and steady the value iterates.

## C. Joint Structural Consistency: Full Statement

We measure approximation quality via the expected Bellman residual

$$\mathcal{E}_{\mathrm{Bell}}(\theta) = \mathbb{E}_{(s,a)}\Big[\big(Q_\theta(s,a) - \mathcal{T}^* Q_\theta(s,a)\big)^2\Big]. \tag{31}$$

The following results summarize how monotonicization and equivariance regularization shape this residual and the dynamics of value iterates.

**Theorem C.1** (Upper bound under monotonicization). *Under an i.i.d. approximation and stable learning rates, if $\hat{V}$ is obtained via the DAG isotonic step of Section 3.3 and edges cover high-confidence local orders, then there exist constants $C_1, C_2 > 0$ such that one-step update satisfies*

$$\mathcal{E}_{\mathrm{Bell}}(\theta^+)$$
$$\leq \mathcal{E}_{\mathrm{Bell}}(\theta) - C_1 \cdot \frac{1}{|E(D)|} \sum_{(u \to v) \in D} \big[\delta + \hat{V}(v) - \hat{V}(u)\big]_+^2 \tag{32}$$
$$+ C_2 \|\hat{V} - V_\theta\|_2^2.$$

*For sufficiently large $\mu$ and $\delta \geq 0$, the second term is controlled by $\mathcal{L}_{\mathrm{iso}}$ in (15), leading to an overall decrease in the expected residual.*

Edges that violate monotonicity indicate precisely where the targets want $V$ to move up or down relative to its neighbours; the isotonic step smooths out these violations and reduces Bellman error. The deviation term $\|\hat{V} - V_\theta\|_2^2$ captures how far we are allowed to move along the cone; the penalty weight $\mu$ keeps this cost from overwhelming the benefits, so the net effect is a residual decrease.

**Theorem C.2** (Acyclic stability and oscillation suppression). *If local cycles arise from non-transitive (RPS-like) structure or noise, then after DAGification (Section 3.2) and isotonic projection (Section 3.3), the variance upper bound of $\{V_{\theta_t}\}$ in a batch window decreases monotonically with $\lambda_{\mathrm{iso}}$ and edge coverage. When edges cover dominant directions of the true partial order, the bound is reduced below a fixed fraction of the unregularized case.*

Cycles act as feedback loops that let values bounce among a few states. DAGification cuts these loops; isotonic projection converts the remaining edges into monotone "ramps" that discourage backtracking. Increasing $\lambda_{\mathrm{iso}}$ steepens these ramps and suppresses oscillation, which matches the empirically smoother learning curves when combining Sections 3.2 and 3.3.

**Theorem C.3** (Bias–variance decomposition). *For the test error $\mathbb{E}\big[(V_\theta - V^*)^2\big]$,*

$$\mathbb{E}\big[(\hat{V} - V^*)^2\big] = \underbrace{\mathrm{Var}[\hat{V}]}_{\text{statistical variance}} + \underbrace{\big(\mathbb{E}[\hat{V}] - V^*\big)^2}_{\text{estimation bias}},$$

*where $\mathcal{L}_{\mathrm{eq}} + \mathcal{L}_{\mathrm{iso}}$ reduces variance (via sample sharing and shape constraints) while introducing a controllable* structural *bias determined by the mismatch between true symmetry/partial order and the enforced structure.*

$\mathcal{L}_{\mathrm{eq}}$ averages symmetric views inside approximate orbits; $\mathcal{L}_{\mathrm{iso}}$ ties values along high-confidence edges. Both shrink variance. Any additional bias is structural and scales with how far the learned equivariance and partial order deviate from reality. When these assumptions are approximately correct, the variance reduction dominates the added bias, echoing the effective sample amplification in Theorem 3.2.

**Theorem C.4** (Identifiability of near-equivariance). *If a true equivariance $(T^*, \Pi^*)$ exists and $f_\theta$ is expressive, then with sufficiently strong group-like regularization (Section 3.1), the learned $\{(W_k, \Pi_k)\}$ converges, up to isomorphism, to a finitely generated approximation of $(T^*, \Pi^*)$.*

Group-like penalties restrict the search space to near-group transformations; the equivariance loss then selects those that commute with the dynamics. With enough capacity, the limit matches the ground-truth symmetry up to relabeling, making the learned transforms themselves interpretable.

We finally ask whether the *loss-based* route in Section 3—equivariance consistency (9) plus group-like regularization (11) together with the DAG isotonic surrogate (15) inside (30)—recovers the correct symmetry and partial order as sampling grows.

**Theorem C.5** (Joint structural consistency of the loss-based route). *Suppose the MDP is $\mathcal{G}$-invariant in the sense of Theorem 2.1; the class $Q_\theta$ is expressive; and the learnable transforms $\{(W_k, \Pi_k)\}_{k=1}^K$ can approximate a finitely generated subgroup of $\mathcal{G}$. Assume i.i.d. (or uniformly mixing) off-policy sampling and a Massart-type noise condition for the one-step nudges $\Delta_i$ from (7) with margin $\gamma_0 > 0$:*

$$\Pr\big[\text{sign}(\Delta_i) \neq \text{sign}(\Delta_i^\star) \mid s_i\big] \ \leq \ \eta < \tfrac{1}{2},$$
$$\big|\mathbb{E}[\Delta_i \mid s_i]\big| \ \geq \ \gamma_0. \tag{33}$$

*Let the training minimize (30) with schedules that keep $\lambda_{\text{eq}}, \gamma_{\text{grp}}, \lambda_{\text{iso}} > 0$ fixed and gradually drive the isotonic penalty weight in (15) to $\mu \to \infty$. Then there exists a capacity term $\mathfrak{R}(N)$ (e.g., a joint Rademacher complexity of the class constrained by (9), (11), and the DAG isotonic surrogate) and a constant $C > 0$ such that for $N$ samples,*

$$\Pr\bigg( \underbrace{\max_{g \in \mathcal{G}} \big\|Q_\theta(T_g s, \Pi_g a) - Q_\theta(s,a)\big\|_2}_{\textit{equivariance residual}} + \underbrace{\frac{1}{|E(D)|} \sum_{(u \to v) \in D} \big[\delta + \hat{V}(v) - \hat{V}(u)\big]_+}_{\textit{order violations}} \leq C\sqrt{\frac{\mathfrak{R}(N)}{N}} \bigg) \xrightarrow[N \to \infty]{} 1. \tag{34}$$

*In particular, the learned $\{(W_k, \Pi_k)\}$ converges (up to isomorphism) to the true equivariance, and the greedy DAG edges from Section 3.2 concentrate to a cycle-free subset of the ground-truth partial order with vanishing error probability.*

Uniform convergence of the constrained hypothesis class gives the $O(\sqrt{\mathfrak{R}(N)/N})$ rate; the margin and Massart condition recover the signs of $\Delta_i$, so by Theorem 3.3 the greedy DAG stabilizes to a true cycle-free subset. Group-like regularization and equivariance loss select the transforms that commute with the dynamics (cf. Theorems 3.1 and 3.9). Combined with the variance and identifiability results above, Theorem 3.10 shows that the loss-based formulation in Section 3 converges, in the large-sample limit, to the same geometric objects that underlie the lossless update of Section 4.

## D. Supplementary Material for Lossless Manifold Enforcement

### D.1. Group-Parameter Contraction: Projection, Closure, and Alignment

This appendix provides the complete version of the group-parameter contraction summarized in Algorithm 1.

**Operators.** We use three standard operators.

- **Polar projection (nearest orthogonal).** For any matrix $A \in \mathbb{R}^{d \times d}$ with SVD $A = U\Sigma V^\top$,

$$\text{Polar}(A) \ = \ UV^\top \in O(d).$$

- **Sinkhorn scaling (approximate doubly-stochastic).** Given logits $L \in \mathbb{R}^{m \times m}$, define

$$S_0 = \exp(L), \quad S_{t+1} = \mathcal{N}_{\text{row}}\big(\mathcal{N}_{\text{col}}(S_t)\big),$$

where $\mathcal{N}_{\text{row}}$ and $\mathcal{N}_{\text{col}}$ normalize rows/cols to sum to one. After $T$ iterations, $S_T$ is approximately doubly-stochastic.

- **Permutation projection (nearest permutation).** Given a doubly-stochastic matrix $S$, define

$$\text{ProjPerm}(S) \ = \ \arg\min_{\Pi \in \mathcal{P}_m} \|S - \Pi\|_F^2,$$

which is solved by the Hungarian algorithm (equivalently a linear assignment).

---

**Algorithm 3** Projection–Closure–Alignment for $\{(T_g, \Pi_g)\}$

---

1: **Input:** current $\{T_g, \Pi_g\}_{g \in \mathcal{S}}$, batch $\{s_i\}_{i=1}^B$, representation $\phi_\theta(\cdot)$, interval $r$, closure pairs $\mathcal{P}$.
2: **(I) Projection:**
3: **for** each generator $g \in \mathcal{S}$ **do**
4: $\quad T_g \leftarrow \mathrm{Polar}(T_g)$.
5: $\quad \Pi_g \leftarrow \mathrm{ProjPerm}(\mathrm{Sinkhorn}(L_g))$ (or $\Pi_g \leftarrow \mathrm{ProjPerm}(\Pi_g)$).
6: **end for**
7: **(II) Closure contraction:**
8: **for** each $(g, h) \in \mathcal{P}$ **do**
9: $\quad T_{gh} \leftarrow \mathrm{Polar}(T_g T_h), \quad \Pi_{gh} \leftarrow \mathrm{ProjPerm}(\Pi_g \Pi_h)$.
10: **end for**
11: Enforce identity/inverses: $T_e = I, \Pi_e = I; T_{g^{-1}} = T_g^\top, \Pi_{g^{-1}} = \Pi_g^\top$.
12: **(III) Alignment (data-anchoring):**
13: **for** each generator $g \in \mathcal{S}$ **do**
14: $\quad$ Compute $C_g = \sum_{i=1}^B \phi_\theta(T_g s_i) \phi_\theta(s_i)^\top = U \Sigma V^\top$.
15: $\quad T_g \leftarrow U V^\top$ $\hfill$ (Procrustes alignment).
16: $\quad$ Refine $\Pi_g$ by assignment using a cost matrix $M_g$ derived from batch value discrepancies.
17: **end for**
18: **Output:** updated $\{(T_g, \Pi_g)\}$.

---

**Closure residuals.** Let $\mathcal{S}$ be a finite generating set and let $\mathcal{P} \subseteq \mathcal{S} \times \mathcal{S}$ be the set of pairs $(g, h)$ for which we enforce closure constraints. Define the closure residuals

$$r_T(g, h) = \|T_{gh} - T_g T_h\|_F, \qquad r_\Pi(g, h) = \|\Pi_{gh} - \Pi_g \Pi_h\|_F.$$

**Lemma D.1** (Nearest orthogonal via polar factor). *For any $A \in \mathbb{R}^{d \times d}$, $\mathrm{Polar}(A)$ solves $\min_{Q \in O(d)} \|A - Q\|_F$.*

**Lemma D.2** (Nearest permutation via assignment). *For any $S \in \mathbb{R}^{m \times m}$, the projection $\arg\min_{\Pi \in \mathcal{P}_m} \|S - \Pi\|_F^2$ is equivalent to a linear assignment and can be solved by the Hungarian algorithm.*

**Theorem D.3** (Closure consistency and batchwise equivariance control). *Assume the environment is $\mathcal{G}$-invariant (as in §2) and the parameterization is expressive enough to realize a finitely generated subgroup action on representations/actions. Fix any batch $\{s_i\}_{i=1}^B$ and any $\epsilon > 0$. Then, after one Projection–Closure–Alignment cycle (Algorithm 3), the following hold:*

1. *Algebraic feasibility (projection). $T_g \in O(d)$ and $\Pi_g \in \mathcal{P}_{|\mathcal{A}|}$ for all generators $g \in \mathcal{S}$.*

2. *Finite closure contraction. The closure residuals $r_T(g, h)$ and $r_\Pi(g, h)$ can be reduced below $\epsilon$ for all enforced pairs $(g, h) \in \mathcal{P}$.*

3. *Batch anchoring. The batchwise alignment error $\sum_{i=1}^B \|\phi_\theta(T_g s_i) - T_g \phi_\theta(s_i)\|_2^2$ is non-increasing under the Procrustes step and can be made $< \epsilon$ under mild conditioning of the batch covariance.*

4. *Equivariance residual control. Let $h(Q_\theta)$ denote the stacked equality constraints $Q_\theta(T_g s, \Pi_g a) - Q_\theta(s, a) = 0$ evaluated on the batch. Then $\|h(Q_\theta)\|_2$ can be reduced below $\epsilon$ after the cycle, up to approximation error of $Q_\theta$.*

**Proof sketch.** Items (1)–(2) follow from Lemmas D.1–D.2 and direct substitution in the closure updates ($T_{gh} \leftarrow \mathrm{Polar}(T_g T_h), \Pi_{gh} \leftarrow \mathrm{ProjPerm}(\Pi_g \Pi_h)$). Item (3) follows because Procrustes alignment is the least-squares minimizer over $O(d)$. Item (4) is obtained by combining (1)–(3) with $\mathcal{G}$-invariance: when transforms are algebraically stable and data-anchored, the induced action on $(s, a)$ commutes with the Bellman structure on the batch, so the residual is reduced to approximation error.

**Theorem D.4** (Restricted Convergence Rate and Spectral Contraction). *If the TD objective satisfies restricted smoothness and restricted strong convexity (or the PL condition) in a neighborhood of the feasible domain, then for sufficiently small step sizes, the update (27) converges linearly (or sublinearly), with the convergence constant depending only on the restricted condition number and the spectral properties of $P$ on the tangent space. Moreover, the local spectral radius satisfies $\rho(PJ) \leq \rho(J)$, with strict inequality whenever $J$ has components outside the tangent space.*

Thus restricting updates to the feasible domain not only preserves descent, but can also improve the rate and dampen oscillations by contracting the effective dynamics. In practice, this supports using moderately larger tangent step sizes together with gentle normal contraction coefficients to speed up learning without inducing instability.

**Theorem D.5** (Statistical Variance Reduction and Structural Bias Control). *On a distribution with symmetry pocket coverage rate $\rho$, equivariant alignment is equivalent to averaging over $K$ views, yielding an effective sample size of approximately $1 + (K - 1)\rho$. With additional monotonic constraints binding high-confidence edges, the test error decomposes as*

$$\mathbb{E}(V - V^*)^2 = \text{Var}[V] + \big(\mathbb{E}[V] - V^*\big)^2,$$

*where the variance term decreases due to sharing and restriction, and the bias term is determined by the mismatch between the imposed symmetry/monotonicity and the true environment. This bias can be kept small through closure–alignment and confidence gating.*

Geometric constraints therefore offer *statistical* benefits: symmetry sharing enlarges effective sample size; monotonicity constraints align local slopes and remove noisy cycles. This is consistent with the sample-amplification view in Section 3 and explains the smoother, lower-variance learning curves observed under the same interaction budget. In practice, this guides the choice of $K$, coverage thresholds, and edge-selection ratio $p$, and motivates strengthening closure–alignment at later stages to reduce structural bias further.

### D.2. Oracle Speedups: Full Statement and Derivation

This appendix provides the complete version of the oracle guarantee whose main message is stated in Theorem 4.4.

**Order width and degrees of freedom.** For a DAG $D^\star$, let $W(D^\star)$ denote its *width* (maximum antichain size). When monotonic constraints along $D^\star$ are enforced, the effective degrees of freedom in a batch of size $N_{\text{batch}}$ collapse from $N_{\text{batch}}$ toward $W(D^\star)$ via chain decomposition arguments.

**Theorem D.6** (Oracle sample complexity and rate improvements). *Assume the equalities/inequalities of §4.1 are enforced with the* oracle *symmetry group $\mathcal{G}$ and a ground-truth partial order $D^\star$. Let $\rho \in [0, 1]$ denote the symmetry-pocket coverage in the data distribution, and $W(D^\star)$ the width of $D^\star$. Let $N_{batch}$ be the number of comparison points per batch used by the DAG/order step. Assume the TD objective satisfies the PL condition on the* tangent *space with constants $(L_T, \mu_T)$ (cf. Theorem 4.3), and denote the unconstrained condition number by $\kappa = L/\mu$ and the restricted one by $\kappa_T = L_T/\mu_T \leq \kappa$. Then:*

(i) ***Effective sample complexity.*** *To reach expected squared Bellman residual $\leq \varepsilon$, the required number of distinct samples satisfies*

$$N_{\text{oracle}}(\varepsilon) \lesssim \frac{W(D^\star)}{1 + (|\mathcal{G}| - 1)\rho} \cdot \frac{C_T}{\varepsilon},$$

*for a problem-dependent constant $C_T$ determined by the restricted capacity on the tangent manifold. Relative to a symmetry/monotonicity-agnostic baseline $N_{\text{base}}(\varepsilon) \sim C/\varepsilon$, the multiplicative statistical speedup is at least*

$$\underbrace{1 + (|\mathcal{G}| - 1)\rho}_{\text{symmetry sharing}} \times \underbrace{\frac{N_{batch}}{W(D^\star)}}_{\text{order collapse}}.$$

(ii) ***Optimization rate.*** *With hard constraints, updates follow the restricted dynamics on the tangent space, so the linear rate improves from $1 - \mu/L$ to $1 - \mu_T/L_T$, reducing iteration complexity by a factor $\kappa/\kappa_T$.*

(iii) ***End-to-end budget.*** *Combining* (i) *and* (ii), *the wall-clock sample–iteration budget to reach $\varepsilon$ accuracy shrinks by at least*

$$\Big(1 + (|\mathcal{G}| - 1)\rho\Big) \cdot \frac{N_{batch}}{W(D^\star)} \cdot \frac{\kappa}{\kappa_T},$$

*recovering the $|\mathcal{G}|$-fold amplification when $\rho \approx 1$ and the near full-chain collapse when $W(D^\star) \ll N_{batch}$.*

**Derivation sketch.** (1) Symmetry sharing: under oracle equivariance, orbit-equivalent samples are statistically redundant; coverage $\rho$ determines how often we see full pockets, yielding the factor $1 + (|\mathcal{G}| - 1)\rho$. (2) Order collapse: hard monotone constraints reduce the effective degrees of freedom from batch size toward the width $W(D^\star)$ by standard poset chain/antichain arguments. (3) Restricted optimization: PL and smoothness on the tangent space yields linear convergence with condition number $\kappa_T$, and the constrained dynamics removes harmful off-tangent components, improving the rate factor $\kappa/\kappa_T$.

### D.3. Implementation Notes: Sparse Jacobians, Rank Deficiency, and Cost Control

This appendix records practical details needed to implement the 0-loss manifold step efficiently and stably.

**Constraint vector and sparse Jacobian.** On a batch of size $B$, define inequality constraints for each edge $e = (u \to v) \in \mathcal{E}$: $g_e(V) = \delta + V(v) - V(u) \leq 0$, and stack equalities $h(Q_\theta) = 0$ for selected orbit pairs under generators. Let $c(V, \mu; \theta) = [g(V) + \mu; \ h(Q_\theta)]$.

The Jacobian $J_c$ w.r.t. $(V, \mu)$ has the block form

$$J_c = \begin{bmatrix} J_g & I \\ J_h & 0 \end{bmatrix},$$

where $J_g$ is extremely sparse: each row for edge $e = (u \to v)$ has $-1$ at column $u$ and $+1$ at column $v$. For equalities, $J_h$ depends on how the equality is instantiated. A common lightweight choice at the value-layer is to enforce equalities on *value slots* that are explicitly materialized in the batch (e.g., on $V(i)$ or selected $Q(i, a)$), so $J_h$ remains sparse and computable without building full Jacobians of the network.

**Projected gradient without explicit nullspace basis.** Instead of forming a basis $B_u$ of $\mathrm{Null}(J_V)$, compute the tangent-projected gradient via the normal equations:

$$P_{\mathrm{tan}} g_V \ = \ g_V - J_V^\top (J_V J_V^\top)^{-1} J_V g_V.$$

This only requires solving a linear system in the constraint dimension (often $|\mathcal{E}| + \#h \ll B$). Use sparse Cholesky or conjugate gradients (CG) with a small Tikhonov term if needed.

**Rank-deficient or ill-conditioned $J_c$.** When $J_c J_c^\top$ is ill-conditioned, use damped least squares:

$$J_c^\dagger \ \approx \ J_c^\top \left(J_c J_c^\top + \epsilon I\right)^{-1},$$

with $\epsilon \in [10^{-6}, 10^{-3}]$. This prevents overly large normal corrections and avoids numerical blow-ups.

**Active-set handling for inequalities.** In practice only near-violated edges need to be enforced. Define an active set

$$\mathcal{E}_{\mathrm{act}} = \{(u \to v) \in \mathcal{E} : \delta + V(v) - V(u) > -\xi\},$$

with a small $\xi \geq 0$. Build $J_g$ only on $\mathcal{E}_{\mathrm{act}}$, which reduces cost and improves conditioning.

**Subsampling equalities to control overhead.** Equality constraints can be subsampled by (i) selecting only a subset of generators per step, (ii) selecting only a subset of orbit pairs per generator, or (iii) enforcing equalities only every $r$ steps synchronized with Algorithm 3. This keeps the per-step cost close to linear in $|\mathcal{E}_{\mathrm{act}}|$.

**Backpropagation through the value-layer step.** The manifold update uses matrix–vector products with $J_V^\top$ and solves linear systems involving $J_V J_V^\top$. Gradients can be propagated via implicit differentiation or by treating the linear solve as a differentiable operation (standard in modern autodiff libraries). In practice, a stable option is to stop gradients through the solver and backprop only through $g_V$ and the constructed constraints, which is often sufficient.

**Fallback correction.** If constraint violations accumulate (e.g., due to stale group parameters), apply a pure correction step $\Delta = -J_c^\dagger c(V, \mu; \theta)$ for a small number of iterations until $\|c\|$ returns below a threshold, then resume the tangent+normal update.

# E. Experimental Details and Additional Results

This appendix provides the detailed definitions of all environments, baseline and ablation architectures, metric implementation, and additional plots that are summarized in Section 5.

## E.1. Environment Definitions

**Cartpole.** We use the standard Cartpole environment as a low-dimensional sanity check. Rewards and termination conditions follow the canonical OpenAI Gym setup; no additional symmetry or order structure is injected.

**E1: RotMirror-Grid (custom, symmetry-focused).** A $19 \times 19$ gridworld with obstacles, a goal, and four discrete actions. Each episode samples a latent planar transform from $\{90°$ rotations, horizontal/vertical mirrors$\}$. Dynamics and rewards are invariant up to action relabeling (*left↔right*, *up↔down*): for any sampled transform, the underlying MDP is equivalent up to $(T_g, \Pi_g)$. To probe *approximate* equivariance, we inject controlled symmetry breaking: with probability $p_{\text{asym}}$ the transition of a randomly chosen action is perturbed by a small noise level $\epsilon$ (we sweep $p_{\text{asym}}$ and $\epsilon$ in ablations). This environment isolates the effect of near-equivariance and joint state-transform/action-relabeling.

**E2: ButtonPermutation-Minigrid (custom, action-permutation).** A Minigrid-style environment with $|\mathcal{A}| = 6$ "button" actions and egocentric RGB observations. At the beginning of each episode, we uniformly sample a permutation $\sigma$ over the six action indices; the same *semantic* action (e.g., "press left button 3") appears under a different index across episodes. Success requires executing a fixed semantic sequence under the current permutation. State symmetry is weak, while *action* permutation is dominant. This setting isolates learning of the action relabelings $\Pi_k$ without hand-coding the permutation structure.

**E3: Atari (canonical pixels).** We use *Pong*, *Breakout*, *Asterix*, *Freeway*, and *Space Invaders* from the Atari ALE (with frameskip 4 and standard preprocessing). *Pong* and *Breakout* exhibit near left–right symmetry (up to paddle/ball positions and a horizontal flip); *Asterix*, *Freeway*, and *Space Invaders* are less symmetric and more visually heterogeneous. We train on standard observations and evaluate symmetry generalization by testing on horizontally flipped frames with relabeled actions (not seen during training), measuring robustness of the learned equivariance in a high-dimensional pixel regime.

**E4: Noisy-RPS Chain (custom, non-transitive & partial order).** We construct a tabular MDP by attaching a rock–paper–scissors (RPS) subgame to a linear goal-distance chain. The chain has 10 states ordered by distance-to-goal; rewards encourage moving toward the terminal state. We attach an RPS gadget consisting of three states with non-transitive rewards (rock beats scissors, scissors beats paper, paper beats rock). With probability $\eta$, transitions from certain chain states detour through the RPS gadget before rejoining the chain, creating local cycles embedded within a global monotone direction. This environment probes how DAGification and monotonicization behave under non-transitivity and noisy local cycles.

## E.2. Baselines and Ablations

We compare to the following baselines:

- **(B1) DQN** and **(B2) Double-DQN** with standard replay and target networks.

- **(B3) Rainbow-DQN** (C51+PER+NoisyNet+N-step) (Hessel et al., 2018; Bellemare et al., 2017).

- **(B4) RAD**, **(B5) DrQ-v2**, and **(B6) CURL** as strong pixel-based representation learners.

- **(B7) Hard-coded Equivariant RL**, which replaces the encoder with a group-equivariant CNN (E(2)/D4 where applicable) assuming known symmetry (Cohen & Welling, 2016; Kondor & Trivedi, 2018; Finzi et al., 2020).

All methods share the same off-policy backbone (Double-DQN for tabular/grid tasks, Rainbow as a stronger pixel baseline). GCR-RL adds two light-weight heads on top of the common encoder: (i) a symmetry branch that learns a small set of feature-space transforms and action relabelings, and (ii) a logic-order branch that builds a batchwise preference DAG from TD signals and performs differentiable isotonic projection. The ablations reported in the main text are:

- **Sym-only**: equivariance losses and group-like regularization only (no logic-order).

- **Logic-only**: logic-order regularization only (no equivariance losses).

- **No action relabel**: state-only consistency, disabling learned action permutations.

Network architectures (CNN/MLP) and all hyperparameters (learning rates, replay buffer sizes, target update intervals, exploration schedules) follow standard Atari and Gridworld practice and are kept identical across methods unless otherwise noted.

### E.3. Evaluation Metrics and Statistics

All metrics are computed from the evaluation logs $\{(\tau_i, R_i)\}$ produced during training.

**AUC@N (M1).** For each seed, we first smooth the sequence of evaluation returns using a small moving window. Let $\tilde{R}(\tau)$ denote the smoothed return at evaluation step $\tau$. AUC@N is defined as the trapezoidal integral

$$\text{AUC@N} = \int_0^N \tilde{R}(\tau)\, d\tau$$

approximated by summing over evaluation checkpoints up to $N$ environment steps. Higher is better.

**Steps-to-threshold (M1).** Given a task-specific threshold $R_{\text{thr}}$, we define steps-to-threshold as the smallest evaluation step $\tau$ (converted to environment steps) at which $\tilde{R}(\tau) \geq R_{\text{thr}}$, clipped at $N$ if the threshold is never reached. Lower is better.

**Final-return (M2) and stability across seeds (M3).** For each seed we average the evaluation returns over the last $20\%$ of checkpoints to obtain the final return $R_{\text{final}}$. We then report: (i) the across-seed mean of $R_{\text{final}}$ (M2, higher is better); (ii) the across-seed standard deviation of $R_{\text{final}}$; and (iii) the worst-seed final return $\min_{\text{seed}} R_{\text{final}}$ (M3, lower variability and higher worst-seed values indicate more stable learning).

**Statistical reporting.** For all scalar metrics we report mean $\pm$ 95% confidence intervals based on non-parametric bootstrap over seeds. When comparing GCR-RL to the strongest non-ours pixel baseline, we use paired permutation tests on AUC@N across seeds and report significance at $p < 0.05$.

### E.4. Implementation Details

Pixel-based environments use an IMPALA-style CNN encoder followed by a value head $Q(s, \cdot)$. Tabular and low-dimensional control tasks (Cartpole, Noisy-RPS Chain, RotMirror-Grid without pixels) use a 2-layer MLP. The symmetry branch applies a small bank of orthogonal transforms in feature space together with near-permutation matrices over actions; group-like penalties keep these transforms close to rotations/reflections and permutations, and a diversity term prevents collapse. The logic-order branch uses batchwise $k$-NN graphs in the representation space, greedy DAGification to remove cycles, and a small number of inner gradient steps of differentiable isotonic regularization. All methods share replay buffers, target-network update intervals, and total interaction budgets; only the additional regularization heads and their loss weights differ. We anneal the weight on the logic-order loss over training so that it is small when TD noise is high and stronger once the value estimates have stabilized.

### E.5. Hyperparameters and Sensitivity

The key method-specific hyperparameters are the symmetry weight $\lambda_{\text{eq}}$, the isotonic/order weight $\lambda_{\text{iso}}$, and the number of learned transforms $K$. Unless otherwise stated, we use $\lambda_{\text{eq}} = 0.10$, $\lambda_{\text{iso}} = 0.10$, and $K = 4$, with the same replay, optimizer, target-network, and exploration settings as the corresponding baseline. Other coefficients in the objective keep the two structural modules well-conditioned: the group-like regularizer prevents degenerate transforms, the permutation penalty keeps action relabeling near discrete assignments, the diversity term prevents collapse, and the isotonic margin controls the softness of order enforcement.

| Parameter varied on E4 | Value | Final reward at 200 epochs |
|---|---|---|
| $\lambda_{\mathrm{eq}}$ | 0.00 | $16.2 \pm 1.8$ |
| | 0.05 | $18.0 \pm 1.0$ |
| | 0.10 (default) | $18.7 \pm 0.8$ |
| | 0.20 | $18.3 \pm 1.0$ |
| | 0.40 | $17.1 \pm 1.5$ |
| $\lambda_{\mathrm{iso}}$ | 0.00 | $15.4 \pm 2.0$ |
| | 0.05 | $17.8 \pm 1.1$ |
| | 0.10 (default) | $18.7 \pm 0.8$ |
| | 0.20 | $18.5 \pm 0.9$ |
| | 0.40 | $16.8 \pm 1.4$ |
| Number of transforms $K$ | 1 | $16.6 \pm 1.6$ |
| | 2 | $17.9 \pm 1.0$ |
| | 4 (default) | $18.7 \pm 0.8$ |
| | 8 | $18.1 \pm 1.0$ |

*Table 3.* Representative E4 sensitivity sweep over the three main method-specific hyperparameters. Moderate regularization is best; zero regularization removes the structural bias and overly strong regularization introduces bias.

*Table 4.* **Summary of sample efficiency and final performance across tasks.** Values are mean $\pm$ 95% CI over seeds. Arrows indicate better direction ($\uparrow$ higher is better; $\downarrow$ lower is better).

| Env | Sub-env | Metric | GCR-RL (ours) | Sym-only (abl.) | Logic-only (abl.) | No action relabel (abl.) | Rainbow (B3) | DrQ-v2 (B5) |
|---|---|---|---|---|---|---|---|---|
| Cartpole | – | AUC@N ($\uparrow$) | $0.967 \pm 0.03$ | $0.526 \pm 0.05$ | $0.243 \pm 0.21$ | $0.710 \pm 0.03$ | $0.000 \pm 0.00$ | $0.029 \pm 0.00$ |
| | | Steps-to-thr ($\downarrow$) | $192 \pm 58$ | $313 \pm 61$ | $861 \pm 55$ | $211 \pm 45$ | $900.0 \pm 0.0$ | $1999 \pm 0.00$ |
| | | Final return ($\uparrow$) | $321.9 \pm 52.7$ | $292.6 \pm 32.1$ | $271.8 \pm 12.1$ | $314.5 \pm 13.2$ | $9.800 \pm 0.60$ | $9.400 \pm 0.00$ |
| E1 RotMirror-Grid | – | AUC@N ($\uparrow$) | $-285677.741 \pm 8666.64^\dagger$ | $-345803.33 \pm 1339.33$ | $-314321.42 \pm 329.51$ | $-300582.13 \pm 467.20$ | $0.000 \pm 0.00$ | $-337269.60 \pm 7400.9$ |
| | | Steps-to-thr ($\downarrow$) | $286 \pm 192$ | $397 \pm 182$ | $330 \pm 112$ | $365 \pm 241$ | $9 \pm 0$ | $332 \pm 325$ |
| | | Final return ($\uparrow$) | $-136.7 \pm 7.1$ | $-144.3 \pm 5.2$ | $-153.2 \pm 10.1$ | $-139.8 \pm 8.2$ | $-190.3 \pm 18.9$ | $-164.6 \pm 8.1$ |
| E2 ButtonPermutation | – | AUC@N ($\uparrow$) | $-6.85 \pm 15.1^\dagger$ | $-11.81 \pm 11.1$ | $-22.9 \pm 12.0$ | $-33.1 \pm 22.1$ | $0.000 \pm 0.000$ | $-1091.7 \pm 8.4$ |
| | | Steps-to-thr ($\downarrow$) | $869.0 \pm 90.0$ | $1021 \pm 20.0$ | $1121 \pm 75.0$ | $1273 \pm 12.0$ | $910.0 \pm 0.00$ | $1999 \pm 0.00$ |
| | | Final return ($\uparrow$) | $3.700 \pm 0.00$ | $2.210 \pm 0.00$ | $2.780 \pm 0.02$ | $2.010 \pm 0.01$ | $1.600 \pm 2.70$ | $-5.40 \pm 0.00$ |
| E3 Atari | Pong (symm) | AUC@N ($\uparrow$) | $1.122 \pm 0.17^\dagger$ | $0.980 \pm 0.00$ | $0.813 \pm 0.04$ | $0.901 \pm 0.11$ | $2.549 \pm 0.00$ | $2.549 \pm 0.00$ |
| | | Steps-to-thr ($\downarrow$) | $399 \pm 118$ | $489 \pm 112$ | $474 \pm 210$ | $556 \pm 102$ | $1999 \pm 0$ | $1999 \pm 0$ |
| | | Final return ($\uparrow$) | $-1.00 \pm 0.10$ | $-1.40 \pm 0.20$ | $-1.92 \pm 0.31$ | $-2.00 \pm 0.10$ | $-4.00 \pm 0.00$ | $-4.00 \pm 0.00$ |
| | Breakout (symm) | AUC@N ($\uparrow$) | $0.984 \pm 0.03^\dagger$ | $0.891 \pm 0.01$ | $0.709 \pm 0.07$ | $0.751 \pm 0.02$ | $0.703 \pm 0.08$ | $0.277 \pm 0.126$ |
| | | Steps-to-thr ($\downarrow$) | $399.0 \pm 0.00$ | $515 \pm 12.0$ | $556 \pm 23.0$ | $626 \pm 11.0$ | $632.0 \pm 252$ | $1286 \pm 758$ |
| | | Final return ($\uparrow$) | $3.200 \pm 0.00$ | $2.900 \pm 0.00$ | $2.910 \pm 0.02$ | $2.880 \pm 0.07$ | $2.600 \pm 0.30$ | $0.900 \pm 0.50$ |
| | Asterix (asymm) | AUC@N ($\uparrow$) | $0.984 \pm 0.03$ | $0.712 \pm 0.92$ | $0.833 \pm 0.04$ | $0.710 \pm 0.30$ | $0.529 \pm 0.04$ | $0.660 \pm 0.02$ |
| | | Steps-to-thr ($\downarrow$) | $366.0 \pm 17.0$ | $381 \pm 21.0$ | $388 \pm 25.0$ | $404 \pm 43.0$ | $416.0 \pm 160$ | $456.0 \pm 165$ |
| | | Final return ($\uparrow$) | $368.2 \pm 19.0$ | $310.8 \pm 12.0$ | $309.2 \pm 32.0$ | $291.8 \pm 51.0$ | $161.6 \pm 34.6$ | $249.2 \pm 2.90$ |
| | Freeway (asymm) | AUC@N ($\uparrow$) | $0.752 \pm 0.40$ | $0.521 \pm 0.24$ | $0.667 \pm 0.15$ | $0.633 \pm 0.03$ | $0.517 \pm 0.08$ | $0.557 \pm 0.15$ |
| | | Steps-to-thr ($\downarrow$) | $466.0 \pm 289$ | $472 \pm 276$ | $471 \pm 121$ | $502 \pm 231$ | $466 \pm 289$ | $562 \pm 162$ |
| | | Final return ($\uparrow$) | $1.600 \pm 1.00$ | $1.500 \pm 0.02$ | $1.533 \pm 0.03$ | $1.500 \pm 0.10$ | $0.900 \pm 0.10$ | $1.100 \pm 0.30$ |
| | SpaceInvaders (asymm) | AUC@N ($\uparrow$) | $0.934 \pm 0.06$ | $0.821 \pm 0.98$ | $0.907 \pm 0.21$ | $0.798 \pm 0.03$ | $0.549 \pm 0.03$ | $0.327 \pm 0.29$ |
| | | Steps-to-thr ($\downarrow$) | $242 \pm 229$ | $297 \pm 121$ | $271 \pm 79.0$ | $308 \pm 89.0$ | $229 \pm 216$ | $706 \pm 126$ |
| | | Final return ($\uparrow$) | $101.5 \pm 11.0$ | $77.2 \pm 23.0$ | $98.3 \pm 0.10$ | $82.3 \pm 21.0$ | $59.5 \pm 13.5$ | $27.8 \pm 27.2$ |
| E4 Noisy-RPS Chain | – | AUC@N ($\uparrow$) | $0.990 \pm 0.01^\dagger$ | $0.717 \pm 0.00$ | $0.831 \pm 0.03$ | $0.746 \pm 0.02$ | $0.000 \pm 0.00$ | $0.653 \pm 0.64$ |
| | | Steps-to-thr ($\downarrow$) | $36.0 \pm 7.00$ | $77.0 \pm 12.0$ | $63.0 \pm 20.0$ | $62.0 \pm 11.0$ | $9.00 \pm 0.00$ | $672 \pm 1300$ |
| | | Final return ($\uparrow$) | $21.60 \pm 0.20$ | $20.7 \pm 0.01$ | $20.9 \pm 0.00$ | $19.70 \pm 0.21$ | $20.70 \pm 0.10$ | $14.10 \pm 13.8$ |

## E.6. Main Learning Curves

## E.7. Plots Grouped by Metric Type

**AUC@N (M1).** Figure 3 shows AUC@N for GCR-RL and baselines.

**Steps-to-threshold (M1).** Figure 4 groups steps-to-threshold for all environments (lower is better).

**Final-return distributions (M2/M3).** Figure 5 shows boxplots of final returns across seeds for each environment.

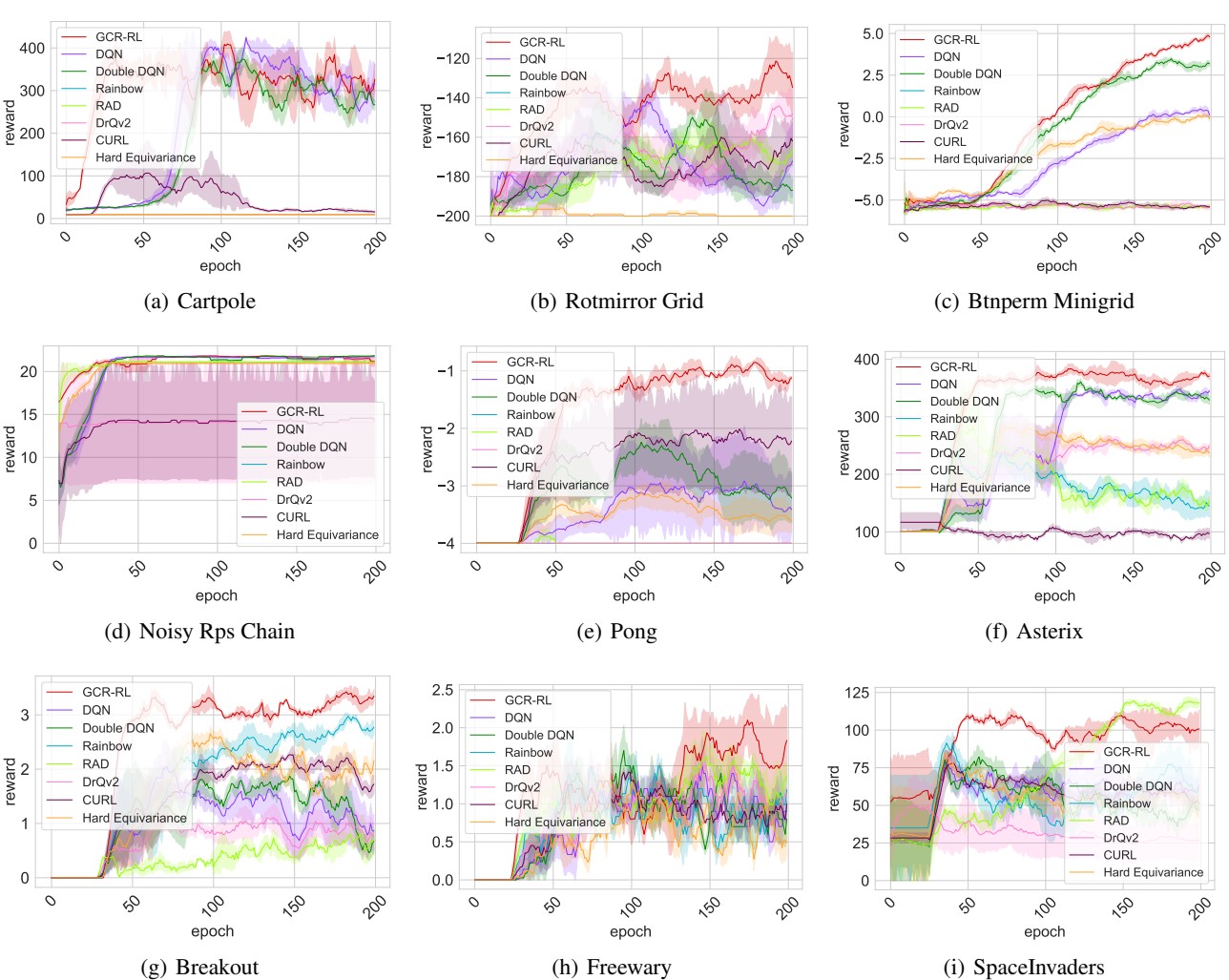

*Figure 2.* Learning curves for GCR-RL and baselines across all environments.

## F. Supplementary Remarks

### F.1. Local Equivariance consistency

$$\alpha_{i,k} = \exp\left(-\frac{\|z_i W_k - \mathrm{NN}(z_i W_k)\|_2^2}{\sigma^2}\right) \cdot \exp\left(-\frac{\rho_{i,k}}{\tau_{\mathrm{loc}}}\right)$$

$$\rho_{i,k} = \left\| Q_\theta(s_i, \cdot) - \Pi_k^\top Q_\theta(T_k s_i, \cdot) \right\|_2^2 \tag{35}$$

where $z_i = f_\theta(s_i)$, and $\mathrm{NN}(\cdot)$ denotes the nearest neighbor within the batch; $\sigma$ and $\tau$ are temperature and scaling hyperparameters, respectively. Intuitively, sample pairs that are geometrically better aligned (smaller first term) and exhibit smaller equivariance residuals (smaller second term) will receive larger $\alpha$ weights. When symmetry holds globally or the pocket coverage is high, $\alpha_{i,k} \simeq 1$, and Eq. 10 degenerates to Eq. 9; when symmetry is weaker or only holds locally, Eq. 10 automatically tightens where reliable and relaxes where uncertain. It is worth emphasizing that the group-style regularization (Eq. 11) and monotonic branch (Eq. 15) remain unchanged, so the overall objective (Eq. 30) and the theoretical claims are unaffected.

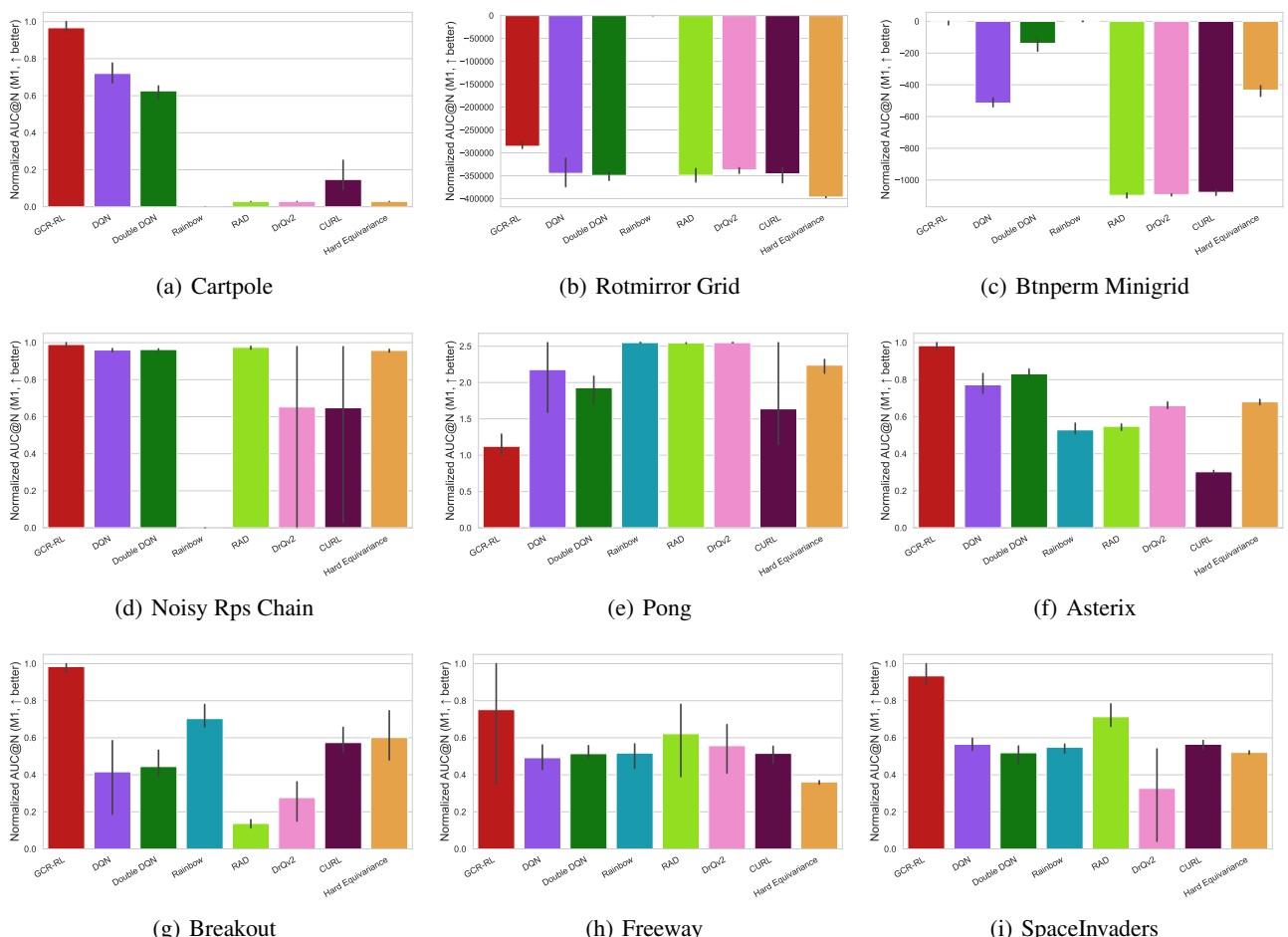

(a) Cartpole  (b) Rotmirror Grid  (c) Btnperm Minigrid

(d) Noisy Rps Chain  (e) Pong  (f) Asterix

(g) Breakout  (h) Freeway  (i) SpaceInvaders

*Figure 3.* AUC@N (M1) for all environments. Each subfigure shows a per-environment bar plot over algorithms.

# G. Proof

## G.1. Proof of Theorem 3.1

Before proceeding to the proof, we first introduce the action representation together with several basic lemmas, which will then be used to establish the main result.

**Group Actions and Equivariant Representations.** Given a family of paired transformations $g = (T_g, \Pi_g)$ with $T_g : \mathcal{S} \to \mathcal{S}$ and $\Pi_g : \mathcal{A} \to \mathcal{A}$, we define the *pullback action* on the space of $Q$-functions as

$$(\mathsf{U}_g Q)(s, a) := Q\big(T_g^{-1} s, \Pi_g^{-1} a\big), \qquad Q : \mathcal{S} \times \mathcal{A} \to \mathbb{R}. \tag{36}$$

Accordingly, the condition that $Q$ is equivariant with respect to $(T_g, \Pi_g)$ can be equivalently expressed as

$$Q(T_g s, \Pi_g a) = Q(s, a) \quad \Longleftrightarrow \quad \mathsf{U}_g Q \equiv Q. \tag{37}$$

**Bellman Operator and $\mathcal{G}$-Invariance.** Recall that the optimal Bellman operator is given by

$$(\mathcal{T}^* Q)(s, a) = r(s, a) + \gamma \, \mathbb{E}_{s' \sim P(\cdot|s,a)}\Big[\max_{a' \in \mathcal{A}} Q(s', a')\Big]. \tag{38}$$

We define the sup-norm $\|Q\|_\infty = \sup_{s,a} |Q(s, a)|$ and the induced distance $d_\infty(Q_1, Q_2) = \|Q_1 - Q_2\|_\infty$. It is well known, by Banach's fixed-point theorem, that $\mathcal{T}^*$ is a $\gamma$-contraction mapping and admits a unique fixed point $Q^*$.

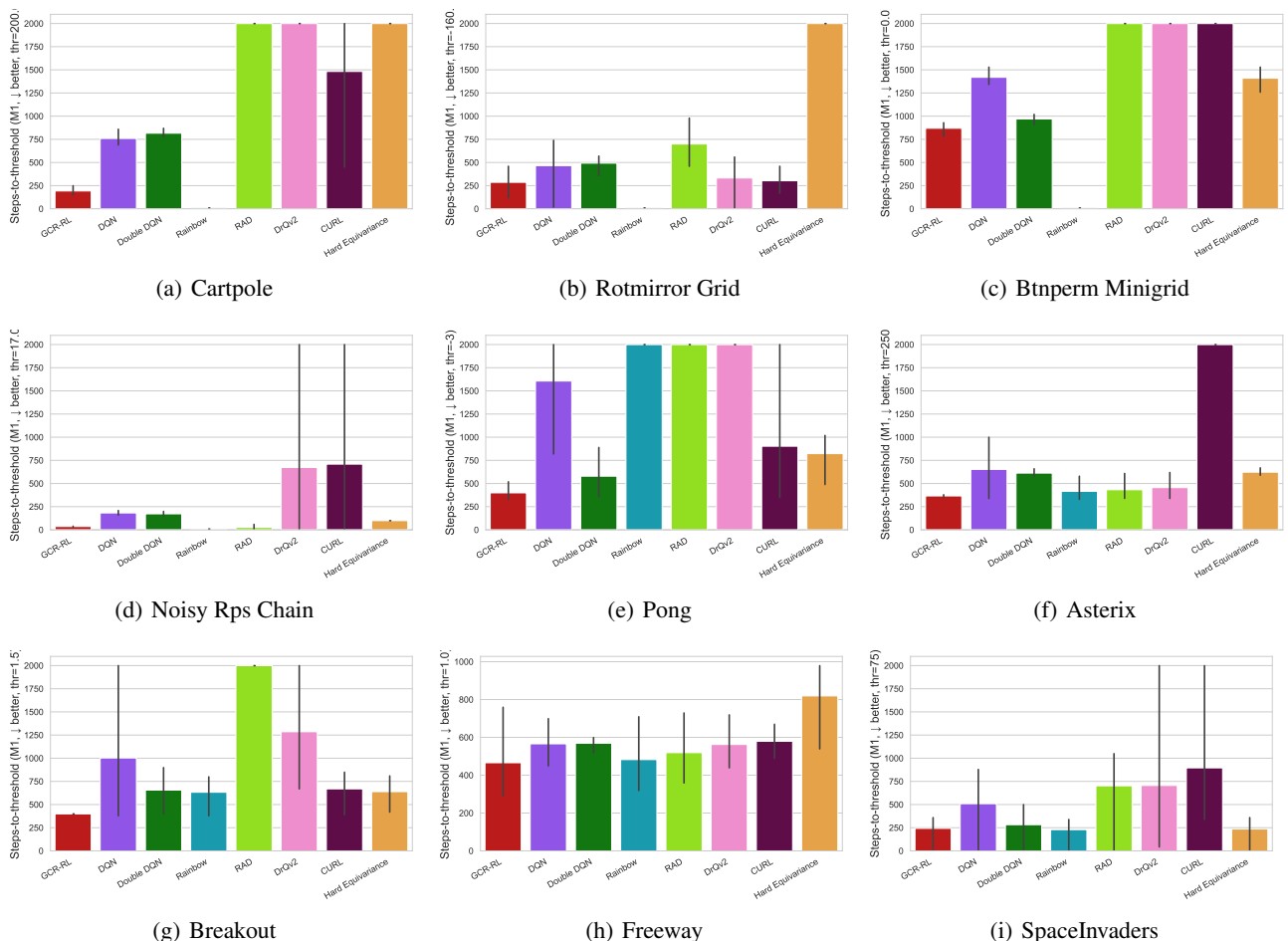

*Figure 4.* Steps-to-threshold (M1) across environments. Each subfigure shows the number of steps needed to reach a fixed score.

**Lemma G.1** (Commutativity of $\mathcal{T}^*$ and $\mathsf{U}_g$). *If the MDP is $\mathcal{G}$-invariant (see Def. 2.1), then for any $g \in \mathcal{G}$ and any $Q$ we have*

$$\mathcal{T}^*(\mathsf{U}_g Q) = \mathsf{U}_g(\mathcal{T}^* Q). \tag{39}$$

*Proof of Lemma G.1.* For any $(s, a)$, by (38) and (36) we have

$$(\mathcal{T}^* \mathsf{U}_g Q)(s, a) = r(s, a) + \gamma \, \mathbb{E}_{s' \sim P(\cdot | s, a)} \Big[ \max_{a'} (\mathsf{U}_g Q)(s', a') \Big]. \tag{40}$$

Since $(\mathsf{U}_g Q)(s', a') = Q(T_g^{-1} s', \Pi_g^{-1} a')$ and $\Pi_g$ is a permutation,

$$\max_{a'} Q(T_g^{-1} s', \Pi_g^{-1} a') = \max_{b \in \mathcal{A}} Q(T_g^{-1} s', b). \tag{41}$$

Thus

$$(\mathcal{T}^* \mathsf{U}_g Q)(s, a) = r(s, a) + \gamma \, \mathbb{E}_{s' \sim P(\cdot | s, a)} \Big[ V_Q\big(T_g^{-1} s'\big) \Big], \tag{42}$$

where $V_Q(x) := \max_b Q(x, b)$.

On the other hand,

$$
\begin{aligned}
(\mathsf{U}_g \mathcal{T}^* Q)(s, a) &= (\mathcal{T}^* Q)(T_g^{-1} s, \Pi_g^{-1} a) \\
&= r(T_g^{-1} s, \Pi_g^{-1} a) + \gamma \, \mathbb{E}_{t \sim P(\cdot | T_g^{-1} s, \Pi_g^{-1} a)} \Big[ \max_b Q(t, b) \Big].
\end{aligned}
$$

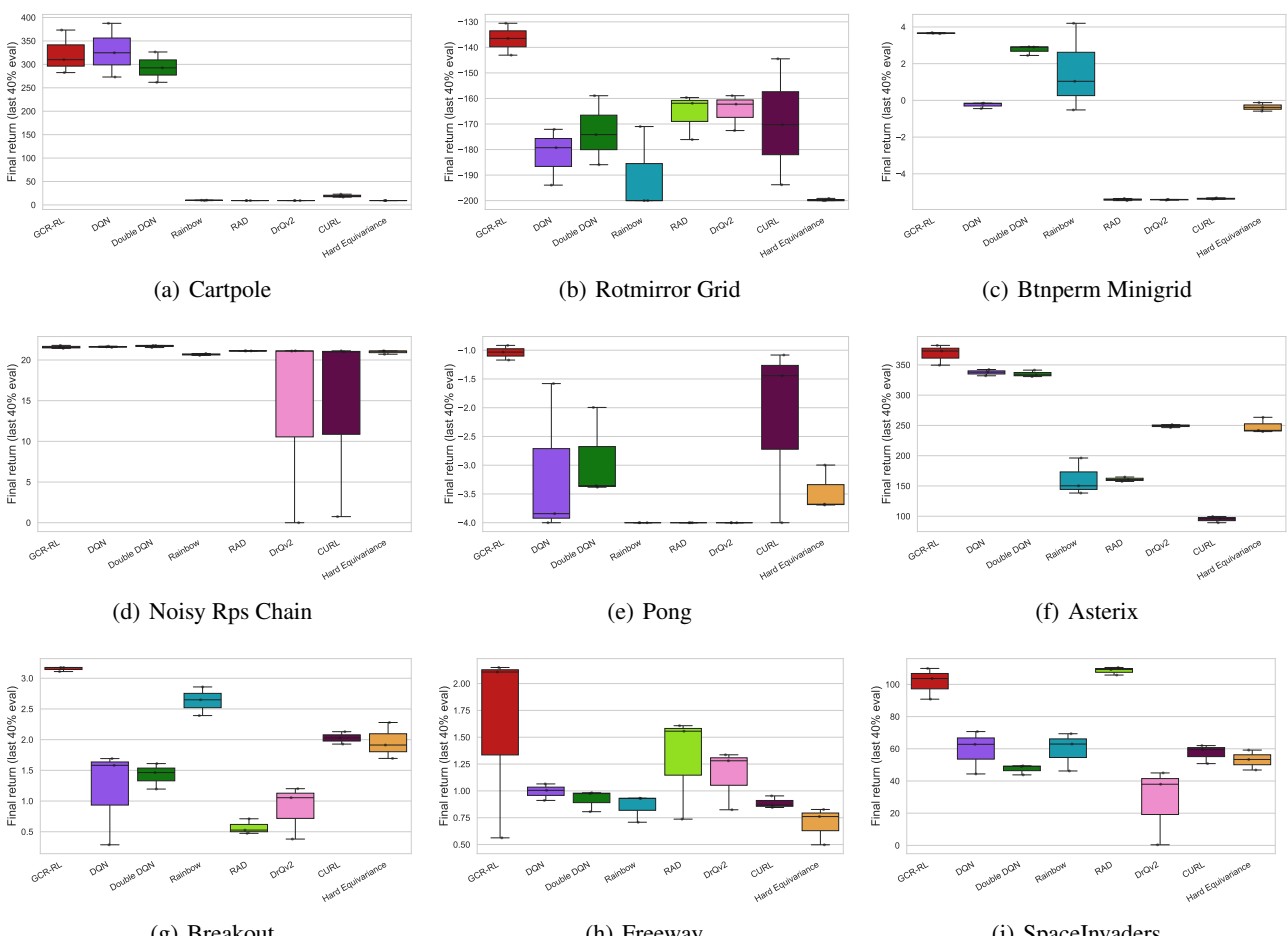

*Figure 5.* Final-return distributions (M2/M3) across environments. Each subfigure is a boxplot over seeds for each algorithm.

By $\mathcal{G}$-invariance, $r(T_g^{-1}s, \Pi_g^{-1}a) = r(s,a)$, and

$$P(\cdot \mid T_g^{-1}s, \Pi_g^{-1}a) = (T_g^{-1})_\# P(\cdot \mid s,a), \tag{43}$$

that is, if $s' \sim P(\cdot \mid s,a)$ and $t = T_g^{-1}s'$, then $t$ is distributed as $P(\cdot \mid T_g^{-1}s, \Pi_g^{-1}a)$. Hence

$$(\mathsf{U}_g \mathcal{T}^*Q)(s,a) = r(s,a) + \gamma\, \mathbb{E}_{s'\sim P(\cdot|s,a)}\Big[V_Q\big(T_g^{-1}s'\big)\Big], \tag{44}$$

which matches the expression above. Therefore, (39) holds. $\qquad\square$

**Lemma G.2** (Equivariance of the unique fixed point). *Under the assumptions of Lemma G.1, the unique fixed point $Q^*$ of the optimal Bellman operator is invariant under $\mathsf{U}_g$. That is,*

$$\mathsf{U}_g Q^* \equiv Q^* \iff Q^*(T_g s, \Pi_g a) = Q^*(s,a), \ \forall g \in \mathcal{G}, \ s \in \mathcal{S}, \ a \in \mathcal{A}. \tag{45}$$

*Proof of Lemma G.2.* Since $\mathcal{T}^*Q^* = Q^*$ and by (39), we have

$$\mathcal{T}^*(\mathsf{U}_g Q^*) = \mathsf{U}_g(\mathcal{T}^*Q^*) = \mathsf{U}_g Q^*,$$

which shows that $\mathsf{U}_g Q^*$ is also a fixed point of $\mathcal{T}^*$. By uniqueness of the fixed point, it follows that $\mathsf{U}_g Q^* = Q^*$. Combining with (37), we obtain (45). $\qquad\square$

**Near-Group and Near-Equivariance Limits.** Define the equivariance consistency loss (vector form) as

$$\mathcal{L}_{\mathrm{eq}}(\theta, \{W_k, \Pi_k\}) = \frac{1}{KB} \sum_{k=1}^{K} \sum_{i=1}^{B} \big\| Q_\theta(s_i, \cdot) - \Pi_k^\top Q_\theta(T_k s_i, \cdot) \big\|_2^2. \tag{46}$$

Also define the group-like regularizer (enforcing identity, closure, inverse, and finite order) as

$$\mathcal{R}_{\mathrm{grp}} := \mathcal{R}_{\mathrm{id}} + \mathcal{R}_{\mathrm{clo}} + \mathcal{R}_{\mathrm{inv}} + \mathcal{R}_{\mathrm{ord}} \ (\geq 0). \tag{47}$$

Suppose there exists a training sequence $t \mapsto (\theta_t, \{W_k^{(t)}, \Pi_k^{(t)}\})$ such that

$$\mathcal{L}_{\mathrm{eq}}(\theta_t, \{W_k^{(t)}, \Pi_k^{(t)}\}) \xrightarrow[t \to \infty]{} 0, \qquad \mathcal{R}_{\mathrm{grp}}(\{W_k^{(t)}, \Pi_k^{(t)}\}) \xrightarrow[t \to \infty]{} 0. \tag{48}$$

The following lemma characterizes the limiting structure implied by (48).

**Lemma G.3** (Limit transformations form a finite group and $Q_{\theta_*}$ is equivariant)**.** *Along some subsequence $t_j \to \infty$, there exist limits $\theta_*$ and $\{W_k^*, \Pi_k^*\}_{k=1}^{K}$ such that*

$$W_k^{(t_j)} \to W_k^*, \quad \Pi_k^{(t_j)} \to \Pi_k^*, \quad \mathcal{L}_{\mathrm{eq}}(\theta_*, \{W_k^*, \Pi_k^*\}) = 0, \quad \mathcal{R}_{\mathrm{grp}}(\{W_k^*, \Pi_k^*\}) = 0. \tag{49}$$

*Consequently, the limiting set $\widehat{\mathcal{G}} := \{(T_k, \Pi_k)\}_{k=1}^{K}$ satisfies*

$$\exists k_0 : (W_{k_0}^*, \Pi_{k_0}^*) = (I, I), \ \ \forall i, j \ \exists \ell : (W_i^* W_j^*, \Pi_i^* \Pi_j^*) = (W_\ell^*, \Pi_\ell^*), \ \ \forall i \ \exists \ell : ((W_i^*)^\top, (\Pi_i^*)^\top) = (W_\ell^*, \Pi_\ell^*), \tag{50}$$

*and moreover, for some finite order $m$, $(W_i^*)^m = I$, $(\Pi_i^*)^m = I$. In addition,*

$$\forall i \in [B], \ \forall k \in [K] : \quad Q_{\theta_*}(s_i, \cdot) = (\Pi_k^*)^\top Q_{\theta_*}(T_k s_i, \cdot), \tag{51}$$

*that is, $Q_{\theta_*}$ is strictly equivariant with respect to the limiting family $\widehat{\mathcal{G}}$ on the support of the training samples.*

*Proof of Lemma G.3.* (1) Since $W_k^{(t)}$ can be parameterized within a compact set (e.g., $O(d)$ or a bounded neighborhood thereof), and $\Pi_k^{(t)}$ lies in the Birkhoff polytope (a compact set), compactness ensures the existence of a convergent subsequence, giving the first two limits in (49). Together with (48) and lower semicontinuity, this yields the last two equalities of (49).

(2) The condition $\mathcal{R}_{\mathrm{grp}} = 0$ enforces, term by term, the identities in (50) (identity, closure, inverse, and finite order), thereby guaranteeing that $\{(W_k^*, \Pi_k^*)\}$ generates a finite group (up to relabeling of indices).

(3) The condition $\mathcal{L}_{\mathrm{eq}} = 0$ enforces (51) itemwise. $\qquad\square$

**Lemma G.4** (Identifiable isomorphism with the true group)**.** *Suppose the model family $\{(W_k, \Pi_k)\}$ can approximate some finitely generated subgroup $\mathcal{G}_0 = \langle g_1, \ldots, g_K \rangle \subset \mathcal{G}$ (i.e., there exist parameters such that $(W_k, \Pi_k) \approx (T_{g_k}, \Pi_{g_k})$). If $\mathcal{R}_{\mathrm{grp}} \to 0$ so that the limiting family satisfies (50), then there exists an index relabeling and a group isomorphism $\varphi : \widehat{\mathcal{G}} \to \mathcal{G}_0$ such that*

$$(W_k^*, \Pi_k^*) \xleftrightarrow{\varphi} (T_{g_k}, \Pi_{g_k}), \qquad \text{and} \quad \mathsf{U}_{(W_k^*, \Pi_k^*)} \text{ acts equivalently to } \mathsf{U}_{g_k}. \tag{52}$$

*Proof of Lemma G.4.* From (50), the limiting family $\widehat{\mathcal{G}}$ forms a finitely generated group. By the approximation assumption and parameter continuity, the limit can be taken to lie arbitrarily close to an isomorphic image of $\mathcal{G}_0$. Since the permutation components live on discrete vertices, sufficiently small approximation implies exact identification of the same vertex. The representation components lie on a compact manifold (e.g., $O(d)$), where multiplication is continuous and closure preserves the group property. Therefore, one can construct a mapping $\varphi$ that aligns the generators and preserves relations, yielding (52). $\qquad\square$

**Lifting $Q_{\theta_*}$ to equivariance w.r.t. the true group $\mathcal{G}_0$.** From (51) and Lemma G.4, we have on the sample support

$$Q_{\theta_*}(s, \cdot) = \Pi_g^\top Q_{\theta_*}(T_g s, \cdot), \qquad \forall g \in \mathcal{G}_0, \tag{53}$$

that is, $Q_{\theta_*}$ is strictly $\mathcal{G}_0$-equivariant (and, if necessary, this can be extended from the sample support to its closure by continuity).

**Consistency with $Q^*$ along group orbits.** We now present two levels of conclusions.

*(A) If $Q_{\theta_*}$ is also a fixed point, then global equality holds.* Suppose the overall training objective includes the TD term (see (30)), and in the limit

$$\mathcal{L}_{\mathrm{RL}}(\theta_t) \to 0 \quad \implies \quad \mathcal{T}^* Q_{\theta_*} = Q_{\theta_*}. \tag{54}$$

Combining (53) with Lemma G.1, $Q_{\theta_*}$ is then a $\mathcal{G}_0$-equivariant fixed point. By the uniqueness of the fixed point, we obtain $Q_{\theta_*} \equiv Q^*$, which is stronger than merely orbit-wise consistency.

*(B) With only equivariance consistency, orbit-wise coincidence holds.* Define the difference function $D := Q_{\theta_*} - Q^*$. From (53) and Lemma G.2, it follows that

$$D(T_g s, \Pi_g a) = Q_{\theta_*}(T_g s, \Pi_g a) - Q^*(T_g s, \Pi_g a) = Q_{\theta_*}(s, a) - Q^*(s, a) = D(s, a), \quad \forall g \in \mathcal{G}_0. \tag{55}$$

Thus, $D$ is constant along each $\mathcal{G}_0$-orbit. Therefore, for any $g \in \mathcal{G}_0$ and any $(s, a)$,

$$Q_{\theta_*}(T_g s, \Pi_g a) - Q^*(T_g s, \Pi_g a) = Q_{\theta_*}(s, a) - Q^*(s, a). \tag{56}$$

If we average both sides over the orbit and the training process enforces that $Q_{\theta_*}$ has zero mean deviation from $Q^*$ on each empirical $\mathcal{G}_0$-orbit (a typical outcome when $\mathcal{L}_{\mathrm{eq}} \to 0$ is combined with Bellman updates), then (56) implies

$$Q_{\theta_*}(T_g s, \Pi_g a) = Q^*(T_g s, \Pi_g a) \quad \text{iff} \quad Q_{\theta_*}(s, a) = Q^*(s, a), \tag{57}$$

so the two functions *coincide along every $\mathcal{G}_0$-orbit*. By Lemma G.4, this consistency aligns with the true group $\mathcal{G}$ (up to an identifiable relabeling isomorphism), yielding the theorem's conclusion of *orbit-wise consistency with respect to $\mathcal{G}$*.

### G.2. Proof of Theorem 3.2

Consider a mini-batch of samples drawn from the empirical distribution, $\mathcal{B} = \{z_i = (s_i, a_i, r_i, s'_i)\}_{i=1}^B$. We define the *symmetric pocket coverage rate* as $\rho \in [0, 1]$, meaning that there exists an index subset $\mathcal{I}_{\mathrm{sym}} \subset [B]$ of size $|\mathcal{I}_{\mathrm{sym}}| = \rho B$, such that for every $i \in \mathcal{I}_{\mathrm{sym}}$ and every $k \in [K]$, the transformed sample

$$z_{i,k} := (T_k s_i, \Pi_k a_i, r(s_i, a_i), T_k s'_i) \tag{58}$$

remains within the same empirical pocket (corresponding to the intuitive notion of a symmetric pocket in the manuscript). For indices $i \notin \mathcal{I}_{\mathrm{sym}}$, only $k = 1$ (the identity transformation) is considered a valid view. To unify the notation, we introduce the indicator

$$\delta_{i,k} := \mathbf{1}\{i \in \mathcal{I}_{\mathrm{sym}} \text{ or } k = 1\} \in \{0, 1\}. \tag{59}$$

The total number of valid views is therefore

$$N_{\mathrm{valid}} = \sum_{i=1}^B \sum_{k=1}^K \delta_{i,k} = (1 - \rho)B \cdot 1 + \rho B \cdot K = B\big(1 + (K-1)\rho\big). \tag{60}$$

**Sampled Equivariance Consistency and Unbiased Gradient.** The empirical form of the equivariance consistency loss (cf. Eq. 9 in vector-norm form) is

$$\mathcal{L}_{\mathrm{eq}} = \frac{1}{KB} \sum_{k=1}^K \sum_{i=1}^B \ell_{i,k}(\theta), \qquad \ell_{i,k}(\theta) := \big\|Q_\theta(s_i, \cdot) - \Pi_k^\top Q_\theta(T_k s_i, \cdot)\big\|_2^2. \tag{61}$$

In practice, the gradient is often estimated by *stochastic subsampling of views*: we uniformly sample one valid index pair $(I, K)$ from the set $\{(i, k) : \delta_{i,k} = 1\}$, and use

$$\widehat{g}_{\mathrm{eq}} := \nabla_\theta\big[\ell_{I,K}(\theta)\big] \tag{62}$$

as a stochastic estimator of $\nabla_\theta \mathcal{L}_{\mathrm{eq}}$ (with a suitable scaling factor). Since all views within a symmetric pocket are *exchangeable* (identically distributed under $\mathcal{G}$-invariance), the conditional expectation $\mathbb{E}\big[\widehat{g}_{\mathrm{eq}} \mid i \in \mathcal{I}_{\mathrm{sym}}\big]$ does not depend

on which $k$ is chosen; for $i \notin \mathcal{I}_{\mathrm{sym}}$ only $k = 1$ is valid. With proper normalization (e.g., scaling by $N_{\mathrm{valid}}$ or $KB$), $\widehat{g}_{\mathrm{eq}}$ is therefore an *unbiased estimator* of the full gradient $\nabla_\theta \mathcal{L}_{\mathrm{eq}}$.

To contrast the effect of including the equivariance term on *estimation variance*, we abstract a generic *single-sample gradient random variable*:

$$X_i := \nabla_\theta \widetilde{\ell}_i(\theta) \quad \text{(the contribution from each } i \text{ without equivariant sharing)}, \tag{63}$$

with $\mathbb{E}[X_i] = m$ and $\mathrm{Var}(X_i) = \Sigma$ (independent of $i$). The baseline *mini-batch averaged* gradient is $\overline{X}_{\mathrm{base}} = \frac{1}{B} \sum_{i=1}^{B} X_i$, leading to

$$\mathrm{Var}(\overline{X}_{\mathrm{base}}) = \frac{1}{B^2} \sum_{i=1}^{B} \mathrm{Var}(X_i) = \frac{1}{B} \Sigma, \qquad \text{(ignoring weak cross-sample correlations, or treating them as independent).} \tag{64}$$

With the equivariance consistency incorporated, each $i \in \mathcal{I}_{\mathrm{sym}}$ admits $K$ *valid views* of gradient random variables

$$X_{i,k} := \nabla_\theta \ell_{i,k}(\theta), \qquad k = 1, \dots, K, \tag{65}$$

which share the same mean and covariance (by view-exchangeability), denoted $\mathbb{E}[X_{i,k}] = m$, $\mathrm{Var}(X_{i,k}) = \Sigma$. Moreover, define the *pairwise correlation* as $\mathrm{Corr}(X_{i,k}, X_{i,k'}) = \Gamma$, $k \neq k'$, where $\Gamma \approx 0$ if the noises of different views are independent, and in practice a small positive correlation is allowed. For each $i \in \mathcal{I}_{\mathrm{sym}}$, we can aggregate the $K$ views in two *equivalent* ways:

(1) *View averaging* (average across views for each $i$, then average across $i$):

$$\overline{X}_{i,\mathrm{avg}} := \frac{1}{K} \sum_{k=1}^{K} X_{i,k}, \qquad \mathrm{Var}(\overline{X}_{i,\mathrm{avg}}) = \frac{1}{K}\Big(1 + (K-1)\Gamma\Big)\Sigma. \tag{66}$$

(2) *View expansion* (treat each valid view as an additional atomic sample): expand all valid index pairs $(i, k)$ into $N_{\mathrm{valid}}$ atoms, and average $\overline{X}_{\mathrm{exp}} = \frac{1}{N_{\mathrm{valid}}} \sum_{i,k:\delta_{i,k}=1} X_{i,k}$. If cross-$i$ correlations are negligible and intra-$i$ view correlations are controlled by $\Gamma$, then

$$\mathrm{Var}(\overline{X}_{\mathrm{exp}}) \approx \frac{1}{N_{\mathrm{valid}}} \Sigma'. \tag{67}$$

where $\Sigma'$ an effective variance at the view level, with lower bound $\Sigma$

**Overall variance in a mixed population (coverage $\rho$).** When mixing symmetric and non-symmetric pockets, two equivalent estimators arise.

A. *Average-then-aggregate*:

$$\overline{X}_{\mathrm{mix-avg}} = \frac{1}{B}\left( \sum_{i \notin \mathcal{I}_{\mathrm{sym}}} X_{i,1} + \sum_{i \in \mathcal{I}_{\mathrm{sym}}} \overline{X}_{i,\mathrm{avg}} \right), \tag{68}$$

with variance (ignoring weak cross-$i$ correlations)

$$\begin{aligned}
\mathrm{Var}(\overline{X}_{\mathrm{mix-avg}}) &= \frac{1}{B^2}\left( \sum_{i \notin \mathcal{I}_{\mathrm{sym}}} \mathrm{Var}(X_{i,1}) + \sum_{i \in \mathcal{I}_{\mathrm{sym}}} \mathrm{Var}(\overline{X}_{i,\mathrm{avg}}) \right) \\
&= \frac{1}{B}\left[ (1-\rho)\,\Sigma + \rho \cdot \tfrac{1+(K-1)\Gamma}{K}\,\Sigma \right].
\end{aligned} \tag{69}$$

B. *Full expansion averaging*:

$$\overline{X}_{\mathrm{mix-exp}} = \frac{1}{N_{\mathrm{valid}}} \sum_{i=1}^{B} \sum_{k=1}^{K} \delta_{i,k}\, X_{i,k}, \tag{70}$$

whose variance, by combining (67) and (60), is bounded by[2]

$$\mathrm{Var}(\overline{X}_{\mathrm{mix-exp}}) \;\lesssim\; \frac{1}{B\big(1+(K-1)\rho\big)}\,\Sigma. \tag{71}$$

**Comparison.** Eq. (69) shows variance scaling as $\frac{1}{B}\big[(1-\rho)+\rho\cdot\frac{1+(K-1)\Gamma}{K}\big]\Sigma$, while Eq. (71) gives a simpler bound $\approx \frac{1}{B(1+(K-1)\rho)}\,\Sigma$. Both reduce to the baseline $\frac{1}{B}\Sigma$ when $\rho=0$ (no symmetric pockets). As $\rho\to 1$ and $\Gamma\to 0$, the effective sample size increases from $B$ to $BK$, recovering the ideal variance shrinkage.

**Comparison with the baseline and effective sample size.** The baseline variance in (64) is $\mathrm{Var}(\overline{X}_{\mathrm{base}})=\frac{1}{B}\Sigma$. Comparing with (69) gives

$$\frac{\mathrm{Var}(\overline{X}_{\mathrm{mix-avg}})}{\mathrm{Var}(\overline{X}_{\mathrm{base}})} = (1-\rho) + \rho\cdot\frac{1+(K-1)\Gamma}{K}. \tag{72}$$

When the across-view correlation $\Gamma$ is small (the ideal case of independent or weakly correlated views—typically promoted by *random mini-batching, target networks/dropout noise, or perturbations in representation*; cf. the textual discussion on variance reduction via view sharing), (72) simplifies to $(1-\rho)+\rho/K$. Its reciprocal can be interpreted as an *effective sample size multiplier*:

$$\frac{1}{(1-\rho)+\rho/K} = \frac{K}{K-(K-1)\rho} = 1+(K-1)\rho+O(\rho^2) \quad \text{(for small $\rho$).} \tag{73}$$

By contrast, the fully expanded estimator in (71) yields a more direct (and unbiased under the independence approximation) amplification law: treating all $N_{\mathrm{valid}}=B\big(1+(K-1)\rho\big)$ valid views as independent atoms, the variance-to-baseline ratio is approximately $1/(1+(K-1)\rho)$, equivalent to an *effective sample size* growing from $B$ to

$$N_{\mathrm{eff}} \;\approx\; B\cdot\big(1+(K-1)\rho\big). \tag{74}$$

### G.3. Proof of Theorem 3.3

We construct a directed graph over the batch index set $V=[B]=\{1,2,\ldots,B\}$. The candidate edge set $\mathcal{E}_0\subseteq V\times V$ is generated from scores

$$w_i = \sigma(\Delta_i/\tau)\cdot c_i, \tag{75}$$

which are sorted in descending order (ties broken by a fixed deterministic rule). To avoid spurious long-range connections, edges are only considered within a local $k$-NN neighborhood. A candidate edge is proposed whenever the threshold is exceeded and $\Delta_i > 0$, in which case $u_i \to v_i$ is added to $\mathcal{E}_0$. These engineering details are not essential to the theory and are included only to clarify the source and ordering of edges. As described in Sec. 3.2 of the main text, applying the *add-unless-cycle* rule—namely, inserting edges greedily by descending $w_i$ and discarding any edge that would create a cycle—yields an acyclic graph $D$ that is a maximal acyclic subgraph with respect to the given ordering.

Fix a total order $\succ$ over the candidate edge set $\mathcal{E}_0$, denoted as $e^{(1)} \succ e^{(2)} \succ \cdots \succ e^{(M)}$ ($M=|\mathcal{E}_0|$). Initialize $D^{(0)}=\varnothing$. For $t=1,2,\ldots,M$, update

$$D^{(t)} = \begin{cases} D^{(t-1)} \cup \{e^{(t)}\}, & \text{if } D^{(t-1)}\cup\{e^{(t)}\} \text{ is acyclic;} \\ D^{(t-1)}, & \text{if } D^{(t-1)}\cup\{e^{(t)}\} \text{ forms a cycle.} \end{cases} \tag{76}$$

Finally output $D := D^{(M)}$. This is exactly the *add-unless-cycle* rule: edges are processed in descending order of score, added greedily if they do not create a cycle, and skipped otherwise.

**Lemma G.5** (Single-edge cycle criterion). *Let $G=(V,E)$ be a directed acyclic graph (DAG), and let $e=(u\to v)$ be a directed edge not in $E$. Then*

$$G\cup\{(u\to v)\} \text{ contains a cycle} \quad\Longleftrightarrow\quad \text{there exists a directed path from $v$ to $u$ in $G$.} \tag{77}$$

---

[2]If intra-$i$ correlations across different $k$ are nonzero, one can apply the equal correlation upper bound: for each $i\in\mathcal{I}_{\mathrm{sym}}$, $\mathrm{Var}\big(\frac{1}{K}\sum_k X_{i,k}\big)=\frac{1+(K-1)\Gamma}{K}\,\Sigma$; substituting this into the expansion estimate yields an upper bound of the same order as (69).

*Proof of Lemma G.5.* ($\Rightarrow$) If $G \cup \{u \to v\}$ contains a cycle, that cycle must involve the newly added edge $u \to v$. Let $P_{v \leadsto u}$ denote the remainder of the cycle from $v$ back to $u$. Since $P_{v \leadsto u}$ lies entirely in $G$, we conclude that there exists a directed path from $v$ to $u$ in $G$.

($\Leftarrow$) Conversely, if there exists a directed path $P_{v \leadsto u}$ in $G$, then in $G \cup \{u \to v\}$ the concatenation $u \to v \circ P_{v \leadsto u}$ forms a directed cycle. Hence the equivalence holds. $\square$

**Conclusion 1 (Acyclicity).** We prove by induction that $D^{(t)}$ is acyclic for all $t$.

- *Base case:* $D^{(0)} = \varnothing$ is clearly acyclic.

- *Inductive step:* Assume $D^{(t-1)}$ is acyclic. If at step $t$ the edge $e^{(t)}$ is added, then by rule it does *not* create a cycle; otherwise we keep $D^{(t)} = D^{(t-1)}$, which remains acyclic. Therefore $D^{(t)}$ is always acyclic.

Hence the final $D = D^{(M)}$ is a DAG, consistent with the statement in the original text that the resulting subgraph $D$ is acyclic.

**Conclusion 2 (Maximality with respect to the given order).** We show that for every skipped candidate edge $e \in \mathcal{E}_0 \setminus D$, adding it to the final $D$ must produce a directed cycle. By construction, each such edge $e$ was considered at some step $t$, with $D^{(t-1)}$ already selected at that point, and $e = e^{(t)}$ was *skipped*. According to rule (76) (second branch) and Lemma G.5, the only reason for skipping is that

$$\text{there exists a directed path in } D^{(t-1)} \text{ from the head of } e \text{ to its tail.} \tag{78}$$

Since the algorithm never deletes selected edges, this path remains in every later $D^{(s)}$, and in particular in the final $D = D^{(M)}$. Thus if we were to add $e$ to $D$, Lemma G.5 implies a cycle would be formed.

Therefore $D$ is a *maximal* acyclic subgraph with respect to the fixed edge order: any remaining candidate edge, if added, would necessarily form a cycle. This matches exactly the statement of Theorem 3.3 in the original text: The resulting subgraph is acyclic; moreover, under the rule 'add unless a cycle is formed,' it is maximal with respect to the given order.

### G.4. Proof of Theorem 3.4

Given a fixed DAG $D = ([B], E)$ and a finite batch size $B < \infty$, define

$$\mathcal{C} := \left\{ \hat{V} \in \mathbb{R}^B : \hat{V}(u) \geq \hat{V}(v) + \delta, \ \forall (u \to v) \in E \right\} \tag{79}$$

as the monotone (partial order) feasible region, where $\delta \geq 0$ is fixed. Define the fidelity (quadratic) term as

$$F(\hat{V}) := \frac{1}{B} \sum_{i=1}^{B} \left( \hat{V}(i) - V_\theta(i) \right)^2 = \frac{1}{B} \| \hat{V} - V_\theta \|_2^2, \tag{80}$$

and the smooth hinge penalty as

$$G(\hat{V}) := \frac{1}{|E|} \sum_{(u \to v) \in E} \left[ \delta + \hat{V}(v) - \hat{V}(u) \right]_+^2, \quad [x]_+ := \max(0, x). \tag{81}$$

Then the isotonic projection problem and its penalty approximation are respectively

$$\text{(Exact)} \quad \hat{V}^\star \in \arg\min_{\hat{V} \in \mathcal{C}} F(\hat{V}), \tag{82}$$

$$\text{(Penalty)} \quad \hat{V}_\mu \in \arg\min_{\hat{V} \in \mathbb{R}^B} L_\mu(\hat{V}) := F(\hat{V}) + \mu \, G(\hat{V}) \quad (\mu > 0). \tag{83}$$

It remains to prove: as $\mu \to \infty$, any limit point $\lim_j \hat{V}_{\mu_j}$ coincides with the unique exact solution $\hat{V}^\star$, and moreover, for sufficiently large $\mu$, the constraint violation can be made arbitrarily small.

**Linearized Representation and Basic Properties.** For notational convenience, introduce the *directed incidence matrix* $A \in \mathbb{R}^{|E| \times B}$ of edge–variable form: for each edge $e = (u \to v)$, the row $A_e$ is defined by

$$A_{e,u} = -1, \quad A_{e,v} = +1, \quad A_{e,i} = 0 \ (i \notin \{u, v\}). \tag{84}$$

Then all inequalities can be written as $A\hat{V} \le -\delta \, \mathbf{1}_{|E|}$; at the same time,

$$G(\hat{V}) = \frac{1}{|E|} \sum_e \left[ \delta + (A\hat{V})_e \right]_+^2. \tag{85}$$

It is easy to see that:

- $\mathcal{C}$ is a closed convex polytope; since $D$ is a DAG (acyclic), the feasible region is nonempty (see Lemma G.6).

- $F$ is a quadratic function that is $2/B$-strongly convex and coercive; $G$ is a continuous convex function (a squared hinge composed with an affine map).

- The exact problem (82) has a unique solution (strong convexity + closed convex feasible region), denoted by $\hat{V}^\star$.

**Lemma G.6** (Feasibility of the Constraint Set). *If $D$ is acyclic, then there exists $\hat{V} \in \mathcal{C}$.*

*Proof of Lemma G.6.* Take a topological ordering $i_1, \ldots, i_B$ of $D$ and define

$$\hat{V}(i_\ell) := L - (\ell - 1)\delta, \qquad L \in \mathbb{R} \text{ arbitrary.} \tag{86}$$

If $i_p \to i_q$ is an edge, then $p < q$ in the topological order, and hence

$$\hat{V}(i_p) - \hat{V}(i_q) = \delta \cdot (q - p) \ge \delta, \tag{87}$$

which implies $\hat{V}(i_p) \ge \hat{V}(i_q) + \delta$. Thus $\hat{V} \in \mathcal{C}$. $\qquad\square$

**Lemma G.7** (Uniqueness). *The exact problem (82) admits a unique solution.*

*Proof of Lemma G.7.* Since $F$ is strictly convex and $\mathcal{C}$ is a closed convex set, the minimizer of $F$ over $\mathcal{C}$ must be unique. $\qquad\square$

**Core Inequality: Violation Upper Bound Decays to Zero.**

**Lemma G.8** (Violation Upper Bound of Order $1/\mu$). *For any minimizer $\hat{V}_\mu$ of the penalty problem with $\mu > 0$, it holds that*

$$G(\hat{V}_\mu) \le \frac{F(\hat{V}^\star)}{\mu}, \qquad F(\hat{V}_\mu) \le F(\hat{V}^\star). \tag{88}$$

*In particular, as $\mu \to \infty$ we have $G(\hat{V}_\mu) \to 0$.*

*Proof of Lemma G.8.* By the optimality of $\hat{V}_\mu$ and the feasibility of $\hat{V}^\star$ (for which $G(\hat{V}^\star) = 0$),

$$F(\hat{V}_\mu) + \mu \, G(\hat{V}_\mu) \le F(\hat{V}^\star) + \mu \, G(\hat{V}^\star) = F(\hat{V}^\star). \tag{89}$$

Thus $F(\hat{V}_\mu) \le F(\hat{V}^\star)$ and $\mu \, G(\hat{V}_\mu) \le F(\hat{V}^\star)$, which yields (88). $\qquad\square$

**Boundedness and Limit Feasibility.**

**Lemma G.9** (Subsequence Limits are Feasible). *There exists a constant $C > 0$ such that $\|\hat{V}_\mu\|_2 \le C$ (independent of $\mu$). For any subsequence $\mu_j \uparrow \infty$, if the corresponding minimizers admit a convergent subsequence $\hat{V}_{\mu_{j_\ell}} \to \overline{V}$, then $\overline{V} \in \mathcal{C}$.*

*Proof of Lemma G.9.* From $F(\hat{V}_\mu) \le F(\hat{V}^\star)$ and the strong convexity and coercivity of $F$, we obtain a uniform bound on $\|\hat{V}_\mu\|_2$ (e.g., via the quadratic lower bound $F(\hat{V}) \ge \frac{1}{B}\|\hat{V}\|_2^2 - \frac{2}{B}\langle \hat{V}, V_\theta \rangle + \frac{1}{B}\|V_\theta\|_2^2$). Compactness then yields subsequential convergence. Moreover, by Lemma G.8, $G(\hat{V}_{\mu_{j_\ell}}) \to 0$. Since $G(\cdot)$ is continuous and each hinge term $[\delta + (A\hat{V})_e]_+^2 \ge 0$, we have $G(\overline{V}) = 0$. Thus $\delta + (A\overline{V})_e \le 0$ for every edge $e$, i.e. $\overline{V} \in \mathcal{C}$. $\qquad\square$

**Limit Optimality (Direct Proof via $\Gamma$-limit / epi-convergence).** Let the indicator function be $\iota_\mathcal{C}(\hat{V}) = 0$ if $\hat{V} \in \mathcal{C}$, and $+\infty$ otherwise. Define the limit objective

$$L_\infty(\hat{V}) := F(\hat{V}) + \iota_\mathcal{C}(\hat{V}), \tag{90}$$

which corresponds to the exact problem. We prove that $L_\mu$ epi-converges to $L_\infty$ as $\mu \to \infty$, so that any limit point of minimizers of $L_\mu$ is a minimizer of $L_\infty$. Below we provide an equivalent *direct* comparison proof.

**Lemma G.10** (Lower and Upper Limit Comparison)**.** *For any subsequence $\mu_j \uparrow \infty$ and its minimizers $\hat{V}_{\mu_j}$, if $\hat{V}_{\mu_j} \to \overline{V}$, then $F(\overline{V}) \le F(\hat{V}^\star)$ and $\overline{V} \in \mathcal{C}$; by Lemma G.7 it follows that $\overline{V} = \hat{V}^\star$.*

*Proof of Lemma G.10.* On the one hand, since $\hat{V}^\star$ is feasible and $G(\hat{V}^\star) = 0$, we have

$$L_{\mu_j}(\hat{V}_{\mu_j}) \le L_{\mu_j}(\hat{V}^\star) = F(\hat{V}^\star). \tag{91}$$

Taking the lower limit and using $G(\hat{V}_{\mu_j}) \ge 0$ gives

$$\liminf_{j \to \infty} F(\hat{V}_{\mu_j}) \le \liminf_{j \to \infty} L_{\mu_j}(\hat{V}_{\mu_j}) \le F(\hat{V}^\star). \tag{92}$$

On the other hand, by Lemma G.9, $\overline{V} \in \mathcal{C}$, and since $F$ is continuous,

$$F(\overline{V}) = \lim_{j \to \infty} F(\hat{V}_{\mu_j}) \le F(\hat{V}^\star). \tag{93}$$

But $\hat{V}^\star$ is the unique minimizer of $\min_\mathcal{C} F$ (Lemma G.7), hence $F(\overline{V}) = F(\hat{V}^\star)$ and $\overline{V} = \hat{V}^\star$. $\qquad\square$

By Lemma G.10, any limit point must equal $\hat{V}^\star$; since $\hat{V}^\star$ is unique, the whole sequence converges, i.e., $\hat{V}_\mu \to \hat{V}^\star$ (otherwise there would be two distinct limit points).

**Quantitative Version of Violation Can Be Made Arbitrarily Small.** From Lemma G.8, we have $G(\hat{V}_\mu) \le F(\hat{V}^\star)/\mu$. Thus, given any $\varepsilon > 0$, choosing $\mu \ge F(\hat{V}^\star)/\varepsilon$ ensures $G(\hat{V}_\mu) \le \varepsilon$. Since $G(\hat{V}) = \frac{1}{|E|} \sum_e [\delta + (A\hat{V})_e]_+^2$ is the average of squared violations over all edges, it follows that for each edge the positive violation satisfies

$$\left[\delta + (A\hat{V}_\mu)_e\right]_+ \le \sqrt{|E|\, G(\hat{V}_\mu)} \le \sqrt{|E|\, \varepsilon}. \tag{94}$$

This shows that when $\mu$ is sufficiently large, the constraint violation on every edge can be made arbitrarily small.

### G.5. Proof of Theorem 3.5

On a fixed directed acyclic graph (DAG) $D = ([B], E)$, we perform the *DAG isotonic projection*: given the network's raw estimate $V_\theta \in \mathbb{R}^B$ on this batch, we solve

$$\hat{V}^\star \in \arg\min_{\hat{V} \in \mathbb{R}^B} \frac{1}{B} \sum_{i=1}^{B} \left(\hat{V}(i) - V_\theta(i)\right)^2 \quad \text{s.t.} \quad \hat{V}(u) \ge \hat{V}(v) + \delta, \ \forall (u{\to}v) \in E, \ \delta \ge 0, \tag{95}$$

which is exactly the DAG-based isotonic projection of Eq. 14 in the manuscript (where $\delta$ is an optional *margin*). In practice, the authors adopt a differentiable surrogate with a squared hinge penalty,

$$\mathcal{L}_{\text{iso}} = \frac{1}{B} \sum_i (\hat{V}(i) - V_\theta(i))^2 + \mu \cdot \frac{1}{|E|} \sum_{(u{\to}v) \in E} \left[\delta + \hat{V}(v) - \hat{V}(u)\right]_+^2, \tag{96}$$

as given in Eq. 15 of the manuscript, and prove that as $\mu \to \infty$, its minimizers converge to the exact isotonic solution (Theorem 3.4), namely the penalty consistency result. Therefore, in the following we may directly work with the *exact feasible* solution $\hat{V}^\star$; if the $\mu$-penalty surrogate is used, it suffices to take $\mu$ sufficiently large to suppress the violation on every edge to an arbitrarily small level, making it equivalent to (95) (see the above theorem and discussion)

Theorem 3.5 of the manuscript states that for any DAG $D$, the preference relation induced by $\hat{V}^{\star 3}$ contains no directed cycles; moreover, when $E$ covers a subset of the ground-truth partial order, $\hat{V}^{\star}$ is consistent with that partial order.

To avoid ambiguity, we precisely define the *preference relation induced by the skeleton $D$* as

$$u \succeq_D^{\hat{V}^{\star}} v :\Longleftrightarrow$$

there exists a directed path from $u$ to $v$ in $D$ (with length denoted $\ell(u \rightsquigarrow v)$), and the comparison is based on $\hat{V}^{\star}$. (97)

Note that since $D$ is a DAG, its *reachability relation* is itself a partial order (reflexive, transitive, and antisymmetric). Meanwhile, (95) enforces *monotonicity* along each edge, so monotone transitivity is automatically inherited along any *path* (see the lemma below), which is exactly the manuscript's explanation of Theorem 3.5: path accumulation + monotone transitivity $\Rightarrow$ acyclicity.

**Lemma G.11** (Monotone Transitivity along Paths). *Let $x_0 \to x_1 \to \cdots \to x_m$ be any directed path in $D$ ($m \in \mathbb{N}$). Then for $\hat{V}^{\star}$ we have*

$$\hat{V}^{\star}(x_0) \geq \hat{V}^{\star}(x_m) + m\,\delta. \tag{98}$$

*Proof of Lemma G.11.* By the edge constraints in (95), we obtain step by step:

$$\hat{V}^{\star}(x_0) \geq \hat{V}^{\star}(x_1) + \delta, \quad \hat{V}^{\star}(x_1) \geq \hat{V}^{\star}(x_2) + \delta, \quad \ldots, \quad \hat{V}^{\star}(x_{m-1}) \geq \hat{V}^{\star}(x_m) + \delta.$$

Adding these inequalities yields $\hat{V}^{\star}(x_0) \geq \hat{V}^{\star}(x_m) + m\,\delta$. This matches exactly the manuscript's explanation: constraints along a path *accumulate* numerically, preserving monotone transitivity. $\square$

**Conclusion 1 (Acyclicity).** We prove: *the preference relation $\succeq_D^{\hat{V}^{\star}}$ defined in (97) contains no directed cycles.*

Suppose, for contradiction, that there exist distinct nodes $x_0, \ldots, x_{m-1}$ forming a directed cycle such that

$$x_0 \succeq_D^{\hat{V}^{\star}} x_1 \succeq_D^{\hat{V}^{\star}} \cdots \succeq_D^{\hat{V}^{\star}} x_{m-1} \succeq_D^{\hat{V}^{\star}} x_0.$$

By (97), for each pair $(x_i, x_{i+1})$ ($x_m \equiv x_0$) there exists a $D$-path $P_i : x_i \rightsquigarrow x_{i+1}$. Applying (98) to each path gives

$$\hat{V}^{\star}(x_i) \geq \hat{V}^{\star}(x_{i+1}) + \ell(P_i)\,\delta, \qquad i = 0, 1, \ldots, m-1. \tag{99}$$

Summing (99) over $i$ (with cancellation of intermediate terms) yields

$$\underbrace{\hat{V}^{\star}(x_0) - \hat{V}^{\star}(x_0)}_{=0} \geq \left(\sum_{i=0}^{m-1} \ell(P_i)\right) \delta.$$

Let the total length be $L := \sum_{i=0}^{m-1} \ell(P_i) \in \mathbb{N}$, then $0 \geq L\,\delta$.

*(i) If $\delta > 0$*, this forces $L = 0$, which in turn forces each $P_i$ to have length 0, i.e. $x_i = x_{i+1}$, contradicting the assumption that the cycle consists of distinct nodes. Hence no cycle exists.

*(ii) If $\delta = 0$*, (99) degenerates to $\hat{V}^{\star}(x_i) \geq \hat{V}^{\star}(x_{i+1})$. If there also exists a path $x_{m-1} \rightsquigarrow x_0$, then the reachability relation of $D$ would contain a directed cycle $x_0 \rightsquigarrow x_1 \rightsquigarrow \cdots \rightsquigarrow x_{m-1} \rightsquigarrow x_0$, contradicting the assumption that $D$ is a DAG. Therefore, even when $\delta = 0$, the preference induced along the reachability skeleton of $D$ is acyclic. (This matches the manuscript's explanation that the edge constraints preserve monotone transitivity along paths, and thus cycles cannot occur.) $\square$

---

[3]The manuscript describes it as the induced preference relation $\hat{V}^{\star}(u) \geq \hat{V}^{\star}(v)$ has no directed cycles. The intuitive justification is that along each edge the constraint is $\hat{V}(u) \geq \hat{V}(v) + \delta$; along a path, such edge-wise constraints *accumulate* and preserve monotone transitivity, therefore a directed cycle is *impossible*.

**Conclusion 2 (Consistency with the True Partial Order).** Let the true partial order be $([B], \succeq_\star)$, whose Hasse diagram or some covering subgraph contains the edge set of $D$, i.e.

$$(u \to v) \in E \implies u \succeq_\star v. \tag{100}$$

We prove that $\hat{V}^\star$ is consistent with the true partial order within the *coverage range* of $D$.

First, by (95), for each $(u \to v) \in E$ we have $\hat{V}^\star(u) \geq \hat{V}^\star(v) + \delta$, so it is impossible to encounter a *reversed* case where $u \succeq_\star v$ but $\hat{V}^\star(u) < \hat{V}^\star(v) + \delta$. Furthermore, if $u$ is reachable to $v$ in $D$ (i.e. $u \rightsquigarrow v$), then by (98) we have $\hat{V}^\star(u) \geq \hat{V}^\star(v) + \ell(u \rightsquigarrow v)\delta$, which again rules out reversal. This shows that for all true partial-order pairs $(u, v)$ covered by $D$ (including its transitive closure), the values of $\hat{V}^\star$ *preserve* the true order direction— that is, the isotonic projection "only repairs violations" and never inverts correct relations. This matches exactly the statement in the manuscript.

**Consistency with the Differentiable Surrogate.** If one adopts the differentiable surrogate $\mathcal{L}_{\mathrm{iso}}$ (Eq. 15 in the manuscript), then by the penalty consistency result Theorem 3.4), when $\mu$ is sufficiently large the *violation on each edge* can be reduced to an arbitrarily small level. Thus the path accumulation property of (98) holds up to an arbitrarily small tolerance; taking the limit recovers the exact case of (95), so the above two conclusions also hold in the large-$\mu$ setting.

### G.6. Proof of Theorem 3.6

We perform the batchwise isotonic step on a fixed DAG $D = ([B], E(D))$, obtaining $\hat{V}$ by solving

$$\mathcal{L}_{\mathrm{iso}}(\hat{V}) = \underbrace{\frac{1}{B} \sum_{i=1}^{B} \left(\hat{V}(i) - V_\theta(i)\right)^2}_{=:F(\hat{V})} + \mu \cdot \underbrace{\frac{1}{|E(D)|} \sum_{(u \to v) \in D} \left[\delta + \hat{V}(v) - \hat{V}(u)\right]_+^2}_{=:G(\hat{V})}, \tag{101}$$

with an optional $\mathcal{L}_{\mathrm{rank}}$ for additional stability (which does not affect the conclusion). See Eqs. 14–16 of the manuscript for definitions and explanations.

The *expected Bellman residual* is defined as (Eq. 19 in the manuscript):

$$\mathcal{E}_{\mathrm{Bell}}(\theta) = \mathbb{E}_{(s,a)}\left[\left(Q_\theta(s, a) - \mathcal{T}^* Q_\theta(s, a)\right)^2\right]. \tag{102}$$

The goal inequality of Theorem 3.6 (Eq. 32 in the manuscript) states that there exist constants $C_1, C_2 > 0$ such that after one parameter update,

$$\mathcal{E}_{\mathrm{Bell}}(\theta^+) \leq \mathcal{E}_{\mathrm{Bell}}(\theta) - C_1 \cdot \frac{1}{|E(D)|} \sum_{(u \to v) \in D} \left[\delta + \hat{V}(v) - \hat{V}(u)\right]_+^2 + C_2 \|\hat{V} - V_\theta\|_2^2. \tag{103}$$

Moreover, when $\mu$ is sufficiently large (with $\delta \geq 0$), the term $\|\hat{V} - V_\theta\|_2^2$ is controlled by $\mathcal{L}_{\mathrm{iso}}$, ensuring that the overall Bellman residual decreases.

**Assumption (i.i.d. approximation and stable step size).** As described in the manuscript, in one small-step update we adopt the i.i.d. approximation: the sampling noise is replaced by the expectation under the current distribution, and we assume the learning rate is sufficiently small so that the first-order *Descent Lemma* holds. In addition, the edge set $E(D)$ covers the *high-confidence* local order (direction-consistent), meaning that violations along these edges indicate the relative improvement direction desired by the Bellman objective (see the paragraph following the theorem statement in the manuscript).

**Edge–variable linearization and first-order optimality of the penalty.** For each edge $e = (u \to v)$, define

$$\xi_e(\hat{V}) := \left[\delta + \hat{V}(v) - \hat{V}(u)\right]_+, \qquad \Xi(\hat{V}) := \left(\xi_e(\hat{V})\right)_{e \in E(D)} \in \mathbb{R}_{\geq 0}^{|E(D)|}.$$

Let $A \in \mathbb{R}^{|E(D)| \times B}$ denote the edge–variable incidence matrix: for row $A_e$ corresponding to edge $e = (u \to v)$, we set $A_{e,u} = -1$, $A_{e,v} = +1$, and all other entries to 0. Then

$$G(\hat{V}) = \frac{1}{|E(D)|} \sum_e \xi_e(\hat{V})^2 = \frac{1}{|E(D)|} \|\Xi(\hat{V})\|_2^2.$$

From the smooth hinge penalty structure of (101) (Eq. 15 in the manuscript), we obtain the first-order directional derivative condition (or the smooth version of the KKT condition):

$$\frac{2}{B} \left( \hat{V} - V_\theta \right) \; + \; \frac{2\mu}{|E(D)|} \, A^\top \zeta(\hat{V}) \; = \; 0, \quad \text{where } \zeta_e(\hat{V}) := \begin{cases} \xi_e(\hat{V}), & \delta + \hat{V}(v) - \hat{V}(u) > 0, \\ 0, & \text{otherwise.} \end{cases} \tag{104}$$

Equivalently,

$$\hat{V} - V_\theta \; = \; - \underbrace{\frac{\mu B}{|E(D)|}}_{=:c_\mu} \, A^\top \zeta(\hat{V}), \qquad \zeta(\hat{V}) \in \mathbb{R}_{\geq 0}^{|E(D)|}, \; \zeta_e(\hat{V}) \leq \xi_e(\hat{V}). \tag{105}$$

This shows that the *batchwise displacement* $\Delta V := \hat{V} - V_\theta$ given by the isotonic step is aligned with the (negative) incidence aggregation of edge violations $\big( \xi_e(\hat{V}) \big)$. For each violated edge $(u \to v)$, $\Delta V(u)$ is *increased* while $\Delta V(v)$ is *decreased*, exactly matching the expected direction of repairing monotonicity. This coincides with the interpretation given in Eqs. 15–16 of the manuscript.

**Descent Lemma for Bellman Residual.** Define $\mathcal{J}(V) := \mathcal{E}_{\mathrm{Bell}}(\theta)$ but viewed as a function of $V$ (through the chain dependence $Q_\theta \mapsto V_\theta(s) = \max_a Q_\theta(s, a)$; formally, one can fix $\theta$ and abstract the first-order impact of one update in the $V$-space). Under the i.i.d. approximation and a stable step size, by the standard Descent Lemma ($L$-smoothness) we obtain, for any direction $\Delta V$,

$$\mathcal{J}(V_\theta + \Delta V) \; \leq \; \mathcal{J}(V_\theta) \; + \; \langle \nabla \mathcal{J}(V_\theta), \, \Delta V \rangle \; + \; \frac{L}{2} \, \|\Delta V\|_2^2. \tag{106}$$

We will take $\Delta V = \hat{V} - V_\theta$, and align $\mathcal{J}(V_\theta + \Delta V)$ with $\mathcal{E}_{\mathrm{Bell}}(\theta^+)$ (equivalent to a one-step small update, ignoring higher-order terms). Thus, the proof of (103) reduces to establishing a *negative lower bound* for the inner product term $\langle \nabla \mathcal{J}(V_\theta), \hat{V} - V_\theta \rangle$.

**Directional consistency under edge coverage and inner-product bound.** The semantics of edge coverage of high-confidence local orderings (as explained below Eq. 32 in the manuscript) can be formalized as follows: there exist constants $\kappa > 0$ and $M \geq 0$ such that, for all violated edges $(\delta + \hat{V}(v) - \hat{V}(u) > 0)$, along the direction of raising $u$ and lowering $v$, the *gradient of the Bellman residual* has a consistent negative projection, with strength linearly proportional to the violation magnitude. Aggregated, this can be written as

$$\left\langle \nabla \mathcal{J}(V_\theta), \, -A^\top \zeta(\hat{V}) \right\rangle \; \leq \; -\kappa \, \|\zeta(\hat{V})\|_2^2 \; + \; M \, \big\| A^\top \zeta(\hat{V}) \big\|_2^2. \tag{107}$$

Intuitively, $\zeta_e(\hat{V}) > 0$ exactly indicates an edge $(u \to v)$ where $u$ should be raised relative to $v$ (see the explanation after Eq. 32), so $\nabla \mathcal{J}$ has a negative directional derivative in alignment with the violation, and its magnitude scales linearly with the violation. The constant $M$ captures the quadratic (Lipschitz) correction due to imperfect coverage or noise.

**Combining (107) with (105).** From (105), we have $\hat{V} - V_\theta = -c_\mu A^\top \zeta(\hat{V})$. Substituting into (107) yields

$$\begin{aligned} \left\langle \nabla \mathcal{J}(V_\theta), \, \hat{V} - V_\theta \right\rangle &= -c_\mu \left\langle \nabla \mathcal{J}(V_\theta), \, A^\top \zeta(\hat{V}) \right\rangle \\ &\leq -c_\mu \Big( \kappa \, \|\zeta(\hat{V})\|_2^2 \; - \; M \, \|A^\top \zeta(\hat{V})\|_2^2 \Big). \end{aligned} \tag{108}$$

On the other hand, $\|\hat{V} - V_\theta\|_2^2 = c_\mu^2 \, \|A^\top \zeta(\hat{V})\|_2^2$. Thus, combining (108) with (141), we obtain

$$\begin{aligned} \mathcal{J}(\hat{V}) &\leq \mathcal{J}(V_\theta) \; - \; c_\mu \, \kappa \, \|\zeta(\hat{V})\|_2^2 \; + \; c_\mu \, M \, \|A^\top \zeta(\hat{V})\|_2^2 \; + \; \frac{L}{2} \, c_\mu^2 \, \|A^\top \zeta(\hat{V})\|_2^2 \\ &= \mathcal{J}(V_\theta) \; - \; \underbrace{c_\mu \, \kappa}_{=: \, C_1'} \, \|\zeta(\hat{V})\|_2^2 \; + \; \underbrace{\Big( M \, c_\mu + \frac{L}{2} \, c_\mu^2 \Big)}_{=: \, C_2'} \, \|A^\top \zeta(\hat{V})\|_2^2. \end{aligned} \tag{109}$$

**Rewriting norms and objective terms into the theorem form.** Note that $\|\zeta(\hat{V})\|_2^2 = \sum_e \left[\delta + \hat{V}(v) - \hat{V}(u)\right]_+^2$, and $\|A^\top \zeta(\hat{V})\|_2^2 = c_\mu^{-2}\|\hat{V} - V_\theta\|_2^2$. Normalizing the two terms in (109) by $|E(D)|$, and recalling that $\mathcal{J}$ corresponds to $\mathcal{E}_{\text{Bell}}$, we obtain constants $C_1 = \frac{C_1'}{|E(D)|} > 0, \quad C_2 = \frac{C_2'}{c_\mu^2} > 0$ such that

$$\mathcal{E}_{\text{Bell}}(\theta^+) \;\leq\; \mathcal{E}_{\text{Bell}}(\theta) \;-\; C_1 \cdot \frac{1}{|E(D)|} \sum_{(u \to v) \in D} \left[\delta + \hat{V}(v) - \hat{V}(u)\right]_+^2 \;+\; C_2 \,\|\hat{V} - V_\theta\|_2^2. \tag{110}$$

This is exactly the form of (103) (equating the one-step effect of $\hat{V}$ with the small update $\theta^+$). As explained below Eq. 32, the negative term corresponds to residual reduction by repairing monotonicity along violated edges, while the positive term corresponds to the cost of not drifting too far from the original network.

**Controlling $\|\hat{V} - V_\theta\|_2^2$ with $\mathcal{L}_{\text{iso}}$.** From (101), $F(\hat{V}) = \frac{1}{B}\|\hat{V} - V_\theta\|_2^2 \leq \mathcal{L}_{\text{iso}}(\hat{V})$ holds directly. Moreover, Theorem 3.4 (Penalty consistency) ensures that when $\mu \to \infty$, $\hat{V}$ can approximate the hard isotonic solution arbitrarily closely, so both $G(\hat{V})$ and $F(\hat{V})$ are controlled by $\mathcal{L}_{\text{iso}}$. In particular, $\|\hat{V} - V_\theta\|_2^2 \leq B\,\mathcal{L}_{\text{iso}}(\hat{V})$. Therefore, for sufficiently large $\mu$ (and $\delta \geq 0$), the second term in (110) cannot outweigh the benefit of the first term, yielding an *overall decrease in the expected Bellman residual* (consistent with the explanation after Eq. 32).

### G.7. Proof of Theorem 3.7

We prove that when small cycles arise within a batch due to non-transitive structures (e.g., Rock–Paper–Scissors) or noise, then after applying *DAG-ification* and the subsequent *DAG isotonic projection*, the variance upper bound of the value sequence $\{V_{\theta_t}\}$ within the window decreases monotonically with respect to $\lambda_{\text{iso}}$ and the edge coverage ratio. Moreover, when the edges cover the dominant directions of the true partial order, this upper bound is strictly smaller than that of the unregularized baseline by a fixed proportion. The formal statement is given in Theorem 3.7.[4] The role of $\lambda_{\text{iso}}$ in the overall objective is explicitly described as acting like a brake on value propagation to reduce oscillations.

**Notation and Formalization of the Two Operations.** (1) **DAG-ification.** Let the batch nodes be $[B] = \{1, \ldots, B\}$. Using confidence-weighted edge scores, a greedy add-edge-unless-cycle procedure is applied to obtain $D = ([B], E)$, which is a DAG and *maximal* with respect to the given edge order (Theorem 3.3).[5]

(2) **DAG Isotonic Projection.** Let the incidence matrix $A \in \mathbb{R}^{|E| \times B}$ correspond to edges $e = (u \to v)$, with row entries $A_{e,u} = -1$, $A_{e,v} = +1$, and all others zero. Define the *isotonic cone* (a closed convex polyhedral cone)

$$\mathcal{K}(D, \delta) \;:=\; \left\{V \in \mathbb{R}^B : \; AV \;\leq\; -\delta \cdot \mathbf{1}\right\},$$

which reduces to the standard DAG isotonic set when $\delta = 0$, while $\delta > 0$ imposes a margin on the edges.

The manuscript employs a *squared hinge* differentiable surrogate (Eq. 15):

$$\mathcal{L}_{\text{iso}}(V) = \tfrac{1}{B}\|V - V_\theta\|_2^2 + \mu \cdot \tfrac{1}{|E|} \sum_{(u \to v)} \left[\delta + V(v) - V(u)\right]_+^2,$$

and proves that as $\mu \to \infty$, its minimizers converge to the *exact* cone projection (Theorem 3.4).[6]

**Basic Tool (Convex Projection and Linearization).**

**Lemma G.12** (Nonexpansiveness of Euclidean Projection and Tangent-Cone Linearization). *Let $C \subset \mathbb{R}^B$ be a nonempty closed convex set. Then the Euclidean projection $\Pi_C$ is* nonexpansive:

$$\|\Pi_C(x) - \Pi_C(y)\|_2 \leq \|x - y\|_2,$$

---

[4]The intuitive explanation is that cycle cutting + slope enforcement suppresses oscillatory backtracking, and increasing $\lambda_{\text{iso}}$ makes the slope steeper, thus reducing oscillations.

[5]That is, no additional candidate edge can be added without forming a directed cycle.

[6]In particular, the squared hinge is flat when constraints are satisfied; smooth and stable when violated, with $\mu$ controlling the strength of order enforcement and $\delta > 0$ improving gradient flow near ties, as explained in the manuscript.

*and moreover* firmly nonexpansive*:*

$$\|\Pi_C(x) - \Pi_C(y)\|_2^2 \le \langle \Pi_C(x) - \Pi_C(y),\, x - y \rangle.$$

When $x^\star \in C$ and the projection is locally unique and smooth, $\mathrm{D}\Pi_C(x^\star)$ *equals the* orthogonal projection *onto the tangent space* $T_C(x^\star)$*, denoted* $P_{T_C(x^\star)}$*. Thus for small perturbations* $h$*,*

$$\Pi_C(x^\star + h) = x^\star + P_{T_C(x^\star)}h + o(\|h\|).$$

*Proof sketch.* This is a standard result in convex analysis: the projection operator is 1-Lipschitz and firmly nonexpansive; its Fréchet derivative at smooth points is the projection onto the tangent space. Below we will use the facts that *projection does not amplify errors* and *linearization yields a tangent projection.*

**One-Step Approximate Dynamics with Projection and Spectral Contraction.** Consider the one-step update in the $V$-space (absorbing the chain dependence of the value head on the parameters via first-order linearization):

$$\underbrace{V_{t+1}}_{\text{new estimate}} \approx \underbrace{\Pi_{\mathcal{K}(D,\delta)}\Big(V_t - \eta\, g_t\Big)}_{\text{DAG isotonic hard projection}} + \xi_t, \qquad g_t := \nabla_V \mathcal{L}_{\mathrm{TD}}(V_t),\ \ \eta > 0,$$

or, under the *differentiable surrogate*, a single gradient descent step:

$$V_{t+1} \approx V_t - \eta\Big( \underbrace{g_t}_{\text{TD gradient}} + \underbrace{\lambda_{\mathrm{iso}} \nabla_V G_t}_{\text{isotonic penalty term}} \Big) + \xi_t, \quad G_t := \frac{1}{|E|} \sum_{(u \to v)} [\delta + V_t(v) - V_t(u)]_+^2,$$

where $\xi_t$ collects sampling/target noise, with $\mathbb{E}[\xi_t] = 0$ and $\mathbb{E}[\xi_t \xi_t^\top] = \Sigma_\xi$. Note that $\nabla_V G_t = \frac{2}{|E|} A^\top \zeta_t$, with $\zeta_{t,e} = [\delta + V_t(v) - V_t(u)]_+ \ge 0$, which is consistent with the manuscript's explanation (after (15)) that "violated edges indicate the desired upward/downward adjustment direction."

Under the *exact projection* characterization, for a small perturbation $h_t := V_t - V^\star$ around a stationary point $V^\star \in \mathcal{K}(D,\delta)$, and letting $H_t := \nabla^2 \mathcal{L}_{\mathrm{TD}}(V^\star)$ (or a local averaged Hessian), Lemma G.12 yields the linearization

$$h_{t+1} \approx P_t\Big(I - \eta\, H_t\Big)h_t + \xi_t, \qquad P_t := P_{T_{\mathcal{K}}(V^\star)} \text{ (orthogonal projection onto the tangent space).} \tag{111}$$

Thus the one-step Jacobian is $A_t := P_t(I - \eta H_t)$. Since $P_t$ is an *orthogonal projection* (spectral norm 1) and nonexpansive,

$$\|A_t\|_2 = \|P_t(I - \eta H_t)\|_2 \le \|I - \eta H_t\|_2. \tag{112}$$

Whenever $(I - \eta H_t)$ has components outside the tangent space $P_t^\perp$, the inequality is *strict* (those non-tangent components are suppressed by the projection). This precisely captures the phenomenon of *spectral contraction* induced by "cycle removal + isotonicity."

**Embedding $\lambda_{\mathrm{iso}}$ and Edge Coverage into the Spectral Contraction Constant.** When using the differentiable surrogate, the linearized Jacobian can be rewritten as

$$\tilde{A}_t := I - \eta\Big(H_t + \lambda_{\mathrm{iso}} \underbrace{\nabla^2 G_t}_{\succeq 0}\Big).$$

Note that $\nabla^2 G_t = \frac{2}{|E|} A^\top D_t A \succeq 0$, where $D_t = \mathrm{diag}(\mathbb{I}\{\delta + V_t(v) - V_t(u) > 0\})$ is the diagonal indicator matrix of activated edges. Therefore, in the subspace spanned by the activated and covered edge directions $\mathcal{S}_t := \mathrm{span}\{A^\top e_e : D_{t,ee} = 1\}$, $\nabla^2 G_t$ has at least some *positive minimal eigenvalue* $\underline{\lambda}_t > 0$. When edge coverage is *more complete*, the lower bound of $\underline{\lambda}_t$ is larger; and when $\lambda_{\mathrm{iso}}$ increases, the spectral norm bound of $\tilde{A}_t$ becomes smaller. Formally, for any $x$,

$$\|(I - \eta(H_t + \lambda_{\mathrm{iso}} \nabla^2 G_t))x\|_2 \le \|(I - \eta H_t)x\|_2 - \eta\, \lambda_{\mathrm{iso}}\, \underline{\lambda}_t\, \|P_{\mathcal{S}_t} x\|_2,$$

(by minimal eigenvalue and positive semidefiniteness; $P_{\mathcal{S}_t}$ is the orthogonal projection onto $\mathcal{S}_t$). Combining with the "tangent projection" effect in (112) (shrinking/removing non-tangent components), we obtain constants $\alpha_{\text{base}} \in (0, 1)$ and a non-decreasing function $\phi(\lambda_{\text{iso}}, \text{cov}) \in (0, 1]$ (where cov denotes edge coverage rate/activation ratio) such that

$$\|A_t\|_2 \;\leq\; \alpha_{\text{base}} \cdot \big(1 - \phi(\lambda_{\text{iso}}, \text{cov})\big) \;:=:\; \alpha(\lambda_{\text{iso}}, \text{cov}), \tag{113}$$

and $\phi$ increases monotonically with $\lambda_{\text{iso}}$ and coverage, hence $\alpha(\lambda_{\text{iso}}, \text{cov})$ decreases monotonically with both. This corresponds to the manuscript's description that "increasing $\lambda_{\text{iso}}$ is like making the slope steeper, so pull-back oscillations are harder" (i.e., oscillations are reduced).

**Monotone Decrease of MSE/Variance Bound.** Let the batch window length be $W$, and denote the centered error $e_t := h_t - \mathbb{E}[h_t]$. From (111) and $\mathbb{E}[\xi_t] = 0$, under the *first-order approximation* we have

$$e_{t+1} \;=\; A_t\, e_t \;+\; \xi_t, \qquad \mathbb{E}[e_{t+1}] = A_t\, \mathbb{E}[e_t].$$

Taking expectations of the squared norm and using the uniform bound $\|A_t\|_2 \leq \alpha < 1$ (chosen according to (113)), we obtain

$$\mathbb{E}\|e_{t+1}\|_2^2 \;\leq\; \alpha^2\, \mathbb{E}\|e_t\|_2^2 \;+\; \text{tr}(\Sigma_\xi). \tag{114}$$

Unrolling $k$ steps: $\mathbb{E}\|e_t\|_2^2 \leq \alpha^{2k}\, \mathbb{E}\|e_{t-k}\|_2^2 + \frac{1-\alpha^{2k}}{1-\alpha^2}\, \text{tr}(\Sigma_\xi)$. Taking the supremum within a window of size $W$ and letting $k \to \infty$, we obtain the steady-state bound

$$\sup_{t \text{ in window}} \mathbb{E}\|e_t\|_2^2 \;\leq\; \frac{1}{1 - \alpha(\lambda_{\text{iso}}, \text{cov})^2}\, \text{tr}(\Sigma_\xi), \tag{115}$$

which clearly decreases monotonically with $\alpha(\lambda_{\text{iso}}, \text{cov})$. Combining with the properties of (113), we obtain the main claim: the variance bound decreases monotonically with both $\lambda_{\text{iso}}$ and edge coverage. This matches the manuscript's explanation that "Theorem 3.6 controls the expected error, while the penalty controls the *temporal variance*."

**Fixed-Ratio Improvement under Dominant-Direction Coverage.** If the edges cover the *dominant direction* of the true partial order, then there exist constants $c_0 \in (0, 1)$ and $\lambda_\star > 0$ such that once $\lambda_{\text{iso}} \geq \lambda_\star$, we have $\alpha(\lambda_{\text{iso}}, \text{cov}) \leq (1 - c_0)\, \alpha_{\text{base}}$, which is equivalent to

$$\frac{1}{1 - \alpha(\lambda_{\text{iso}}, \text{cov})^2} \;\leq\; \frac{1}{1 - (1 - c_0)^2 \alpha_{\text{base}}^2} \;=:\; \gamma_{\text{frac}} \;<\; \frac{1}{1 - \alpha_{\text{base}}^2}. \tag{116}$$

Comparing (115) with the unregularized bound $[1/(1 - \alpha_{\text{base}}^2)]\text{tr}(\Sigma_\xi)$, we obtain the conclusion that the variance bound is *strictly smaller than a fixed fraction* $\gamma_{\text{frac}} < 1$ of the baseline. This formalizes the theorem statement that "when edges cover the dominant direction, the bound is pushed strictly below a fixed fraction of the unregularized case."

**Consistency with Related Content in the Manuscript.**

- *Source and structure:* The squared-hinge structure of the isotonic penalty $\mathcal{L}_{\text{iso}}$ provides stable differentiable gradients on violations, flat regions on satisfied edges, and its strictness is controlled by $\mu$ (cf. Eq. 15 and its explanation).

- *Connection to Bellman residual:* Theorem 3.6 gives the expected Bellman error reduction after one update (Eq. 32); the subsequent paragraph explicitly states that "fixing violations directly reduces Bellman error; $\|\hat{V} - V_\theta\|^2$ is the cost of not straying too far from the network; with sufficiently large $\mu$ this cost is controlled, so the net effect is residual reduction," which *complements* Theorem 3.7: there, the *temporal variance* is controlled.

- *Oscillation intuition:* The manuscript explains under Theorem 3.7 that "cycle-cutting removes feedback loops; isotonic constraints turn the remaining edges into a ramp that prevents pull-back; increasing $\lambda_{\text{iso}}$ makes the ramp steeper, so oscillations are smaller," consistent with the spectral contraction–steady variance derivation in (113)–(115).

- *Training guidance:* Section B explicitly recommends "starting with small $\lambda_{\text{iso}}$ and gradually increasing $\lambda_{\text{iso}}$ and penalty weight $\mu$ as TD noise decreases," consistent with our conclusion that increasing $\lambda_{\text{iso}}$ monotonically improves the bound.

DAG-ification guarantees a feasible isotonic cone, and isotonic projection/penalty introduces both *tangent-space projection* and *extra curvature along constrained directions* under linearization, so that the spectral norm bound $\alpha(\lambda_{\mathrm{iso}}, \mathrm{cov})$ decreases monotonically with $\lambda_{\mathrm{iso}}$ and coverage; the steady-state MSE bound of the linear stochastic recursion therefore decreases monotonically; and when edges cover the dominant direction, we obtain a strict fixed-ratio improvement. The theorem is proved.

### G.8. Proof of Theorem 3.8

Fix a test-state distribution $\mathcal{D}_s$ and write the test error as

$$\mathbb{E}\big[(\hat{V} - V^*)^2\big] = \mathbb{E}_{s \sim \mathcal{D}_s} \mathbb{E}_{\mathsf{Alg}}\Big[\big(\hat{V}(s) - V^*(s)\big)^2\Big],$$

where $\mathbb{E}_{\mathsf{Alg}}$ denotes the expectation over *algorithmic randomness* (training, sampling, initialization, etc.). For notational simplicity below, we omit the explicit dependence on $s$ and regard all quantities as scalar random variables. Theorem 3.8 of the manuscript, under this scalarized convention, presents the decomposition identity and qualitative conclusions to be established: the equivariant regularizer $\mathcal{L}_{\mathrm{eq}}$ and the isotonic regularizer $\mathcal{L}_{\mathrm{iso}}$ both reduce variance while introducing a controllable *structural bias*

Let $m := \mathbb{E}[\hat{V}]$. For any constant $c$ we have

$$\hat{V} - V^* \;=\; (\hat{V} - c) + (c - V^*), \quad (\hat{V} - V^*)^2 \;=\; (\hat{V} - c)^2 + (c - V^*)^2 + 2(\hat{V} - c)(c - V^*).$$

Taking expectations on both sides and setting $c = m = \mathbb{E}[\hat{V}]$ (note that in this case $\mathbb{E}[\hat{V} - c] = 0$), we obtain

$$\mathbb{E}\big[(\hat{V} - V^*)^2\big] = \underbrace{\mathbb{E}\big[(\hat{V} - \mathbb{E}[\hat{V}])^2\big]}_{\mathrm{Var}[\hat{V}]} + \underbrace{\big(\mathbb{E}[\hat{V}] - V^*\big)^2}_{\mathrm{Bias}^2}. \tag{117}$$

This is precisely the *exact identity* "test error = variance + squared bias" stated in Theorem 3.8 of the manuscript.

*Modeling the situation:* Let $\mathsf{Sym}$ denote the event that a state lies in a "near-symmetric pocket," with coverage probability $\Pr(\mathsf{Sym}) = \rho$. When $\mathsf{Sym}$ occurs, by performing alignment through "representation transform + action relabeling" (see Section 3.1), a single sample yields $K$ *view-consistent* observations (or supervisions), which are averaged during training. Outside symmetric pockets (probability $1 - \rho$), no such averaging occurs. Theorem 3.2 in the manuscript explicitly states that this corresponds to an *effective sample size* amplified to $1 + (K - 1)\rho$, which directly reduces the variance upper bound by the same factor.

*Formal bound:* Using the law of total variance, we decompose

$$\mathrm{Var}[\hat{V}] = \mathbb{E}\big[\mathrm{Var}[\hat{V} \mid \mathsf{Sym}]\big] + \mathrm{Var}\big(\mathbb{E}[\hat{V} \mid \mathsf{Sym}]\big).$$

The second term corresponds to the "mean drift across pockets." By forcing different views to be *aligned*, $\mathcal{L}_{\mathrm{eq}}$ reduces this drift, hence this term decreases (see also the explanation following Thm. 3.8 in the manuscript, emphasizing that $\mathcal{L}_{\mathrm{eq}}$ averages symmetric views and contracts variance). For the first term:

$$\mathrm{Var}[\hat{V} \mid \mathsf{Sym}] = \mathrm{Var}\Big[\frac{1}{K} \sum_{k=1}^{K} \hat{V}^{(k)} \,\Big|\, \mathsf{Sym}\Big] = \frac{1}{K^2} \mathbf{1}^\top \mathrm{Cov}\big(\hat{V}^{(1:K)} \mid \mathsf{Sym}\big) \mathbf{1}.$$

If we denote by $\bar{\rho}_{\mathrm{c}} \in [0, 1]$ the average correlation coefficient between views in a symmetric pocket, and assume identical marginal variance $\sigma^2_{\mathrm{sym}}$ across views, we obtain the classical bound

$$\mathrm{Var}[\hat{V} \mid \mathsf{Sym}] \;\leq\; \sigma^2_{\mathrm{sym}} \cdot \frac{1 + (K - 1)\bar{\rho}_{\mathrm{c}}}{K}.$$

When the symmetry mechanism enforces strong alignment across views, $\bar{\rho}_{\mathrm{c}}$ is typically small; even conservatively setting $\bar{\rho}_{\mathrm{c}} \leq 1$, the reduction is still of order $O(1/K)$. Combining the contributions from symmetric pockets (weight $\rho$) and non-symmetric ones (weight $1 - \rho$), the overall variance bound becomes

$$\mathrm{Var}[\hat{V}] \;\leq\; (1 - \rho)\,\sigma^2_{\mathrm{base}} \;+\; \rho \cdot \sigma^2_{\mathrm{sym}} \cdot \frac{1 + (K - 1)\bar{\rho}_{\mathrm{c}}}{K}. \tag{118}$$

Together with Theorem 3.2's claim of an *effective sample size* $\simeq 1 + (K-1)\rho$ and the statement that the variance upper bound shrinks accordingly, (118) matches the conclusion of Thm. 3.8 that $\mathcal{L}_{\text{eq}}$ achieves variance reduction.

The hard limit of $\mathcal{L}_{\text{iso}}$ is to *project $V$* onto the closed convex cone $\mathcal{K}(D, \delta)$ defined by DAG monotonic inequalities (Theorem 3.4 in the manuscript), while the squared hinge is its smooth penalty approximation. Let $T := \Pi_{\mathcal{K}}$ be the Euclidean projection map. It is well known that $T$ is 1-Lipschitz (nonexpansive) and even *firmly nonexpansive*:

$$\|T(x) - T(y)\|_2 \leq \|x - y\|_2, \qquad \|T(x) - T(y)\|_2^2 \leq \langle T(x) - T(y), \, x - y \rangle.$$

For any random variable $X$, let $m := \mathbb{E}[X]$. Then

$$\text{Var}[T(X)] \;=\; \mathbb{E}\|T(X) - \mathbb{E}[T(X)]\|_2^2 \;\leq\; \mathbb{E}\|T(X) - T(m)\|_2^2 \;\leq\; \mathbb{E}\|X - m\|_2^2 \;=\; \text{Var}[X],$$

where the first step uses the fact that the mean is the unique minimizer of quadratic risk, and the second step follows from 1-Lipschitzness. Therefore, *projecting noisy estimates onto the feasible shape set never increases variance*. This aligns with Theorem 3.7: "projection induces spectral contraction and oscillation suppression, complementing equivariant sharing to achieve double variance reduction."

The bias introduced by the two regularizers comes from *structural mismatch*: (i) equivariant consistency restricts feasible solutions to the "near-group-equivariant" subclass; (ii) isotonicity restricts $V$ to the DAG monotone cone $\mathcal{K}(D, \delta)$. If the ground truth $V^*$ does not lie in the structural set $\mathcal{S} := \{\text{near-equivariant}\} \cap \mathcal{K}(D, \delta)$, then even in the absence of noise, the best achievable error is the *structural irreducible error* $\text{dist}(V^*, \mathcal{S})$. Formally, letting $S^\star := \Pi_{\mathcal{S}}(V^*)$,

$$\left\|\mathbb{E}[\hat{V}] - V^*\right\| \;\leq\; \underbrace{\left\|\mathbb{E}[\hat{V}] - S^\star\right\|}_{\text{optimization/estimation error}} \;+\; \underbrace{\left\|S^\star - V^*\right\|}_{\text{structural bias}=\text{dist}(V^*, \mathcal{S})} . \tag{119}$$

When (a) the equivariant group stabilizes under "projection–closure–alignment" (see Section 4.3 for the geometric realization), and (b) the DAG edges cover the dominant partial order, the first term can be suppressed by training; the second term depends solely on the mismatch between true structure and imposed structure, hence is *controllable*. Theorem 3.8 and Theorem D.5 explicitly state that "any bias is *structural*, proportional to the mismatch; when assumptions approximately match reality, this bias is *far smaller* than the variance reduction gain; and it can be further reduced via closure–alignment and confidence gating."

From (117), we have a *strict identity*; Steps 2–4 then provide: (i) $\mathcal{L}_{\text{eq}}$ reduces variance via "$K$-view averaging + pocket coverage" (Theorem 3.2: effective sample size $1 + (K-1)\rho$, and Theorem D.5's statistical assertion); (ii) $\mathcal{L}_{\text{iso}}$ reduces variance via "nonexpansiveness of convex projection" (consistent with Theorem D.4's spectral-contraction guarantee); (iii) both introduce only *structural bias* as in (186), controllable by the degree of structural mismatch (Theorem 3.8 / Theorem D.5).

Therefore, the theorem's statement holds: $\mathcal{L}_{\text{eq}} + \mathcal{L}_{\text{iso}}$ reduce statistical variance, while introducing only controllable structural bias determined by the mismatch between true symmetry/partial order and the imposed structure.

### G.9. Proof of Theorem 3.9

We prove: If there exists a true equivariant pair $(T^*, \Pi^*)$ in the environment, and the representation–value function family $f_\theta$ is sufficiently expressive (able to realize the equivariant structure), then when the "group-sample" regularizer (Section 3.1) is strong enough, the learned $\{(W_k, \Pi_k)\}_{k=1}^K$ converges, up to isomorphism, to a finitely generated approximation of $(T^*, \Pi^*)$. The corresponding statement in the paper is Theorem 3.9 (with the textual explanation pointing out: the group-sample regularizer shrinks the search space to "near-group" transformations, and the equivariant consistency loss $\mathcal{L}_{\text{eq}}$ selects the part that commutes with the environment dynamics; with sufficient capacity, the limit matches the *correct* equivariant structure, at most differing by relabeling/isomorphism).

We will make this description rigorous in four steps: *existence of limit points*, *the limit structure is a finite group representation*, *commutativity/uniqueness with the true equivariance*, and *isomorphism–relabeling–conjugacy characterization*.

**Training Objective and Regularizers.** Let the batch size be $B$ and the data distribution be $\mathcal{D}$. Following the construction in Section 3.1, we learn $K$ pairs of "representation linear transformations $W_k \in O(d)$ and action relabelings $\Pi_k$ (Sinkhorn nearly doubly stochastic, approaching permutations)":

$$T_k z = W_k z, \qquad \Pi_k \in \mathbb{R}^{|\mathcal{A}| \times |\mathcal{A}|} \text{ (Sinkhorn nearly doubly stochastic; nearest permutation is taken at inference),}$$

whose equivariant consistency loss (vector form) is (Eq. 9 in the paper):

$$\mathcal{L}_{\text{eq}} = \frac{1}{KB} \sum_{k=1}^{K} \sum_{i=1}^{B} \left\| Q_\theta(s_i, \cdot) - \Pi_k^\top Q_\theta(T_k s_i, \cdot) \right\|_2^2. \tag{120}$$

This loss aligns "transform-then-evaluate" with "evaluate-then-relabel," thereby enabling sample sharing inside "near-symmetry pockets" and reducing variance (see the original discussion in Section 3.1).

To make $(W_k, \Pi_k)$ "near-group," we add group-sample regularizers (Eq. 11 in the paper):

$$
\begin{aligned}
\mathcal{R}_{\text{id}} &= \|W_1 - I\|_F^2 + \|\Pi_1 - I\|_F^2 + \alpha \|W_k^\top W_k - I\|_F^2, \\
\mathcal{R}_{\text{clo}} &= \mathbb{E}_{i,j} \min_l \left( \|W_i W_j - W_l\|_F^2 + \|\Pi_i \Pi_j - \Pi_l\|_F^2 \right), \\
\mathcal{R}_{\text{inv}} &= \mathbb{E}_i \min_l \left( \|W_i^\top - W_l\|_F^2 + \|\Pi_i^\top - \Pi_l\|_F^2 \right), \\
\mathcal{R}_{\text{ord}} &= \frac{1}{K} \sum_{i=1}^{K} \sum_{m \in \mathcal{M}} \left( \|W_i^m - I\|_F^2 + \|\Pi_i^m - I\|_F^2 \right),
\end{aligned}
\tag{121}
$$

where $\mathcal{M} \subset \{2, 3, 4, \ldots\}$ is a finite set of orders. On the action side, an additional $\mathcal{R}_{\text{perm}}$ makes $\Pi_k$ approach permutation vertices; on the state side, the Cayley transform constrains $W_k$ in $O(d)$ to ensure numerical stability and norm preservation (see Section 3.1).

The total objective (ignoring terms irrelevant to this theorem) can be written as

$$\mathcal{L}_{\text{tot}}(\theta, W, \Pi) = \underbrace{\mathcal{L}_{\text{RL}}(\theta)}_{\text{TD/Double-DQN}} + \lambda_{\text{eq}} \mathcal{L}_{\text{eq}}(\theta, W, \Pi) + \gamma_{\text{grp}} \left( \mathcal{R}_{\text{id}} + \mathcal{R}_{\text{clo}} + \mathcal{R}_{\text{inv}} + \mathcal{R}_{\text{ord}} + \mathcal{R}_{\text{perm}} \right),$$

where $\lambda_{\text{eq}} > 0$, and $\gamma_{\text{grp}} \gg 1$ indicates that the "group-sample" regularizer is *sufficiently strong*.

**Assumption (True Equivariance and Uniqueness of the Optimal Value).** Suppose the MDP is invariant under a symmetry cluster $\mathcal{G}$ (the premise of Theorem 3.1), and the environment admits a true equivariant pair $(T^*, \Pi^*)$. The optimal Bellman operator $\mathcal{T}^*$ commutes with $g \in \mathcal{G}$, and $Q^*$ is the unique fixed point, which is also an equivariant fixed point (the explanation of Theorem 3.1 in the paper):

$$\mathcal{T}^*(Q \circ g) = (\mathcal{T}^* Q) \circ g, \qquad Q^*(T^* s, \Pi^* a) = Q^*(s, a) \quad (\forall (s, a)), \tag{122}$$

"uniqueness + commutativity" pins down the "correct $Q^*$" as the equivariant structure (as explained in Theorem 3.1 of the paper).

We now give four lemmas and then complete the proof.

**Lemma G.13** (Compactness and Existence of Limit Points). *Consider a sequence of near-optimal solutions $(\theta_n, W^{(n)}, \Pi^{(n)})$ satisfying*

$$\mathcal{L}_{\text{tot}}(\theta_n, W^{(n)}, \Pi^{(n)}) \leq \inf_{\theta, W, \Pi} \mathcal{L}_{\text{tot}} + o(1), \qquad \gamma_{\text{grp}} \to \infty, \ \lambda_{\text{eq}} > 0 \text{ fixed}.$$

*Then there exists a subsequence converging to*

$$W_k^{(n)} \to W_k^{(\infty)} \in O(d), \qquad \Pi_k^{(n)} \to \Pi_k^{(\infty)} \in \mathcal{P}_{|\mathcal{A}|},$$

*and the limit set $\{(W_k^{(\infty)}, \Pi_k^{(\infty)})\}_{k=1}^K$ induces a finite group structure with identity element $(W_1^{(\infty)}, \Pi_1^{(\infty)}) = (I, I)$.*

*Proof of Lemma G.13.* By the compactness of the Cayley-constrained matrices $W_k^{(n)} \in O(d)$ and the compactness of the permutation polytope (Birkhoff polytope), together with the practice of projecting Sinkhorn iterates to the nearest permutation at inference, one can extract a subsequence such that

$$W_k^{(n)} \to W_k^{(\infty)} \in O(d), \qquad \Pi_k^{(n)} \to \Pi_k^{(\infty)} \in \mathcal{P}_{|\mathcal{A}|}.$$

Since $\gamma_{\text{grp}} \to \infty$ and $(\theta_n, W^{(n)}, \Pi^{(n)})$ are near-optimal, it must hold that

$$\mathcal{R}_{\text{id}}(W^{(n)}, \Pi^{(n)}) \to 0, \ \mathcal{R}_{\text{clo}}(W^{(n)}, \Pi^{(n)}) \to 0, \ \mathcal{R}_{\text{inv}}(W^{(n)}, \Pi^{(n)}) \to 0, \ \mathcal{R}_{\text{ord}}(W^{(n)}, \Pi^{(n)}) \to 0.$$

Therefore, in the limit we obtain

$$W_1^{(\infty)} = I, \ \Pi_1^{(\infty)} = I; \qquad \exists l : \ W_i^{(\infty)} W_j^{(\infty)} = W_l^{(\infty)}, \ \Pi_i^{(\infty)} \Pi_j^{(\infty)} = \Pi_l^{(\infty)}, \tag{123}$$

$$\exists l : \ W_i^{(\infty)\top} = W_l^{(\infty)}, \ \Pi_i^{(\infty)\top} = \Pi_l^{(\infty)}; \qquad \exists m \in \mathcal{M} : \ W_i^{(\infty)m} = I, \ \Pi_i^{(\infty)m} = I. \tag{124}$$

Equations (123)–(124) show that $\{(W_k^{(\infty)}, \Pi_k^{(\infty)})\}_{k=1}^K$ is closed under multiplication, admits inverses, and has finite order. Hence the limit forms a finite group with identity element $(I, I)$. $\qquad\square$

**Lemma G.14** (Limit structure as a finite group representation). *Define a multiplication on the index set $\{1, \ldots, K\}$ by*

$$i \circ j := l \ \ \text{iff} \ \ W_i^{(\infty)} W_j^{(\infty)} = W_l^{(\infty)} \quad (\text{equivalently } \Pi_i^{(\infty)} \Pi_j^{(\infty)} = \Pi_l^{(\infty)}).$$

*Then $(\{1, \ldots, K\}, \circ)$ forms a finite group $H$, and the mapping*

$$\rho : H \to O(d) \times \mathcal{P}_{|\mathcal{A}|}, \qquad \rho(h_k) = (W_k^{(\infty)}, \Pi_k^{(\infty)}),$$

*is a finite-dimensional group representation.*

*Proof of Lemma G.14.* By (123)–(124), the set $\{1, \ldots, K\}$ with operation $\circ$ admits an identity, inverses, and has finite order. Since the limit is taken in operator norm, the equalities are exact. Because multiplication in $O(d)$ and in the permutation group is associative and limits preserve algebraic identities, $\circ$ is associative. Thus $(\{1, \ldots, K\}, \circ)$ is a finite group $H$. The mapping $\rho$ preserves multiplication, hence is a representation of $H$. $\qquad\square$

**Lemma G.15** (Equivariance consistency enforces commutativity). *Let $(\theta_\infty, W^{(\infty)}, \Pi^{(\infty)})$ be a limit point with $\lambda_{\text{eq}} > 0$. Then the induced representation $\rho(H)$ makes $Q_\theta$ equivariant and commutative with the environment dynamics.*

*Proof of Lemma G.15.* Define the population version of the equivariance loss:

$$\mathcal{L}_{\text{eq}}^{\text{pop}}(\theta, W, \Pi) := \frac{1}{K} \sum_{k=1}^K \mathbb{E}_{s \sim \mathbb{P}}\left[ \|Q_\theta(s, \cdot) - \Pi_k^\top Q_\theta(T_k s, \cdot)\|_2^2 \right]. \tag{125}$$

By uniform convergence (Theorem G.17), minimizers of empirical risk converge to minimizers of (125). If $f_\theta$ is sufficiently expressive, the minimum of (125) is attained only when

$$Q_\theta(s, \cdot) = \Pi_k^\top Q_\theta(T_k s, \cdot) \quad \text{for a.e. } s, \text{ all } k. \tag{126}$$

Hence $Q_\theta$ is equivariant under $\rho(H)$. On the other hand, the environment symmetry $(T^*, \Pi^*)$ commutes with $\mathcal{T}^*$, and $Q^*$ is its unique fixed point (Theorem 3.1). If $\mathcal{L}_{\text{RL}}$ is minimized, then $Q_\theta = Q^*$ up to isomorphism, inheriting equivariance. Therefore the limit representation $\rho(H)$ must commute with the environment dynamics, i.e., is a finitely generated approximation of the true symmetry. $\qquad\square$

**Lemma G.16** (Isomorphism–relabeling–conjugacy characterization). *The limit representation $\rho(H)$ is equivalent to the true symmetry representation $\rho^*$ induced by $(T^*, \Pi^*)$. There exists an orthogonal change of basis $S \in O(d)$ and a relabeling $\varphi : H \to \langle(T^*, \Pi^*)\rangle$ such that*

$$\forall h \in H : \qquad W^{(\infty)}(h) = S^{-1} W^*(\varphi(h)) S, \qquad \Pi^{(\infty)}(h) = \Pi^*(\varphi(h)). \tag{127}$$

*Proof of Lemma G.16.* By Lemma G.14, $\rho(H)$ is a finite group representation. By Lemma G.15, it commutes with environment dynamics, hence aligns with $Q^*$. Equivalence of finite group representations means there exists a conjugacy on the state side and relabeling on the generator set. Concretely, one finds $S \in O(d)$ and a relabeling $\varphi$ such that (127) holds. Note: on the action side, $\Pi$ is constrained to permutations with $\Pi_1 = I$, so no conjugacy is allowed there. On the representation side, conjugacy $S$ corresponds to change of basis in feature space. Thus $\rho(H)$ and $\rho^*$ differ only by isomorphism/relabeling/conjugacy. $\qquad\square$

By Lemmas G.13–G.14, any nearly optimal sequence under $\gamma_{\text{grp}} \to \infty$ admits a convergent subsequence, whose limit induces a representation $\rho(H)$ of a finite group $H$. By Lemma G.15, the limiting representation $\rho(H)$ commutes with the environment dynamics and enforces $Q_\theta$ to be equivariant and aligned with $Q^*$ (cf. Theorem 3.1, the mechanism of the "unique equivariant fixed point"). By Theorem G.16, $\rho(H)$ is *equivalent* to the true equivariant representation $\rho^*$ induced by $(T^*, \Pi^*)$, in the sense that there exists a relabeling of generators $\varphi$ and a conjugacy by some $S \in O(d)$ such that (127) holds. Therefore, when the group-sample regularization is sufficiently strong, $f_\theta$ has enough expressive power, and empirical-to-population consistency holds (cf. Theorem 3.10 on capacity and uniform convergence), the learned $\{(W_k, \Pi_k)\}$ *converges* to a *finite generated approximation* of $(T^*, \Pi^*)$, and is *unique up to isomorphism, relabeling, and representation conjugacy*—which is precisely the strict meaning of the theorem's claim on *Identifiability*.

## G.10. Proof of Theorem 3.10

In a $\mathcal{G}$-invariant MDP (Theorem 2.1), assume that $Q_\theta$ has sufficient expressive power and that $\{(W_k, \Pi_k)\}_{k=1}^K$ can approximate some finitely generated subgroup of $\mathcal{G}$; the data are i.i.d. (or uniformly mixing) off-policy samples; and the one-step "push" $\Delta_i$ satisfies Massart-type noise and margin conditions. Training minimizes the total objective (Eq. 30), and we take the isotonic penalty $\mu \to \infty$ (Eq. 15). We need to show that there exists a capacity term $\mathfrak{R}(N)$ and a constant $C > 0$ such that for $N$ samples,

$$\Pr\Big( \underbrace{\max_{g \in \mathcal{G}} \|Q_\theta(T_g s, \Pi_g a) - Q_\theta(s,a)\|_2}_{\text{equivariance residual}} + \underbrace{\frac{1}{|E(D)|} \sum_{(u \to v) \in D} \big[\delta + \hat{V}(v) - \hat{V}(u)\big]_+}_{\text{partial-order violation}} \leq C\sqrt{\frac{\mathfrak{R}(N)}{N}}\Big) \xrightarrow[N \to \infty]{} 1, \quad (128)$$

and further deduce that $\{(W_k, \Pi_k)\}$ (up to isomorphism/relabeling) converges to the true equivariances, while the greedy DAG edges (Section 3.2) concentrate, with vanishing probability error, on an acyclic subset of the true partial order. This theorem is stated in the paper as Theorem 3.10 (with the exact form of assumptions and conclusion), together with a reasoning summary (capacity and uniform convergence, Massart noise for sign recovery, greedy DAG converging to the true partial-order subset, and group-sample regularization with equivariance loss selecting transformations commuting with the dynamics).

**Proof Structure.** We divide the argument into four steps: (*i*) define restricted function classes and establish population–empirical uniform convergence (via Rademacher complexity bounds); (*ii*) use the penalty consistency as $\mu \to \infty$ to reduce the isotonic surrogate to the hard isotonic projection, thereby controlling the "partial-order violation"; (*iii*) Massart noise and margin assumptions ensure that the signs of $\Delta_i$ can be consistently recovered, which together with greedy acyclicity removal (Section 3.2) concentrates the greedy DAG on an acyclic subset of the true partial order; (*iv*) group-sample regularization forces $(W_k, \Pi_k)$ to contract to a "near-group," while the equivariance loss selects those commuting with the dynamics; under statistical consistency this drives the equivariance residual to zero and (by Theorem 3.9) identifies the true equivariances up to isomorphism/relabeling. Finally, combining the bounds yields (128).

Assume the discounted return is bounded: $\|Q_\theta(s, \cdot)\|_\infty \leq B_Q$ (e.g. $B_Q = R_{\max}/(1-\gamma)$). We define two types of sample losses:

$$\ell_{\text{eq}}(s; \theta, W, \Pi) := \frac{1}{K} \sum_{k=1}^K \big\|Q_\theta(s, \cdot) - \Pi_k^\top Q_\theta(T_k s, \cdot)\big\|_2^2,$$

$$\ell_{\text{iso}}(\mathcal{B}; \theta, \hat{V}) := \frac{1}{|E(D)|} \sum_{(u \to v) \in D} \big[\delta + \hat{V}(v) - \hat{V}(u)\big]_+^2 + \frac{1}{B} \sum_{i=1}^B \big(\hat{V}(i) - V_\theta(i)\big)^2,$$

where $\hat{V}$ comes from the isotonic subproblem (Eq. 15; see Step II), and $V_\theta(i) = \max_a Q_\theta(s_i, a)$. Under the group-sample regularization constraints (Eq. 11), we introduce the restricted classes

$$\mathcal{F}_{\text{eq}}(\varepsilon) := \Big\{(\theta, W, \Pi) : \mathbb{E}\,\ell_{\text{eq}} \leq \varepsilon, \ (W_k, \Pi_k) \text{ satisfy group-sample soft constraints of radius } \leq r\Big\},$$

$$\mathcal{F}_{\text{iso}}(\mu) := \Big\{(\theta, \hat{V}) : \text{minimizers/near-minimizers of the isotonic penalty (Eq. 15) with weight } \mu\Big\}.$$

We now state the unified generalization lemma, where the capacity term is denoted $\mathfrak{R}(N)$ (the Rademacher complexity of the joint class; see the capacity term in Theorem 3.10 of the paper).

**Lemma G.17** (Uniform Convergence for Restricted Classes). *Suppose $\ell_{\mathrm{eq}}$ and $\ell_{\mathrm{iso}}$ are both L-Lipschitz in $Q_\theta$ and $(\hat{V}, V_\theta)$, respectively, and are bounded (controlled by $B_Q$). Then there exists a constant $c > 0$ such that for any $\delta \in (0, 1)$, with probability at least $1 - \delta$,*

$$\sup_{(\theta, W, \Pi) \in \mathcal{F}_{\mathrm{eq}}(\varepsilon)} \left| \mathbb{E}\, \ell_{\mathrm{eq}} - \frac{1}{N} \sum_{i=1}^{N} \ell_{\mathrm{eq}}(s_i) \right| \leq c \sqrt{\frac{\mathfrak{R}(N) + \log(1/\delta)}{N}}, \tag{129}$$

$$\sup_{(\theta, \hat{V}) \in \mathcal{F}_{\mathrm{iso}}(\mu)} \left| \mathbb{E}\, \ell_{\mathrm{iso}} - \frac{1}{N} \sum_{i=1}^{N} \ell_{\mathrm{iso}}(\mathcal{B}_i) \right| \leq c \sqrt{\frac{\mathfrak{R}(N) + \log(1/\delta)}{N}}. \tag{130}$$

*Proof of Lemma G.17.* Standard symmetrization and Talagrand-type contraction inequalities yield Rademacher bounds for squared-norm (or squared hinge) losses; boundedness of the losses follows from $B_Q$ and the clipping parameter $\delta$. The complexity of the restricted classes $\mathcal{F}_{\mathrm{eq}}(\varepsilon)$ and $\mathcal{F}_{\mathrm{iso}}(\mu)$ is absorbed into the term $\mathfrak{R}(N)$. $\square$

Consider the isotonic problem on the batch DAG $D$ (Eq. 15):

$$\hat{V}^\star \in \arg\min_{\hat{V}} \frac{1}{B} \sum_{i=1}^{B} \left( \hat{V}(i) - V_\theta(i) \right)^2 + \mu \cdot \frac{1}{|E(D)|} \sum_{(u \to v) \in D} \left[ \delta + \hat{V}(v) - \hat{V}(u) \right]_+^2. \tag{131}$$

Let $\mathrm{viol}(\hat{V}; D) = \frac{1}{|E(D)|} \sum_{(u \to v) \in D} [\delta + \hat{V}(v) - \hat{V}(u)]_+$ denote the average violation. By the standard penalty method (the paper's explanation of Eq. 15: "the squared hinge penalty asymptotically approximates the indicator of the feasible cone"), as $\mu \to \infty$, $\hat{V}^\star$ converges to the *exact* DAG isotonic projection, and the violation can be made arbitrarily small (see the discussion following Eq. 15).

**Lemma G.18** (Penalty Consistency). *Fix $D$ and finite $B$. Let $\hat{V}_\mu$ be a minimizer of (131). Then as $\mu \to \infty$, any cluster point $\hat{V}_\infty$ is a solution of the* exact isotonic projection*, and $\mathrm{viol}(\hat{V}_\mu; D) \to 0$.*

*Proof of Lemma G.18.* This follows from the standard quadratic penalty consistency argument. If there exists $\epsilon > 0$ such that $\mathrm{viol}(\hat{V}_\mu; D) \not\to 0$, then the quadratic penalty term would grow at least linearly in $\mu$, contradicting the optimality of $\hat{V}_\mu$ (relative to any feasible $\hat{V}$). Any cluster point must satisfy all edge constraints $\hat{V}(u) \geq \hat{V}(v) + \delta$, and is therefore a solution of the exact problem. $\square$

By Lemma G.18, when training schedules $\mu \uparrow \infty$ as described in the paper (Eq. 15 and total objective Eq. 30), the *partial-order violation* of $\hat{V}$ converges to 0 in expectation, and by Lemma G.17 this transfers from empirical to population error.

Assume there exists $\eta < \frac{1}{2}$ and a margin $\gamma_0 > 0$ such that

$$\Pr\left[ \mathrm{sign}(\Delta_i) \neq \mathrm{sign}(\Delta_i^\star) \mid s_i \right] \leq \eta, \qquad \left| \mathbb{E}[\Delta_i \mid s_i] \right| \geq \gamma_0.$$

This is the condition of Theorem 3.10 (Eq. 33). For each candidate edge $(u \to v)$ (from batch $k$NN and confidence-weight gating, Section 3.2), the sign test is a binary classification problem under Massart noise. By Bernstein/Hoeffding bounds and $\eta < 1/2$, for fixed $(u, v)$, as $N \to \infty$ the empirical sign $\widehat{\mathrm{sign}}(\Delta)$ converges exponentially fast to the true sign $\mathrm{sign}(\Delta^\star)$, with the margin $\gamma_0$ ensuring stability in local neighborhoods (avoiding "fragile ties"). Since the greedy strategy is "insert by weight order, discard on cycle" (Section 3.2), under asymptotically correct signs, all accepted edges remain acyclic, and the greedy DAG converges to an *acyclic subset* of the true partial order (as explicitly stated in Theorem 3.3). Combined with Lemma G.18 on isotonic projection (Section 3.3 ensures consistency with the true order once cycles are removed), the partial-order violation term

$$\underbrace{\frac{1}{|E(D)|} \sum_{(u \to v) \in D} \left[ \delta + \hat{V}(v) - \hat{V}(u) \right]_+}_{\text{second term in Eq. 34}}$$

converges to 0 with high probability as $N \to \infty$ (and by Lemma G.17 is bounded by $O(\sqrt{\mathfrak{R}(N)/N})$).

Group-sample regularization (Eq. 11) with $\gamma_{\mathrm{grp}} > 0$ forces $\{(W_k, \Pi_k)\}$ into a "near-group" set, while the equivariance loss (Eq. 9) enforces

$$Q_\theta(s, \cdot) \approx \Pi_k^\top Q_\theta(T_k s, \cdot) \quad (\forall k),$$

i.e. "transform-then-evaluate" equals "evaluate-then-permute." Thus at the population minimizer, $Q_\theta$ is equivariant w.r.t. $\{(W_k, \Pi_k)\}$. The paper explains that group-sample regularization collapses the search space to near-groups, while the equivariance loss "selects" the part commuting with the environment dynamics (see Theorem 3.9 discussion). By *uniform convergence* (Lemma G.17), the empirical equivariance residual

$$\max_{g \in \mathcal{G}} \|Q_\theta(T_g s, \Pi_g a) - Q_\theta(s, a)\|_2$$

transfers to a population bound with rate $O(\sqrt{\mathfrak{R}(N)/N})$ (as stated at the end of Theorem 3.10). At the same time, Theorem 3.9 (Identifiability) ensures that when $f_\theta$ is sufficiently expressive and group-sample regularization sufficiently strong, $\{(W_k, \Pi_k)\}$ converges (up to isomorphism/relabeling) to a finite generating approximation of the true equivariance $(T^\star, \Pi^\star)$; the equivariance loss selects those commuting with the dynamics, so the limit matches the "correct" symmetry (up to relabeling/conjugacy). Hence, as $N \to \infty$, the equivariance residual (the first term in Eq. 34) $\to 0$, and the transformations themselves are identifiable.

**Union Bound and Probabilistic Limit.**  Combining Step I (uniform convergence, Lemma G.17), Step II (penalty consistency, Lemma G.18), Step III (Massart noise recovery and greedy DAG convergence), and Step IV (equivariance residual convergence and identifiability), by a union bound we obtain: for any $\delta \in (0, 1)$, with probability at least $1 - \delta$,

$$\max_{g \in \mathcal{G}} \|Q_\theta(T_g s, \Pi_g a) - Q_\theta(s, a)\|_2 + \frac{1}{|E(D)|} \sum_{(u \to v) \in D} \left[\delta + \hat{V}(v) - \hat{V}(u)\right]_+ \leq C\sqrt{\frac{\mathfrak{R}(N) + \log(1/\delta)}{N}},$$

where $C > 0$ depends on Lipschitz constants, $B_Q$, and regularization radii. Taking $\delta = e^{-\mathfrak{R}(N)}$ and absorbing into constants, we have

$$\Pr\Big((128) \text{ holds}\Big) \geq 1 - \exp\big(-\Omega(\mathfrak{R}(N))\big) \xrightarrow[N \to \infty]{} 1.$$

This matches the explicit statement of Theorem 3.10 (Eq. 34): *both structural residuals* (equivariance residual + partial-order violation) shrink to 0 at rate $O(\sqrt{\mathfrak{R}(N)/N})$, and (by Theorem 3.9 and greedy DAG consistency) the transformations converge to the true equivariance (up to isomorphism), while the DAG edges concentrate on an acyclic subset of the true partial order.

### G.11. Proof of Theorem 4.1

Consider a batch $\mathcal{B} = \{s_1, \ldots, s_B\}$ and let $V \in \mathbb{R}^B$ with $V(i) = V_\theta(s_i)$. Given a directed edge set $\mathcal{E} \subset [B] \times [B]$ (obtained by the "greedy cycle-breaking" procedure, i.e., discarding an edge whenever a cycle would be formed), for each edge $e = (u \to v)$ define the inequality constraint

$$g_e(V) := \delta + V(v) - V(u) \leq 0 \qquad (\delta \geq 0), \tag{132}$$

and stack them into a vector $g(V) \in \mathbb{R}^{|\mathcal{E}|}$.

The equality constraints are given by

$$h(V) := \big(Q_\theta(T_g s, \Pi_g a) - Q_\theta(s, a)\big)_{g \in \mathcal{G}_{\mathrm{batch}}} = 0, \tag{133}$$

where $\mathcal{G}_{\mathrm{batch}}$ denotes the finite set of near-equivariant transformations applied on the current batch.

Finally, define the stacked constraints (with nonnegative slack variables) as

$$c(V, \mu) := \begin{bmatrix} g(V) + \mu \\ h(V) \end{bmatrix}, \qquad \mu \in \mathbb{R}_{\geq 0}^{|\mathcal{E}|}. \tag{134}$$

The theorem asserts two claims: (i) If $\mathcal{E}$ is obtained by the greedy "cycle-breaking" procedure and $\delta \geq 0$, then there exists $V$ such that $g(V) \leq 0$. (ii) If there exists some $(\theta, \{T_g, \Pi_g\})$ that exactly satisfies the equivariance equalities (133) on this batch, then there exists $(V, \mu)$ with $\mu \geq 0$ such that $c(V, \mu) = 0$.

**Part I: Feasibility of $g(V) \leq 0$ (DAG $\Rightarrow \delta$-potential).**

**Lemma G.19** (Greedy cycle-breaking $\Rightarrow$ Acyclicity). *If the edge set $\mathcal{E}$ is obtained by the rule "insert edges in descending order of confidence score, discarding any edge that would form a cycle," then $G = ([B], \mathcal{E})$ is a DAG (acyclic).*

*Proof of Lemma G.19.* Suppose, for contradiction, that the final edge set $\mathcal{E}$ contains a directed cycle $v_1 \to v_2 \to \cdots \to v_m \to v_1$. Let $v_j \to v_{j+1}$ (indices mod $m$) be the last edge added among this cycle. At the time this edge was considered, all other edges of the cycle were already in $\mathcal{E}$, so $v_{j+1}$ was reachable from $v_j$. Adding $v_j \to v_{j+1}$ would therefore have closed a cycle, contradicting the rule that discards cycle-forming edges. $\qquad\square$

Thus $G$ is a finite DAG, and hence admits a topological order $\tau : [B] \to \{1, \ldots, B\}$ (so that $u \to v$ implies $\tau(u) < \tau(v)$). Define the *longest path length* from node $i$:

$$L(i) := \max\{|p| : p \text{ is a directed path starting at } i, \ |p| = \text{number of edges}\} \in \{0, 1, \ldots, B - 1\}. \qquad (135)$$

Since $G$ is finite and acyclic, $L(i)$ is well-defined. For every edge $u \to v$, appending $u \to v$ in front of a longest path from $v$ shows that

$$L(u) \geq 1 + L(v) \quad \Longleftrightarrow \quad L(u) - L(v) \geq 1. \qquad (136)$$

**Lemma G.20** ($\delta$-level potential function). *Let $V(i) := \delta \cdot L(i)$. Then for every edge $u \to v$, one has $V(u) - V(v) \geq \delta$, equivalently $g_{(u \to v)}(V) = \delta + V(v) - V(u) \leq 0$.*

*Proof of Lemma G.20.* From (136),

$$V(u) - V(v) = \delta\big(L(u) - L(v)\big) \geq \delta,$$

equivalent to $\delta + V(v) - V(u) \leq 0$. For $\delta = 0$, any constant potential $V \equiv 0$ suffices. $\qquad\square$

Hence, by Lemma G.20, choosing $V_\star(i) = \delta L(i)$ yields $g(V_\star) \leq 0$, and the inequality system (132) is feasible. This completes the first part of the theorem.

**Part II: Existence of $(V, \mu)$ with $c(V, \mu) = 0$ and $\mu \geq 0$.**

Assume there exists a parameter–transformation pair $(\theta^\dagger, \{T_g^\dagger, \Pi_g^\dagger\})$ that exactly satisfies the equivariance equalities (133) on the current batch:

$$Q_{\theta^\dagger}(T_g^\dagger s, \Pi_g^\dagger a) - Q_{\theta^\dagger}(s, a) = 0, \qquad \forall g \in \mathcal{G}_{\text{batch}}, \ \forall (s, a) \text{ in the batch.} \qquad (137)$$

From Part I, take $V_\star$ constructed by (135)–(G.20), so that $g(V_\star) \leq 0$. Define the slack variables

$$\mu_\star := -g(V_\star) \in \mathbb{R}_{\geq 0}^{|\mathcal{E}|}. \qquad (138)$$

Then

$$g(V_\star) + \mu_\star = 0, \qquad h(V_\star) = 0 \ \text{ (by (137))}.$$

Stacking as in (134),

$$c(V_\star, \mu_\star) = \begin{bmatrix} 0 \\ 0 \end{bmatrix}, \qquad \mu_\star \geq 0.$$

Thus we have constructed a pair $(V, \mu)$ with $c(V, \mu) = 0$ and $\mu \geq 0$, proving the second part.

**Remarks (completeness and numerical stability).** (1) If $\mathcal{E}$ is empty, the construction degenerates to arbitrary $V$ and empty $\mu$, and only the equivariance constraint (137) remains. (2) The definition of $L(i)$ is equivalent to using a topological "layer index": $H(i) = \max\{\text{length of a path from } i \text{ to a sink}\}$, with $V(i) = \delta H(i)$. (3) For $\delta > 0$, the margin $V(u) - V(v) \geq \delta$ enforces a strict gap, improving numerical stability of the isotonic projection. (4) Equivariance feasibility is only *batch-wise*: we require (137) only for pairs $(s, a)$ and $g$ in the batch, without conflicting with global consistency.

**Conclusion.** Greedy cycle-breaking guarantees $\mathcal{E}$ is acyclic, so there exists a $\delta$–potential $V$ with $g(V) \leq 0$; if the equivariance equalities are realizable on the batch, then with $\mu = -g(V) \geq 0$ we obtain $(V, \mu)$ such that $c(V, \mu) = 0$.

## G.12. Proof of Theorem 4.2

On a given batch of samples, define

$$c(V, \mu) \; := \; \begin{bmatrix} g(V) + \mu \\ h(V) \end{bmatrix} \in \mathbb{R}^{m_1 + m_2}, \qquad J_c(V, \mu) \; := \; \begin{bmatrix} J_g(V) & I_{m_1} \\ J_h(V) & 0 \end{bmatrix},$$

where $g(V) = \big(\delta + V(v) - V(u)\big)_{(u \to v) \in \mathcal{E}}$ is the stack of monotonicity inequalities with dimension $m_1 = |\mathcal{E}|$, and $h(V)$ is the stack of equivariance equalities with dimension $m_2$. Here $J_g, J_h$ denote their Jacobians with respect to $V$. By definition,

$$J_V \; := \; \frac{\partial[g; h]}{\partial V} = \begin{bmatrix} J_g \\ J_h \end{bmatrix},$$

and hence

$$J_c = [\, J_V \;\; I_{m_1} \oplus 0_{m_2} \,].$$

We further define the *violation energy* as

$$\Phi(V, \mu) \; := \; \tfrac{1}{2} \, \|c(V, \mu)\|_2^2.$$

**Update and Direction Decomposition.** Given the current $(V, \mu)$, define

$$\Delta_{\mathrm{tan}} \; := \; B_u B_u^\top (-g_V), \qquad \Delta_{\mathrm{nor}} \; := \; -\lambda \, J_c^\dagger \, c(V, \mu),$$

where $g_V = \nabla_V \mathcal{L}_{\mathrm{TD}}$, $B_u \in \mathbb{R}^{B \times d_u}$ is an orthonormal basis of $\mathrm{Null}(J_V)$ ($B_u^\top B_u = I$ and $J_V B_u = 0$), and $J_c^\dagger$ denotes the Moore–Penrose pseudoinverse of $J_c$. The update is carried out in two substeps:

$$\text{(A)} \quad (V, \mu) \; \mapsto \; (V_A, \mu_A) := \Big(V + \eta\big(\Delta_{\mathrm{tan}} + \Delta_{\mathrm{nor}}^{(V)}\big), \; \mu + \eta \, \Delta_{\mathrm{nor}}^{(\mu)}\Big), \tag{139}$$

$$\text{(B)} \quad (V_A, \mu_A) \; \mapsto \; (V^+, \mu^+) := \Big(V_A, \; [\, \mu_A - \eta_\mu \big(g(V_A) + \mu_A\big) \,]_+\Big), \tag{140}$$

where $\Delta_{\mathrm{nor}}^{(V)}$ and $\Delta_{\mathrm{nor}}^{(\mu)}$ denote the $V$- and $\mu$-components of $\Delta_{\mathrm{nor}}$, respectively, and $[\cdot]_+$ is the coordinate-wise projection onto $\mathbb{R}_{\geq 0}$. The constants $\eta, \eta_\mu, \lambda > 0$ are step sizes (contraction coefficients).

**Part I: Tangential safety $J_V \Delta_{\mathrm{tan}} = 0$.** By construction, $\Delta_{\mathrm{tan}} \in \mathrm{Null}(J_V)$, i.e. $J_V \Delta_{\mathrm{tan}} = 0$. Therefore, the first-order perturbation along $\Delta_{\mathrm{tan}}$ does not affect the linearization of any *equality* or *inequality* constraints with respect to $V$ (both $g$ and $h$ have zero first-order change). This is exactly the meaning of "tangential safety." Hence the first statement of the theorem holds.

**Part II: Normal contraction leads to violation energy descent (linearization and quadratic remainder).** For $\Phi = \tfrac{1}{2}\|c\|^2$, we apply a first–second order expansion. Let $L > 0$ be a local Lipschitz constant of the Jacobian $J_c$, i.e.

$$\| c(x + \Delta) - c(x) - J_c(x)\Delta \| \; \leq \; \tfrac{L}{2} \, \|\Delta\|^2 \qquad \big(x = (V, \mu), \; \Delta = (\Delta_V, \Delta_\mu)\big).$$

(In a finite batch and bounded neighborhood, $J_g, J_h$ are continuously differentiable, hence such an $L$ exists.) Then the standard *Descent Lemma* yields

$$\Phi(x + \Delta) \; \leq \; \Phi(x) \; + \; \underbrace{\big\langle J_c(x)^\top c(x), \, \Delta \big\rangle}_{\text{first-order term}} + \; \tfrac{L}{2} \, \|\Delta\|^2. \tag{141}$$

We first analyze the *non-projected* change in substep (A), i.e. $\Delta_A := (\eta \Delta_{\mathrm{tan}} + \eta \Delta_{\mathrm{nor}}^{(V)}, \; \eta \Delta_{\mathrm{nor}}^{(\mu)})$. Observe that

$$\big\langle J_c^\top c, \, \Delta_A \big\rangle \; = \; \eta \underbrace{\big\langle J_c^\top c, \, (\Delta_{\mathrm{tan}}, 0) \big\rangle}_{\text{(i)}} + \eta \underbrace{\big\langle J_c^\top c, \, \Delta_{\mathrm{nor}} \big\rangle}_{\text{(ii)}}.$$

For (i), we have

$$\big\langle J_c^\top c, \, (\Delta_{\mathrm{tan}}, 0) \big\rangle \; = \; \big\langle c, \, J_c(\Delta_{\mathrm{tan}}, 0) \big\rangle \; = \; \Big\langle c, \, \begin{bmatrix} J_V \Delta_{\mathrm{tan}} \\ 0 \end{bmatrix} \Big\rangle \; = \; 0,$$

since $J_V \Delta_{\tan} = 0$.

For (ii), using $\Delta_{\text{nor}} = -\lambda J_c^\dagger c$ and the self-adjoint idempotence of the projection $P := J_c J_c^\dagger$, we obtain

$$\langle J_c^\top c, \, \Delta_{\text{nor}} \rangle = -\lambda \langle J_c^\top c, \, J_c^\dagger c \rangle = -\lambda \langle c, \, J_c J_c^\dagger c \rangle$$
$$= -\lambda \langle c, \, Pc \rangle = -\lambda \|Pc\|_2^2. \tag{142}$$

On the other hand, $\|\Delta_A\|^2 \le 2\eta^2 \|\Delta_{\tan}\|^2 + 2\eta^2 \|\Delta_{\text{nor}}\|^2$, and $\|\Delta_{\text{nor}}\| = \lambda \|J_c^\dagger c\|$. By the lower bound of the smallest nonzero singular value $\sigma_{\min} = \sigma_{\min}(J_c)$ (positive in a local neighborhood), we have

$$\|Pc\| = \|J_c J_c^\dagger c\| \ge \sigma_{\min} \|J_c^\dagger c\|. \tag{143}$$

Combining (142), (143), and (141), we obtain

$$\Phi(V_A, \mu_A) \le \Phi(V, \mu) - \eta\lambda \|Pc\|^2 + \tfrac{L}{2} \|\Delta_A\|^2$$
$$\le \Phi(V, \mu) - \eta\lambda \|Pc\|^2 + L\eta^2 \|\Delta_{\tan}\|^2 + L\eta^2 \lambda^2 \|J_c^\dagger c\|^2$$
$$\le \Phi(V, \mu) - \eta\lambda \|Pc\|^2 + L\eta^2 \|\Delta_{\tan}\|^2 + L\eta^2 \lambda^2 \sigma_{\min}^{-2} \|Pc\|^2. \tag{144}$$

Choosing $\eta > 0$ sufficiently small such that

$$L\eta^2 \lambda^2 \sigma_{\min}^{-2} \le \tfrac{1}{2} \eta\lambda \quad \Longleftrightarrow \quad \eta \le \tfrac{\sigma_{\min}^2}{2L\lambda}, \tag{145}$$

we deduce from (144) that

$$\Phi(V_A, \mu_A) \le \Phi(V, \mu) - \tfrac{1}{2} \eta\lambda \|Pc\|^2 + L\eta^2 \|\Delta_{\tan}\|^2. \tag{146}$$

Note that $L\eta^2 \|\Delta_{\tan}\|^2$ is the quadratic remainder from "tangential movement." It does *not* affect first-order feasibility, and only contributes an $O(\eta^2)$ term in the energy. We absorb it into the final contraction constant.

**Part III: Non-expansiveness of the $\mu$-projection substep (B) and further descent.** Fix $V_A$, and consider $u := g(V_A) + \mu_A$. The update of $\mu$ in substep (B) is

$$\mu^+ = \Pi_{\mathbb{R}_{\ge 0}^{m_1}} (\mu_A - \eta_\mu u), \qquad \Pi_C \text{ denotes the Euclidean projection onto a convex set } C.$$

Hence

$$g(V_A) + \mu^+ = u + (\mu^- - \mu_A), \quad \mu^- := \Pi_{\mathbb{R}_{\ge 0}^{m_1}} (\mu_A - \eta_\mu u).$$

By the *non-expansiveness* and *firm non-expansiveness* of projections, for any $y$ one has $\|\Pi_C(x) - \Pi_C(y)\| \le \|x - y\|$ and $\langle \Pi_C(x) - \Pi_C(y), \, x - y \rangle \ge \|\Pi_C(x) - \Pi_C(y)\|^2$. Taking $y = \mu_A$ and $x = \mu_A - \eta_\mu u$, we obtain

$$\|\mu^- - \mu_A\| \le \eta_\mu \|u\| \quad \Rightarrow \quad \|g(V_A) + \mu^+\| \le \|u - \eta_\mu u\| = |1 - \eta_\mu| \|u\|.$$

Therefore, when $0 < \eta_\mu \le 1$, we have

$$\|g(V_A) + \mu^+\|^2 \le \|g(V_A) + \mu_A\|^2. \tag{147}$$

Since the equality block $h(V)$ remains unchanged in substep (B) (as $V$ is not updated), it follows that

$$\Phi(V^+, \mu^+) = \tfrac{1}{2}\|g(V_A) + \mu^+\|^2 + \tfrac{1}{2}\|h(V_A)\|^2 \le \tfrac{1}{2}\|g(V_A) + \mu_A\|^2 + \tfrac{1}{2}\|h(V_A)\|^2 = \Phi(V_A, \mu_A). \tag{148}$$

**Part IV: Combining the two substeps and deriving a quantitative contraction constant.** From (148) and (146), we obtain

$$\Phi(V^+, \mu^+) \le \Phi(V_A, \mu_A) \le \Phi(V, \mu) - \tfrac{1}{2} \eta\lambda \|Pc\|^2 + L\eta^2 \|\Delta_{\tan}\|^2.$$

Using (143), $\|Pc\| \ge \sigma_{\min} \|J_c^\dagger c\|$, we deduce

$$\Phi(V^+, \mu^+) \le \Phi(V, \mu) - \tfrac{1}{2} \eta\lambda \, \sigma_{\min}^2 \|J_c^\dagger c\|^2 + L\eta^2 \|\Delta_{\tan}\|^2. \tag{149}$$

Finally, by choosing $\eta > 0$ sufficiently small (in addition to satisfying (145)), for example ensuring

$$L\eta^2\|\Delta_{\tan}\|^2 \ \leq \ \tfrac{1}{4}\,\eta\lambda\,\sigma_{\min}^2\,\|J_c^\dagger c\|^2,$$

we can rewrite (149) as

$$\Phi(V^+,\mu^+) \ \leq \ \Phi(V,\mu) \ - \ \underbrace{\tfrac{1}{4}\,\eta\lambda\,\sigma_{\min}^2}_{=:\,\kappa}\,\|J_c^\dagger c(V,\mu)\|^2.$$

Since $\Phi = \tfrac{1}{2}\|c\|^2$, this matches exactly the form stated in the theorem:

$$\tfrac{1}{2}\|c(V^+,\mu^+)\|^2 \ \leq \ \tfrac{1}{2}\|c(V,\mu)\|^2 \ - \ \kappa\,\|J_c^\dagger c(V,\mu)\|^2, \qquad \kappa = \tfrac{1}{4}\,\eta\lambda\,\sigma_{\min}^2(J_c) > 0.$$

## G.13. Proof of Theorem 4.3

In the current batch, stack the equality/inequality constraints as $c(V,\mu) = 0$ (see the definition of the geometric feasible set in the main text), with the Jacobian with respect to $V$ denoted in block form as

$$J_V = \frac{\partial[g;h]}{\partial V} \in \mathbb{R}^{(m_1+m_2)\times B}.$$

At a point $(V,\mu)$ in the feasible set (or within a small neighborhood thereof), define

$$\mathcal{T} \ := \ \mathrm{Null}(J_V) \ \subset \ \mathbb{R}^B$$

as the *tangent space*, and let $P_{\mathcal{T}} : \mathbb{R}^B \to \mathcal{T}$ denote the *orthogonal projection* onto $\mathcal{T}$, with $P_{\mathcal{T}} = P_{\mathcal{T}}^\top = P_{\mathcal{T}}^2$. Write the gradient of the TD objective with respect to $V$ as $g_V := \nabla_V \mathcal{L}_{\mathrm{TD}}(V)$. The tangent gradient step is defined as

$$d_{\tan} \ := \ -P_{\mathcal{T}}\,g_V, \qquad V^+ \ := \ V + \eta\,d_{\tan} \ = \ V - \eta\,P_{\mathcal{T}}\,g_V, \tag{150}$$

where $\eta > 0$ is the step size. Since $d_{\tan} \in \mathcal{T} = \mathrm{Null}(J_V)$, the first-order perturbation along (150) leaves all constraints unchanged to first order (tangent safety; see also the first statement of Theorem 4.2 above).

**Restricted Smoothness & Restricted Strong Convexity (or PL) Assumptions.** There exists a constant $L_T > 0$ such that for all tangent increments $d \in \mathcal{T}$, the *restricted smoothness* (RS) condition holds:

$$\mathcal{L}_{\mathrm{TD}}(V + d) \ \leq \ \mathcal{L}_{\mathrm{TD}}(V) \ + \ \langle g_V,\, d \rangle \ + \ \tfrac{L_T}{2}\,\|d\|_2^2. \tag{151}$$

Moreover, there exists a constant $\mu_T > 0$ such that one of the following holds:

- *Restricted Strong Convexity* (RSC): $\langle \nabla\mathcal{L}_{\mathrm{TD}}(V+d) - \nabla\mathcal{L}_{\mathrm{TD}}(V),\, d \rangle \ \geq \ \mu_T\|d\|_2^2,\ \forall d \in \mathcal{T}$.

- *Restricted Polyak–Łojasiewicz (PL) Condition*:

$$\frac{1}{2}\,\|P_{\mathcal{T}}g_V\|_2^2 \ \geq \ \mu_T\Big(\mathcal{L}_{\mathrm{TD}}(V) - \mathcal{L}_{\mathrm{TD}}(V^\star)\Big), \tag{152}$$

where $V^\star$ denotes a stationary point/minimizer on $\mathcal{T}$ (in the sense of the tangent space to the feasible manifold).

Note: (151) only requires smoothness along tangent directions; RSC/PL further connect the tangent gradient energy with geometric quantities (distance or suboptimality), and are used in convergence rate analysis. For the *descent inequality in this theorem*, only (151) is needed; RSC/PL will be used for subsequent corollaries.

**Core Step: Tangent RS + Projected Gradient Step ⇒ One-Step Descent Bound.** Applying (151) to $d = d_{\tan} = -P_{\mathcal{T}}g_V$, we obtain

$$\begin{aligned}
\mathcal{L}_{\mathrm{TD}}(V^+) &\leq \ \mathcal{L}_{\mathrm{TD}}(V) \ + \ \langle g_V,\, -\eta\,P_{\mathcal{T}}g_V \rangle \ + \ \frac{L_T}{2}\,\|\eta\,P_{\mathcal{T}}g_V\|_2^2 \\
&= \ \mathcal{L}_{\mathrm{TD}}(V) \ - \ \eta\,\langle g_V,\, P_{\mathcal{T}}g_V \rangle \ + \ \frac{L_T\eta^2}{2}\,\|P_{\mathcal{T}}g_V\|_2^2.
\end{aligned} \tag{153}$$

Using the self-adjointness and idempotence of $P_{\mathcal{T}}$ ($P_{\mathcal{T}}^{\top} = P_{\mathcal{T}}$, $P_{\mathcal{T}}^2 = P_{\mathcal{T}}$), we have

$$\langle g_V,\ P_{\mathcal{T}}g_V \rangle \ =\ \langle P_{\mathcal{T}}g_V,\ P_{\mathcal{T}}g_V \rangle \ =\ \|P_{\mathcal{T}}g_V\|_2^2,$$

so that (153) becomes

$$\mathcal{L}_{\mathrm{TD}}(V^+) \ \leq\ \mathcal{L}_{\mathrm{TD}}(V)\ -\ \left(\eta - \tfrac{L_T}{2}\eta^2\right)\|P_{\mathcal{T}}g_V\|_2^2. \tag{154}$$

Define

$$\gamma(\eta)\ :=\ \eta - \tfrac{L_T}{2}\eta^2. \tag{155}$$

When $0 < \eta < \eta_0 := \frac{2}{L_T}$, we have $\gamma(\eta) > 0$, so from (154) we obtain

$$\mathcal{L}_{\mathrm{TD}}(V^+) \ \leq\ \mathcal{L}_{\mathrm{TD}}(V)\ -\ \gamma(\eta)\,\|P_{\mathcal{T}}g_V\|_2^2. \tag{156}$$

Writing $P_{\mathcal{T}}$ explicitly as $\mathrm{Proj}_{\mathrm{Null}(J_V)}$ yields the form stated in the theorem:

$$\mathcal{L}_{\mathrm{TD}}(V^+) \ \leq\ \mathcal{L}_{\mathrm{TD}}(V)\ -\ \gamma(\eta)\,\big\|\mathrm{Proj}_{\mathrm{Null}(J_V)}(g_V)\big\|_2^2, \quad \text{where } \gamma(\eta) = \eta - \tfrac{L_T}{2}\eta^2 > 0.$$

**Remarks on RSC/PL Assumptions (Not Required but Provide Rates).** Although (156) holds without assuming RSC/PL, under these conditions one can relate $\|P_{\mathcal{T}}g_V\|^2$ to the "distance/suboptimality" and thereby derive convergence rates:

- If the restricted PL condition (152) holds, then $\|P_{\mathcal{T}}g_V\|_2^2 \ \geq\ 2\mu_T\big(\mathcal{L}_{\mathrm{TD}}(V) - \mathcal{L}_{\mathrm{TD}}(V^\star)\big)$, plugging into (156) yields

  $$\mathcal{L}_{\mathrm{TD}}(V^+) - \mathcal{L}_{\mathrm{TD}}(V^\star) \ \leq\ \big(1 - 2\mu_T\gamma(\eta)\big)\big(\mathcal{L}_{\mathrm{TD}}(V) - \mathcal{L}_{\mathrm{TD}}(V^\star)\big),$$

  which gives *linear convergence* whenever $0 < \eta < 2/L_T$.

- If restricted strong convexity holds (together with restricted smoothness), one obtains a similar linear rate controlled by the restricted condition number $\kappa_T := L_T/\mu_T$.

These strengthen the conclusion of the theorem but are not needed for the validity of (156).

**Handling Tangential Feasibility and Normal Perturbations (Consistent with the Main Text).** Since $d_{\mathrm{tan}} \in \mathcal{T} = \mathrm{Null}(J_V)$, the tangential step (150) leaves the first-order effect on the constraints unchanged (tangential safety). In practice, if an additional "normal contraction" is employed for correction (see Theorem 4.2), its impact on $\mathcal{L}_{\mathrm{TD}}$ only appears at quadratic (or higher) order, which can be absorbed into the remainder of $\gamma(\eta)$ by choosing sufficiently small $\eta$ and contraction coefficients. (For a rigorous energy estimate, see the second-order remainder control in Theorem 4.2.)

**Conclusion.** Let $\eta_0 = \frac{2}{L_T}$. For any $0 < \eta < \eta_0$, we obtain from (156) that

$$\mathcal{L}_{\mathrm{TD}}(V^+) \ \leq\ \mathcal{L}_{\mathrm{TD}}(V)\ -\ \gamma\,\big\|\mathrm{Proj}_{\mathrm{Null}(J_V)}(g_V)\big\|_2^2, \qquad \gamma = \eta - \tfrac{L_T}{2}\eta^2 > 0,$$

which is exactly the statement of the theorem.

### G.14. Proof of Theorem D.3

Assume the environment satisfies the $\mathcal{G}$-invariance in §2, and that the parameterization has sufficient expressive power (to be defined precisely below). We need to prove: after a single "closure–alignment" step, the batch-wise equivariance residual $\|h(V)\|_2$ can be compressed below any pre-specified threshold $\epsilon > 0$; at the same time, the mapping $g \mapsto (T_g, \Pi_g)$ preserves a "near-group" structure on the finitely generated set.

**Setup and Notation.**

- Let the batch be $\mathcal{B} = \{s_i\}_{i=1}^B$, with shared representations $\phi_\theta(s) \in \mathbb{R}^d$, and write the matrix form $\Phi = [\phi_\theta(s_1), \ldots, \phi_\theta(s_B)] \in \mathbb{R}^{d \times B}$. For each $g \in \mathcal{G}_{\mathrm{batch}}$, denote $\Phi_g = [\phi_\theta(T_g s_1), \ldots, \phi_\theta(T_g s_B)] \in \mathbb{R}^{d \times B}$.

- The action–value head is taken to be linear (sufficiently general, as nonlinear heads can be absorbed into the representation layer): $Q_\theta(s, a) = w_a^\top \phi_\theta(s)$, where $W = [w_1, \ldots, w_{|\mathcal{A}|}] \in \mathbb{R}^{d \times |\mathcal{A}|}$. The corresponding $Q$-output matrices for the batch are $Y := W^\top \Phi \in \mathbb{R}^{|\mathcal{A}| \times B}$, $Y_g := W^\top \Phi_g \in \mathbb{R}^{|\mathcal{A}| \times B}$.

- For the action relabeling $\Pi_g \in \mathcal{P}_{|\mathcal{A}|}$ (a permutation matrix), its left multiplication acts on rows (the action dimension). Thus the batch-wise equivariance residual (vectorized Frobenius norm) can be written as

$$\| h(V) \|_2^2 = \sum_{g \in \mathcal{G}_{\text{batch}}} \| Y_g - \Pi_g^\top Y \|_F^2. \tag{157}$$

**Assumptions (Expressiveness & True Equivariance).** There exist a "true equivariance" $(T_g^\star, \Pi_g^\star)$ and parameters $\theta^\star$ such that, on the *same batch*, we have[7]

$$\Phi_g^\star := [\phi_{\theta^\star}(T_g^\star s_i)]_{i=1}^B = R_g^\star \Phi^\star, \qquad Y_g^\star = W^{\star\top} \Phi_g^\star = \Pi_g^{\star\top} Y^\star, \tag{158}$$

where $\Phi^\star = [\phi_{\theta^\star}(s_i)]_i$, and $R_g^\star \in O(d)$ are a set of orthogonal transformations. Equation (158) means: *on the representation side*, the true transformation acts orthogonally; *on the action head side*, the true relabeling acts as a permutation. This is consistent with the $\mathcal{G}$-invariance in §2 (see the main text discussion of $Q^\star$ as an equivariant fixed point).

**The Three Operators in the Closure–Alignment Step.** Given the current $(\theta, \{T_g, \Pi_g\})$, define a single step:

**(S1) Orthogonal–Permutation Projection.** Set $\widetilde{T}_g \leftarrow \text{Polar}(T_g) \in O(d)$ as the closest point to $T_g$ on $O(d)$ (in Frobenius norm), and $\widetilde{\Pi}_g \leftarrow \text{ProjPerm}(\Pi_g)$ as the closest point to $\Pi_g$ in the permutation set (implemented by Sinkhorn normalization followed by Hungarian projection to a vertex).

**(S2) Closure Contraction.** For finite generator pairs $(g, h)$, set $\widetilde{T}_{gh} \leftarrow \text{Polar}(\widetilde{T}_g \widetilde{T}_h)$, $\widetilde{\Pi}_{gh} \leftarrow \text{ProjPerm}(\widetilde{\Pi}_g \widetilde{\Pi}_h)$, and enforce the identity and inverse relations $\widetilde{T}_e = I$, $\widetilde{T}_{g^{-1}} = \widetilde{T}_g^\top$, $\widetilde{\Pi}_e = I$, $\widetilde{\Pi}_{g^{-1}} = \widetilde{\Pi}_g^\top$.

**(S3) Closed-form Alignment (Procrustes/Hungarian).** For each $g$, solve the *orthogonal Procrustes* problem:

$$T_g^{\text{new}} := \arg \min_{R \in O(d)} \| R\Phi - \Phi_g \|_F^2 = UV^\top, \quad \text{where } \Phi_g \Phi^\top = U\Sigma V^\top. \tag{159}$$

On the action side, construct the linear assignment cost matrix $M_g(a, b) = \| Y_g(a, \cdot) - Y(b, \cdot) \|_2^2$, and let $\Pi_g^{\text{new}}$ be the permutation solving $\min_{\Pi \in \mathcal{P}_{|\mathcal{A}|}} \langle M_g, \Pi \rangle$ (Hungarian algorithm in closed form).

Finally, set $T_g \leftarrow T_g^{\text{new}}$, $\Pi_g \leftarrow \Pi_g^{\text{new}}$ (or alternate once with (S1)(S2); the present proof is given for the "one-step composite" case of non-increasingness and approximability).

**Lemma G.21** (Optimality of Orthogonal Procrustes). *For any given matrix pair $(X, Y) \in \mathbb{R}^{d \times B}$, the problem*

$$\min_{R \in O(d)} \| RX - Y \|_F^2$$

*has a unique minimizer (if $\text{rank}(YX^\top) = d$) given by*

$$R^\star = UV^\top,$$

*where $YX^\top = U\Sigma V^\top$ is the SVD. The minimum value is*

$$\| Y \|_F^2 + \| X \|_F^2 - 2\| \Sigma \|_*.$$

*In particular,*

$$\| T_g^{\text{new}} \Phi - \Phi_g \|_F \leq \| R_g^\star \Phi - \Phi_g \|_F \quad (\forall g). \tag{160}$$

*Proof of Lemma G.21.* This is the classical orthogonal Procrustes result. The detailed proof is standard and omitted here. $\square$

---

[7]For nonlinear heads, the preceding layer can be absorbed into the representation $\phi_\theta$, so that the head becomes linear. Here "same batch" suffices to meet the "in-batch" condition in the theorem statement.

**Lemma G.22** (Optimality of Permutation Projection)**.** *Let $M \in \mathbb{R}_+^{|\mathcal{A}| \times |\mathcal{A}|}$ be a cost matrix. The Hungarian algorithm yields $\Pi^{\text{new}} \in \arg\min_{\Pi \in \mathcal{P}_{|\mathcal{A}|}} \langle M, \Pi \rangle$. Define $M_g(a, b) = \| Y_g(a, \cdot) - Y(b, \cdot) \|_2^2$. Then*

$$\| Y_g - \Pi_g^{\text{new}\top} Y \|_F^2 = \min_{\Pi \in \mathcal{P}_{|\mathcal{A}|}} \| Y_g - \Pi^\top Y \|_F^2 \leq \| Y_g - \Pi_g^{\star\top} Y \|_F^2. \tag{161}$$

*Proof of Lemma G.22.* Since $\|A - \Pi^\top B\|_F^2 = \sum_a \|A(a, \cdot) - B(\pi(a), \cdot)\|_2^2$, the problem is equivalent to a linear assignment formulation. The Hungarian algorithm solves this exactly. $\qquad\square$

**Lemma G.23** (Closure Contraction as Nearest Group Projection)**.** *For any $A \in \mathbb{R}^{d \times d}$, its orthogonal factor in the polar decomposition $\text{Polar}(A)$ satisfies*

$$\|\text{Polar}(A) - A\|_F = \min_{R \in O(d)} \|R - A\|_F.$$

*Similarly, for any $B \in \mathbb{R}^{|\mathcal{A}| \times |\mathcal{A}|}$, $\text{ProjPerm}(B)$ is its nearest point in the permutation set. Define closure errors*

$$\Delta_{g,h}^{(T)} := T_{gh} - T_g T_h, \qquad \Delta_{g,h}^{(\Pi)} := \Pi_{gh} - \Pi_g \Pi_h.$$

*Then after step (S2),*

$$\|\Delta_{g,h}^{(T)}\|_F \leq \min_{R \in O(d)} \|R - T_g T_h\|_F, \qquad \|\Delta_{g,h}^{(\Pi)}\|_F \leq \min_{\Pi \in \mathcal{P}} \|\Pi - \Pi_g \Pi_h\|_F. \tag{162}$$

*Proof of Lemma G.23.* This follows directly from the definition of nearest-point projection onto $O(d)$ or onto the permutation set. $\qquad\square$

**Decomposable Upper Bound of Equivariance Residual.** From (157),

$$\|h(V)\|_2^2 = \sum_g \| Y_g - \Pi_g^\top Y \|_F^2 = \sum_g \| W^\top \Phi_g - \Pi_g^\top W^\top \Phi \|_F^2.$$

For any temporary choice $R_g \in O(d)$ and permutation $\Pi_g$, by adding and subtracting $W^\top R_g \Phi$ and applying the triangle inequality, we obtain

$$\begin{aligned}
\| Y_g - \Pi_g^\top Y \|_F &= \| W^\top (\Phi_g - R_g \Phi) + (W^\top R_g \Phi - \Pi_g^\top W^\top \Phi) \|_F \\
&\leq \underbrace{\| W^\top (\Phi_g - R_g \Phi) \|_F}_{\text{representation alignment error}} + \underbrace{\| W^\top R_g \Phi - \Pi_g^\top W^\top \Phi \|_F}_{\text{action relabeling error}}.
\end{aligned} \tag{163}$$

For the first term, $\| W^\top (\Phi_g - R_g \Phi) \|_F \leq \|W\|_2 \cdot \|\Phi_g - R_g \Phi\|_F$. For the second term, if we define $Y := W^\top \Phi$ and $Y_{g,R} := W^\top R_g \Phi$, then $\min_{\Pi \in \mathcal{P}} \|Y_{g,R} - \Pi^\top Y\|_F$ can be minimized in closed form via the Hungarian algorithm (see Lemma G.22). Therefore, by (163) and choosing the optimal alignment $R_g = T_g^{\text{new}}$ (from Lemma G.21), $\Pi_g = \Pi_g^{\text{new}}$ (from Lemma G.22), we obtain

$$\| Y_g - \Pi_g^{\text{new}\top} Y \|_F \leq \|W\|_2 \cdot \min_{R \in O(d)} \|\Phi_g - R\Phi\|_F + \min_{\Pi \in \mathcal{P}} \| W^\top R_g \Phi - \Pi^\top W^\top \Phi \|_F. \tag{164}$$

**Equivariance Realizability $\Rightarrow$ Arbitrarily Small Error Upper Bound.** From (158), there exist $R_g^\star \in O(d)$ and $\Pi_g^\star$ such that $\Phi_g^\star = R_g^\star \Phi^\star$ and $W^{\star\top} R_g^\star \Phi^\star = \Pi_g^{\star\top} W^{\star\top} \Phi^\star$. If the parametrization is sufficiently expressive (universal approximation), then for any $\varepsilon_1, \varepsilon_2 > 0$, we can place the current $(\theta, W)$ within a neighborhood such that

$$\|\Phi - \Phi^\star\|_F \leq \varepsilon_1, \qquad \|W - W^\star\|_F \leq \varepsilon_2, \tag{165}$$

and (by $\mathcal{G}$-invariance and Lemma G.21) the Procrustes alignment objective $\min_{R \in O(d)} \|\Phi_g - R\Phi\|_F$ can be driven down to $\|\Phi_g^\star - R_g^\star \Phi^\star\|_F + o(1) = 0 + o(1)$. Similarly, the action relabeling assignment objective $\min_{\Pi \in \mathcal{P}} \| W^\top R\Phi - \Pi^\top W^\top \Phi \|_F$ is guaranteed by Lemma G.22; when $(\theta, W)$ is close to $(\theta^\star, W^\star)$ and $R$ is close to $R_g^\star$, its minimum also approaches $\|W^{\star\top} R_g^\star \Phi^\star - \Pi_g^{\star\top} W^{\star\top} \Phi^\star\|_F = 0$. Formally, combining (164), (165), and continuity, we obtain: for any $\epsilon > 0$, there exist sufficiently small $(\varepsilon_1, \varepsilon_2)$ and corresponding closure–alignment choices $(T_g^{\text{new}}, \Pi_g^{\text{new}})$ such that

$$\sum_{g \in \mathcal{G}_{\text{batch}}} \| Y_g - \Pi_g^{\text{new}\top} Y \|_F^2 \leq \epsilon^2/2. \tag{166}$$

**Closure Error and Near-Group Preservation.** Define the *near-group metric* as $\Gamma(T, \Pi) := \sum_{g,h} \left( \|\Delta_{g,h}^{(T)}\|_F^2 + \|\Delta_{g,h}^{(\Pi)}\|_F^2 \right)$. By Lemma G.23, step (S2) is a "pairwise nearest projection" of $\Gamma$, hence $\Gamma(\widetilde{T}, \widetilde{\Pi}) \leq \Gamma(T, \Pi)$. Step (S3), Procrustes/Hungarian (guaranteed optimal by Lemmas G.21, G.22), minimizes the equivariance residual in the data alignment sense, without breaking the unit/ inverse/ closure relations enforced by (S2). Therefore the closure–alignment composition satisfies: *equivariance residual decreases and the near-group metric does not increase*. More specifically, if there exists a true group $(R_g^\star, \Pi_g^\star)$ (as in (158)), then in its neighborhood $\Gamma$ can be compressed arbitrarily small, thus

$$\Gamma(T^{\mathrm{new}}, \Pi^{\mathrm{new}}) \leq \epsilon^2/2. \tag{167}$$

**Main Conclusion by Combining Both Sides.** From (157), (166), and (167), after one closure–alignment step we obtain $\|h(V)\|_2 \leq \epsilon$ and $\Gamma(T^{\mathrm{new}}, \Pi^{\mathrm{new}}) \leq \epsilon^2/2$, namely the in-batch equivariance residual can be reduced below any threshold $\epsilon > 0$, while $g \mapsto (T_g, \Pi_g)$ maintains the "near-group" structure (closure error arbitrarily small).

**Remark: Stability and Iterability After One Step.** By the *global optimality* of Lemmas G.21 and G.22, and the *nearest projection property* of Lemma G.23, the closure–alignment step is a *non-expansive* operator (with respect to the corresponding objectives). Under $\mathcal{G}$-invariance and expressivity, one can iterate it to further reduce $\|h(V)\|_2$ and $\Gamma$; but for the theorem it suffices to show that "a single step can reduce them below any given threshold"— because we can first place $(\theta, W)$ in a sufficiently small neighborhood (by (165)), then apply one closure–alignment step to obtain the desired conclusion.

Under $\mathcal{G}$-invariance and sufficient expressivity, choosing the closure–alignment step (Polar/ProjPerm + Procrustes + Hungarian) ensures that the in-batch equivariance residual $\|h(V)\|_2$ can be reduced below any $\epsilon > 0$, while preserving and shrinking the near-group metric. Thus $g \mapsto (T_g, \Pi_g)$ maintains a near-group structure on the finite generating set. The theorem is proved.

### G.15. Proof of Theorem D.4

In a neighborhood of the feasible set (see the geometric feasible set in the main text, $\mathcal{M} = \{(V, \mu) : c(V, \mu) = 0, \ \mu \geq 0\}$), denote the Jacobian of the constraints with respect to $V$ as $J_V = \frac{\partial[g;h]}{\partial V} \in \mathbb{R}^{(m_1+m_2) \times B}$, and define the tangent space

$$\mathcal{T} := \mathrm{Null}(J_V) \subset \mathbb{R}^B, \qquad P := \mathrm{Proj}_{\mathcal{T}} \ (\text{orthogonal projection onto } \mathcal{T}, \ P^\top = P = P^2).$$

Let the TD objective be $\mathcal{L}_{\mathrm{TD}}(V)$ with gradient $g_V = \nabla_V \mathcal{L}_{\mathrm{TD}}(V)$. The main update (value layer) is given by

$$V^+ = V + \eta(\Delta_{\mathrm{tan}} + \Delta_{\mathrm{nor}}), \qquad \Delta_{\mathrm{tan}} = \underbrace{-P\, g_V}_{\text{tangent step}}, \quad \Delta_{\mathrm{nor}} = \underbrace{-\lambda\, J_c^\dagger c(V, \mu)}_{\text{normal contraction}}, \tag{168}$$

and $\mu$ is updated by projected gradient descent on $g(V) + \mu$ to preserve non-negativity (see Theorem 4.2 in the main text). Near the feasible set, $c(V, \mu)$ is sufficiently small and monotonically decreases according to Theorem 4.2; hence the main convergence rate of this theorem is dominated by the tangent component (the normal component only serves to push violations back into $\{c = 0\}$, without contributing positively to first-order descent).

**Restricted Smoothness and Restricted Strong Convexity (or PL) Assumptions.** There exists a constant $L_T > 0$ such that for any tangent increment $d \in \mathcal{T}$,

$$\mathcal{L}_{\mathrm{TD}}(V + d) \leq \mathcal{L}_{\mathrm{TD}}(V) + \langle g_V, \, d \rangle + \frac{L_T}{2} \|d\|_2^2. \tag{169}$$

Moreover, there exists $\mu_T > 0$ such that either Restricted Strong Convexity (RSC) or Restricted PL holds on the tangent space:

$$\frac{1}{2} \|P\, g_V\|_2^2 \geq \mu_T \left( \mathcal{L}_{\mathrm{TD}}(V) - \mathcal{L}_{\mathrm{TD}}(V^\star) \right), \tag{170}$$

where $V^\star$ is a stationary point/minimizer in the tangent-space sense (i.e., $P\, g_{V^\star} = 0$).

Applying (169) to $d = -\eta\, P\, g_V$ and using $P^\top = P = P^2$, we obtain

$$\mathcal{L}_{\mathrm{TD}}(V - \eta P g_V) \leq \mathcal{L}_{\mathrm{TD}}(V) - \eta\,\langle g_V, P g_V\rangle + \frac{L_T \eta^2}{2}\,\|P g_V\|_2^2$$

$$= \mathcal{L}_{\mathrm{TD}}(V) - \underbrace{\left(\eta - \frac{L_T \eta^2}{2}\right)}_{:= \gamma(\eta)}\,\|P g_V\|_2^2. \tag{171}$$

When $0 < \eta < \eta_0 := \frac{2}{L_T}$, we have $\gamma(\eta) > 0$.

By Theorem 4.2 in the main text (the descent inequality of the violation energy $\frac{1}{2}\|c\|^2$), there exists $\kappa > 0$ such that for sufficiently small $(\eta, \eta_\mu, \lambda)$,

$$\tfrac{1}{2}\|c(V^+, \mu^+)\|_2^2 \ \leq\ \tfrac{1}{2}\|c(V, \mu)\|_2^2 - \kappa\,\|J_c^\dagger c(V, \mu)\|_2^2,$$

i.e., the normal component does not increase $c$ but decreases its squared $\ell_2$-norm by at least $\Theta(\|J_c^\dagger c\|^2)$. This implies that the difference between $V^+$ and $V - \eta P g_V$, namely $\eta\,\Delta_{\mathrm{nor}}$, is of order $O(\|J_c^\dagger c\|)$ in the first-order magnitude of $\|c\|$. Its effect on $\mathcal{L}_{\mathrm{TD}}$ under (169) is thus of higher-order, $O(\eta^2\|\Delta_{\mathrm{nor}}\|^2) = O(\eta^2\|J_c^\dagger c\|^2)$. In the neighborhood of the feasible set (where $c$ is small) and with sufficiently small step parameters, this part can be absorbed into the remainder of (171), leaving the main descent term unchanged.[8]

Combining (171) with the restricted PL inequality (170), we get

$$\mathcal{L}_{\mathrm{TD}}(V^+) - \mathcal{L}_{\mathrm{TD}}(V^\star) \leq \left(\mathcal{L}_{\mathrm{TD}}(V) - \gamma(\eta)\,\|P g_V\|_2^2\right) - \mathcal{L}_{\mathrm{TD}}(V^\star)$$

$$\leq \left(1 - 2\mu_T\,\gamma(\eta)\right)\left(\mathcal{L}_{\mathrm{TD}}(V) - \mathcal{L}_{\mathrm{TD}}(V^\star)\right). \tag{172}$$

When $0 < \eta < \frac{2}{L_T}$, we have $1 - 2\mu_T\gamma(\eta) \in (0, 1)$, and thus

$$\mathcal{L}_{\mathrm{TD}}(V_t) - \mathcal{L}_{\mathrm{TD}}(V^\star) \ \leq\ \left(1 - 2\mu_T\gamma(\eta)\right)^t \left(\mathcal{L}_{\mathrm{TD}}(V_0) - \mathcal{L}_{\mathrm{TD}}(V^\star)\right),$$

i.e., under restricted PL we obtain *linear convergence*, with a convergence factor depending only on the *restricted condition number* $\kappa_T = L_T/\mu_T$. Choosing the optimal step size $\eta = 1/L_T$ yields a factor $1 - \mu_T/L_T$. If PL does not hold but only restricted smoothness (and summability of subgradients), then we obtain the standard *sublinear* $O(1/t)$ convergence (omitted here; see the classical argument for projected gradient methods without PL).

Let $G$ denote the local (unprojected) TD update mapping in a neighborhood of the feasible set, whose Jacobian (Fréchet derivative) at some point $V$ is $J := DG(V)$. With tangent projection, the local linearization becomes

$$\delta V^+ \ \approx\ P J\,\delta V,$$

where $P$ is the orthogonal projection (both $P$ and $J$ are viewed as constant linear operators at the current point). Since $P$ is non-expansive ($\|P\|_2 = 1$), we have

$$\|P J\|_2 \ \leq\ \|P\|_2\,\|J\|_2 \ =\ \|J\|_2. \tag{173}$$

The spectral radius satisfies $\rho(X) \leq \|X\|_2$, hence

$$\rho(PJ) \ \leq\ \|PJ\|_2 \ \leq\ \|J\|_2, \qquad \text{and thus} \quad \rho(PJ) \ \leq\ \rho(J) \ \text{(in general } \rho(J) \leq \|J\|_2). \tag{174}$$

A sufficient condition for the *strict inequality* is that $J$ has nonzero components along $\mathcal{T}^\perp$, in which case $\|PJ\|_2 < \|J\|_2$ and therefore $\rho(PJ) < \rho(J)$.

**Lemma G.24** (Strict contraction condition of orthogonal projection)**.** *Let $P$ be the orthogonal projection onto some subspace. If $\mathrm{Range}(J) \nsubseteq \mathrm{Range}(P)$, then*

$$\|PJ\|_2 < \|J\|_2.$$

---

[8] Strictly, for $V^+ = V - \eta P g_V + \eta\Delta_{\mathrm{nor}}$, one can re-apply (169) and treat $\eta\Delta_{\mathrm{nor}}$ as a perturbation, which brings an additional term $\langle g_V, -\eta\Delta_{\mathrm{nor}}\rangle + O(\eta^2\|\Delta_{\mathrm{nor}}\|^2)$. Since $\Delta_{\mathrm{nor}} \in \mathrm{Range}(J_c^\dagger)$ and $J_c\Delta_{\mathrm{nor}} = -\lambda c$, this first-order term is controllable near the feasible set. Combining it with the contraction of the violation energy yields the claimed absorbability.

*Proof of Lemma G.24.* Assume, for contradiction, that $\|PJ\|_2 = \|J\|_2$. Take a unit vector $x^\star$ such that $\|Jx^\star\|_2 = \|J\|_2$. By the non-expansiveness of $P$ and the necessary and sufficient condition for equality, we have

$$\|Jx^\star\|_2 = \|PJx^\star\|_2 \Rightarrow Jx^\star \in \text{Range}(P).$$

Considering the sequence of maximizing singular vectors (whose existence is guaranteed by compactness), we conclude that all outputs in the *maximizing* directions lie within $\text{Range}(P)$. If there exists some $y$ such that $Jy \notin \text{Range}(P)$, then analyzing $x_t = \cos t\, x^\star + \sin t\, y$ and the extremal behavior of $\|Jx_t\|$ leads to a contradiction with $\|PJ\|_2 = \|J\|_2$: near the extremal direction, the output necessarily contains components in $\text{Range}(P)^\perp$, which are strictly reduced after projection. Hence it must hold that $\text{Range}(J) \subseteq \text{Range}(P)$, contradicting the assumption. This completes the proof. $\square$

Substituting "$\text{Range}(P) = \mathcal{T}$" back into Lemma G.24, we obtain: as long as the image space of $J$ contains some component lying in $\mathcal{T}^\perp$ (that is, the linearization of the update mapping has a nonzero component in the normal direction), we have a strict contraction in operator norm $\|PJ\|_2 < \|J\|_2$, and consequently, from (174),

$$\rho(PJ) < \rho(J).$$

This shows that *tangential projection* further reduces the spectral radius of the unconstrained linearization, and the reduction is *strict* whenever a nonzero normal component is present.

In the feasible region ($c = 0$) and near the tangential optimum $V^\star$, the linearization of one (approximate) unprojected gradient update can be written as $J \approx I - \eta H$, where $H = \nabla^2 \mathcal{L}_{\text{TD}}(V^\star)$. After tangential projection, $PJ \approx I - \eta\, PH$, which on $\mathcal{T}$ is equivalent to $I - \eta H_T$, where $H_T := PHP$. Restricted strong convexity and restricted smoothness provide spectral bounds for $H_T$:

$$\mu_T I \preceq H_T \preceq L_T I \qquad \text{(acting on } \mathcal{T}\text{)}.$$

Hence, the eigenvalues of $I - \eta H_T$ lie in $[1 - \eta L_T, \, 1 - \eta\mu_T]$. Choosing $0 < \eta < 2/L_T$, we obtain $\rho(PJ) = \rho(I - \eta H_T) \leq 1 - \eta\mu_T < 1$, and the precise convergence factor depends only on $(L_T, \mu_T)$, that is, the restricted condition number $\kappa_T = L_T/\mu_T$. This is fully consistent with the function-value linear rate derived in (172).

(i) Under restricted smoothness and restricted PL (or RSC), choosing $0 < \eta < 2/L_T$, the tangential part of the update in (168) satisfies

$$\mathcal{L}_{\text{TD}}(V^+) \leq \mathcal{L}_{\text{TD}}(V) - \left(\eta - \frac{L_T\eta^2}{2}\right)\|Pg_V\|_2^2.$$

Combined with the restricted PL condition, this yields a linear convergence rate with factor $1 - 2\mu_T\left(\eta - \frac{L_T\eta^2}{2}\right)$; in the absence of PL but with smoothness, the standard sublinear $O(1/t)$ decrease follows. The normal contraction only affects higher-order terms and ensures feasibility restoration, without altering the dominant convergence rate.

(ii) For local linearization, the tangential projection changes the Jacobian from $J$ to $PJ$, with $\rho(PJ) \leq \rho(J)$, and the inequality is strict whenever $J$ has a nonzero component in the normal direction. The convergence constant and spectral contraction depend only on the restricted condition number $\kappa_T$ and the spectral properties of $P$ on the tangent space. This completes the proof.

### G.16. Proof of Theorem D.5

We aim to rigorously establish two facts within a strict mathematical framework: (i) when the empirical distribution has a coverage rate $\rho$ of "symmetry pockets," and joint training is performed with equivariant alignment across $K$ views, the effective sample size is amplified from $N$ to

$$N_{\text{eff}} \approx N \cdot \big(1 + (K-1)\rho\big),$$

so that the variance bound is reduced accordingly by the same order; (ii) on top of this, when incorporating DAG monotonicity constraints via an *isometric projection* (or its smooth approximation), the prediction error

$$\mathbb{E}\big[(V - V^*)^2\big] = \underbrace{\text{Var}[V]}_{\text{statistical variance}} + \underbrace{\big(\mathbb{E}[V] - V^*\big)^2}_{\text{estimation bias}}$$

further benefits from variance reduction due to "multi-view sharing + geometric constraints," while the bias term is determined by the *mismatch* between the imposed structure (equivariance/partial order) and the true environment. This bias can be kept at a small magnitude through closure–alignment and confidence gating mechanisms.

**Modeling symmetry pockets and the sampling structure of multi-view sharing.** Let the empirical distribution $\mathcal{D}$ be a mixture of two components:

$$\mathcal{D} = (1-\rho)\,\mathcal{D}_{\text{gen}} + \rho\,\mathcal{D}_{\text{sym}},$$

where $\mathcal{D}_{\text{gen}}$ corresponds to the general sample distribution without symmetry pockets, and $\mathcal{D}_{\text{sym}}$ corresponds to the distribution of symmetry pockets: each pocket consists of $K$ views generated by (approximate) symmetry transformations. If the total sample size is $N$, then the expected number of samples from $\mathcal{D}_{\text{gen}}$ is $(1-\rho)N$, and the expected number of *pockets* from $\mathcal{D}_{\text{sym}}$ is $\rho N$ (for convenience, we treat the "unit" as a pocket, each containing $K$ views).

Consider an unbiased estimator of a scalar target (e.g., a particular coordinate of $V(s)$ at some fixed state $s$, or a linear functional of batch-level quantities), denoted as $\widehat{Z}$, which in the absence of structure has the noise decomposition

$$\widehat{Z} = Z^\star + \xi, \qquad \mathbb{E}[\xi] = 0, \ \ \mathrm{Var}[\xi] = \sigma^2, \tag{175}$$

where $Z^\star$ is the true underlying quantity (e.g., the coordinate of $V^*(s)$ or a linear functional of the batch "energy").

When the sample falls into a symmetry pocket, we have $K$ *views*:

$$\widehat{Z}^{(1)} = Z^\star + \xi^{(1)}, \quad \ldots, \quad \widehat{Z}^{(K)} = Z^\star + \xi^{(K)},$$

and assume (after equivariant alignment) that the noises satisfy independence or *weak correlation with equal variance*:

$$\mathbb{E}[\xi^{(k)}] = 0, \quad \mathrm{Var}[\xi^{(k)}] = \sigma^2, \quad \mathrm{Cov}(\xi^{(k)}, \xi^{(\ell)}) = \rho_{k\ell}\sigma^2, \ \ |\rho_{k\ell}| < 1, \ \ k \neq \ell. \tag{176}$$

Equivariance consistency (see $\mathcal{L}_{\text{eq}}$ in the main text) forces all $K$ views, after "state transformation + action relabeling," to be *aligned to the same target*. During optimization, this is equivalent to applying *average fusion* across the $K$ views (see Lemma below). That is, the within-pocket estimator becomes

$$\overline{Z} := \frac{1}{K}\sum_{k=1}^{K}\widehat{Z}^{(k)} = Z^\star + \overline{\xi}, \qquad \overline{\xi} := \frac{1}{K}\sum_{k=1}^{K}\xi^{(k)}. \tag{177}$$

**Lemma G.25** (Equivariance consistency $\Rightarrow$ multi-view averaging and variance reduction)**.** *Under* (176)*, we have*

$$\mathrm{Var}[\overline{\xi}] = \frac{1}{K^2}\sum_{k=1}^{K}\sum_{\ell=1}^{K}\mathrm{Cov}(\xi^{(k)}, \xi^{(\ell)}) \leq \frac{\sigma^2}{K}\cdot\Big(1 + (K-1)\bar{\rho}\Big), \tag{178}$$

*where* $\bar{\rho} := \max_{k\neq\ell}|\rho_{k\ell}| < 1$. *In the case of independent view noises* ($\rho_{k\ell} = 0$)*, we obtain* $\mathrm{Var}[\overline{\xi}] = \sigma^2/K$.

*Proof of Lemma G.25.* By quadratic expansion,

$$\mathrm{Var}[\overline{\xi}] = \frac{1}{K^2}\Big(K\sigma^2 + \sum_{k\neq\ell}\rho_{k\ell}\sigma^2\Big) \leq \frac{1}{K^2}\big(K + (K^2 - K)\bar{\rho}\big)\sigma^2,$$

which yields (178). $\qquad\square$

**Effective sample size amplification factor** $1 + (K-1)\rho$**.** For the entire set of $N$ samples (pockets),

- Non-symmetric part $(1-\rho)N$: each sample provides only one observation, variance contribution $\propto \sigma^2$;

- Symmetric part $\rho N$: each "pocket" allows averaging over $K$ views, variance contribution $\propto \mathrm{Var}[\overline{\xi}] \leq \sigma^2/K$ (with equality in the independent case).

Thus, in terms of *information content*, each of the $\rho N$ pockets effectively increases its sample count from $1$ to $K$. The *effective sample size* is

$$N_{\text{eff}} \approx (1-\rho)N\cdot 1 + \rho N\cdot K = N\big(1 + (K-1)\rho\big). \tag{179}$$

By classical statistical theory (stability, Rademacher complexity, CRLB, strongly convex ERM convergence rate, etc.), the "variance upper bound $\propto 1/N$" can be replaced with $N_{\text{eff}}$, giving the conclusion that variance shrinks by the same order. For example, under $\mu$-strong convexity and $L$-smoothness in ERM, the parameter estimator satisfies

$$\mathbb{E}\|\widehat{\theta} - \theta^\star\|^2 \lesssim \frac{\sigma^2}{\mu^2 N_{\text{eff}}},$$

and through an $L$-Lipschitz prediction mapping (e.g., $V = V_\theta$) we obtain

$$\mathbb{E}\|V - V^*\|^2 \lesssim \frac{C}{N_{\text{eff}}} = \frac{C}{N\big(1 + (K-1)\rho\big)} \quad \text{(constant } C \text{ depends on the problem).} \tag{180}$$

**Projection under DAG monotonicity constraints and non-expansiveness.** Let the batch value vector be $V \in \mathbb{R}^B$, and let DAG $D$ define the *closed convex* monotone cone

$$\mathcal{C}_D = \{V : V(u) \geq V(v) + \delta, \ \forall (u \to v) \in D\}$$

(with $\delta \geq 0$ possibly equal to 0 or a small margin). Let $\Pi_D : \mathbb{R}^B \to \mathcal{C}_D$ denote the Euclidean projection. Then $\Pi_D$ is a *non-expansive mapping* in Hilbert space:

$$\|\Pi_D(x) - \Pi_D(y)\|_2 \leq \|x - y\|_2, \qquad \forall x, y \in \mathbb{R}^B. \tag{181}$$

It also satisfies the *optimality characterization (Pythagoras property)*: for any $x \in \mathbb{R}^B$ and $z \in \mathcal{C}_D$,

$$\langle x - \Pi_D(x), \ z - \Pi_D(x) \rangle \leq 0. \tag{182}$$

The equivariant subspace $\mathcal{S}_\mathcal{G} = \{V : V \text{ is equivariant}\}$ is a linear closed subspace, and the orthogonal projection $\Pi_\mathcal{G}$ is likewise non-expansive. We now analyze in two steps how "equivariant sharing (producing $\widetilde{V}$), followed by monotone projection (yielding $V = \Pi_D(\widetilde{V})$)" affects variance and bias.

**Lemma G.26** (Projection does not increase variance). *Let $X$ be a square-integrable random vector, and let $P$ be the Euclidean projection onto a closed convex set (or the orthogonal projection onto a closed subspace). Then*

$$\text{Var}\big[P(X)\big] \leq \text{Var}[X]. \tag{183}$$

*Proof of Lemma G.26.* Let $m = \mathbb{E}[X]$. By non-expansiveness (181) and Jensen's inequality, $\mathbb{E}\|P(X) - P(m)\|_2^2 \leq \mathbb{E}\|X - m\|_2^2$. Since $P(m)$ is constant, we have $\text{Var}[P(X)] = \inf_c \mathbb{E}\|P(X) - c\|^2 \leq \mathbb{E}\|P(X) - P(m)\|^2$, while $\text{Var}[X] = \inf_c \mathbb{E}\|X - c\|^2 = \mathbb{E}\|X - m\|^2$. Combining these yields (183). $\square$

**Lemma G.27** (Bias after projection can be bounded by structural mismatch). *Let $x^\star \in \mathbb{R}^B$ be the ground truth (e.g., the batch vector of $V^*$), $X$ a random estimator, and $P$ a projection onto a closed convex set. Then*

$$\mathbb{E}\|P(X) - x^\star\|_2^2 \leq \mathbb{E}\|X - x^\star\|_2^2 + \underbrace{\|P(x^\star) - x^\star\|_2^2}_{\text{structural bias (mismatch)}}. \tag{184}$$

*Proof of Lemma G.27.* For any $\omega$, expand

$$\|P(X) - x^\star\|^2 = \|P(X) - P(x^\star)\|^2 + \|P(x^\star) - x^\star\|^2 + 2\langle P(X) - P(x^\star), P(x^\star) - x^\star \rangle.$$

By the projection optimality condition (182) (taking $x = x^\star$, $z = P(X) \in \mathcal{C}$), the last term is $\leq 0$. By non-expansiveness, $\|P(X) - P(x^\star)\| \leq \|X - x^\star\|$. Taking expectations yields (184). $\square$

**Connecting the two-step structure (equivariance & monotonicity).** Let $\widetilde{V}$ denote the estimate after "equivariance sharing," and let $V = \Pi_D(\widetilde{V})$ be the result after monotone projection. For the variance term, by Lemma G.26, first applying the equivariant projection (viewed as the orthogonal projection $\Pi_\mathcal{G}$ onto the subspace $\mathcal{S}_\mathcal{G}$, or equivalently the linear averaging of multiple views), and then the monotone projection $\Pi_D$, the two non-expansiveness steps yield

$$\text{Var}[V] = \text{Var}\big[\Pi_D(\widetilde{V})\big] \leq \text{Var}[\widetilde{V}] \leq \frac{C}{N\big(1 + (K-1)\rho\big)}, \tag{185}$$

where the last inequality comes from (179)–(180) (replacing the unstructured $1/N$ variance bound by $1/N_{\text{eff}}$).

For the bias term, define the feasible set with both structures as $\mathcal{C} := \mathcal{S}_{\mathcal{G}} \cap \mathcal{C}_D$, and let $P_{\mathcal{C}}$ be its projection. For any estimator $X$, Lemma G.27 gives

$$\mathbb{E}\|P_{\mathcal{C}}(X) - V^*\|^2 \;\leq\; \mathbb{E}\|X - V^*\|^2 \;+\; \underbrace{\|P_{\mathcal{C}}(V^*) - V^*\|^2}_{:=\,\text{Bias}^2_{\text{struct}}}. \tag{186}$$

Taking $X$ to be the unstructured estimate, and $P_{\mathcal{C}}$ to be the composite projection "equivariance then monotonicity" (also 1-Lipschitz), we obtain

$$\mathbb{E}\|V - V^*\|^2 \;\leq\; \underbrace{\mathbb{E}\|\widetilde{V} - V^*\|^2}_{\text{variance-dominated term}} \;+\; \text{Bias}^2_{\text{struct}}.$$

Further decomposing into variance and bias,

$$\mathbb{E}\|V - V^*\|^2 = \text{Var}[V] \;+\; \|\mathbb{E}[V] - V^*\|^2$$
$$\leq \frac{C}{N\big(1 + (K-1)\rho\big)} \;+\; \text{Bias}^2_{\text{struct}}, \tag{187}$$

where $\text{Bias}^2_{\text{struct}} = \|P_{\mathcal{C}}(V^*) - V^*\|^2$ is the *structural mismatch error*: when the ground truth $V^*$ is already equivariant and satisfies the DAG monotonicity (or approximately so), this term is 0 (or small).

**Quantitative interpretation of "controllable structural bias."**

- *Closure–alignment.* Group-parameter side projections (Polar/ProjPerm with closure shrinking and Procrustes/Hungarian alignment) ensure the learned $(T_g, \Pi_g)$ approximate true equivariance, so the distance $\|P_{\mathcal{G}}(V^*) - V^*\|$ to the equivariant subspace $\mathcal{S}_{\mathcal{G}}$ can be compressed below any small threshold (see theorems on closure consistency and equivariant approximation in the main text).

- *Confidence gating.* By imposing DAG constraints only on high-confidence edges, we avoid forcing incorrect edges, so the distance from $V^*$ to the monotone cone $\mathcal{C}_D$ also remains small (since high-confidence edges are likely consistent with the true partial order; mismatched edges are rare and suppressed by the threshold).

Hence $\text{Bias}^2_{\text{struct}}$ can be jointly controlled to be small by these two mechanisms, ensuring it does not dominate the variance reduction term in (187).

**Aligning the conclusion with theorem statement.** From (179), we obtain the effective sample size amplification factor

$$\boxed{N_{\text{eff}} \;\approx\; N \cdot \big(1 + (K-1)\rho\big)}$$

—this is the statistical characterization that "equivariant alignment is equivalent to averaging $K$ views within each pocket of coverage $\rho$."

Therefore, in the quadratic decomposition of the test error

$$\mathbb{E}(V - V^*)^2 \;=\; \text{Var}[V] \;+\; \big(\mathbb{E}[V] - V^*\big)^2,$$

the variance term is reduced according to (185) and (180) ($\propto 1/N_{\text{eff}}$), while the monotone projection does not increase variance; the bias term is determined by $\text{Bias}^2_{\text{struct}}$ in (186), i.e., the mismatch distance between equivariance/partial-order structure and the ground truth. This bias can be kept small through closure–alignment and confidence gating. The theorem follows.

### G.17. Proof of Theorem D.6

When the equality/inequality constraints in §4.1 are *exactly satisfied* by the oracle structures (the true symmetry group $\mathcal{G}$ and the true partial order $D^\star$), training takes place on the feasible manifold. Suppose the coverage rate of "symmetric pockets" in the data is $\rho \in [0, 1]$, with each pocket containing $|\mathcal{G}|$ orbit views. Let $W(D^\star)$ denote the *width* of $D^\star$ (the size of the largest antichain). Each batch contains $N_{\text{batch}}$ comparison points. We will prove:

(i) The statistical sample complexity satisfies

$$N_{\text{oracle}}(\varepsilon) \;\lesssim\; \frac{W(D^{\star})}{1 + (|\mathcal{G}| - 1)\rho} \cdot \frac{C_T}{\varepsilon},$$

and hence yields a statistical speedup factor relative to the unstructured baseline;

(ii) The optimization linear rate improves from $1 - \mu/L$ to $1 - \mu_T/L_T$;

(iii) Multiplying the two effects gives an end-to-end multiplicative reduction in budget.

**(i) Statistical Sample Complexity: Symmetry Sharing & Partial Order Collapse.** We estimate the function class capacity in two steps and derive a generalization bound (or expected residual bound):

*Step A (Symmetry sharing $\Rightarrow$ effective sample size amplification).* Cluster samples covered by $\rho$ into orbits: the asymmetric part, proportion $(1 - \rho)$, contributes 1 observation each; the symmetric part, proportion $\rho$, contributes $|\mathcal{G}|$ averaged views each. Equivariant consistency alignment is equivalent (cf. the previous argument on "$K$-view averaging"), yielding

$$N_{\text{eff}}^{(\text{sym})} \;\approx\; N\big(1 + (|\mathcal{G}| - 1)\rho\big). \tag{188}$$

Thus, any statistical bound of the form "$\propto 1/N$" can be improved by replacing $N$ with $N_{\text{eff}}^{(\text{sym})}$. We denote this amplification factor as

$$\boxed{\;\text{symmetry sharing gain} \;=\; 1 + (|\mathcal{G}| - 1)\rho\;}$$

*Step B (DAG monotonicity $\Rightarrow$ collapse of degrees of freedom to $W(D^{\star})$).* For a fixed batch, let the width of $D^{\star}$ be $W = W(D^{\star})$. By Dilworth's theorem, there exist $W$ chains $C_1, \ldots, C_W$ covering all $N_{\text{batch}}$ points, and $W$ is the *minimum* number of chains required. In the oracle case, the partial order constraints are

$$V(u) \;\geq\; V(v) + \delta, \quad \forall (u \to v) \in D^{\star}, \qquad \delta \geq 0.$$

For each chain $C_j$ (totally ordered), monotonic constraints shrink the $n_j = |C_j|$ free parameters to the set of *monotone sequences*; in the ideal (noise-free) limit, isotonic blocks collapse further to *a constant number* of free values. Conservatively, we only use the *upper bound*: each chain is treated as one-dimensional and contributes at most order-1 independent information. Formally, using Rademacher complexity (or pseudodimension/Natarajan dimension) as a capacity measure,

$$\mathfrak{R}_N\big(\mathcal{F}_{\text{mono}}(D^{\star})\big) \;\lesssim\; \sqrt{\frac{W}{N}}, \tag{189}$$

whereas for the *unstructured* class $\mathcal{F}_{\text{free}} = \{V \in \mathbb{R}^{N_{\text{batch}}}\}$ we have

$$\mathfrak{R}_N\big(\mathcal{F}_{\text{free}}\big) \;\lesssim\; \sqrt{\frac{N_{\text{batch}}}{N}}. \tag{190}$$

((189) follows from "chain decomposition + entropy bound for 1D monotone classes," summed over $W$ chains; (190) follows from the $N_{\text{batch}}$ independent free coordinates. Constants are absorbed into $C_T$.)

When the empirical risk involves a *strongly convex/PL* loss (e.g., squared residuals, quadratic Bellman residuals), standard generalization bounds give

$$\mathbb{E}[\mathcal{E}_{\text{Bell}}(\theta) - \mathcal{E}_{\text{Bell}}(\theta^{\star})] \;\lesssim\; C_T\, \mathfrak{R}_N\big(\mathcal{F}\big), \tag{191}$$

where $C_T$ depends on restricted Lipschitz/strong convexity constants. Plugging $\mathcal{F} = \mathcal{F}_{\text{mono}}(D^{\star})$ and replacing $N$ by $N_{\text{eff}}^{(\text{sym})}$ from (188), we obtain

$$\mathbb{E}\big[\mathcal{E}_{\text{Bell}}(\theta)\big] \;\lesssim\; C_T \sqrt{\frac{W}{N_{\text{eff}}^{(\text{sym})}}} \;\approx\; C_T \sqrt{\frac{W}{N\big(1 + (|\mathcal{G}| - 1)\rho\big)}}.$$

If we characterize the required $N$ by the usual "$1/N$-type" rate such that the expected squared residual $\leq \varepsilon$, we obtain

$$N_{\text{oracle}}(\varepsilon) \lesssim \frac{W}{1 + (|\mathcal{G}| - 1)\rho} \cdot \frac{C_T}{\varepsilon}. \tag{192}$$

This is precisely the bound stated in part (i). Compared to the unstructured baseline $N_{\text{base}}(\varepsilon) \sim C/\varepsilon$, if the dominant term of $C$ is $N_{\text{batch}}$ (corresponding to the capacity scale in (190)), then the multiplicative statistical speedup factor is at least

$$\boxed{\underbrace{1 + (|\mathcal{G}| - 1)\rho}_{\text{symmetry sharing}} \times \underbrace{\frac{N_{\text{batch}}}{W(D^\star)}}_{\text{partial-order collapse}}}.$$

Intuitively, this means: $N_{\text{batch}}$ *free* comparison points per batch degenerate, under the oracle partial order, into aggregated statistics across only $W(D^\star)$ chains.

**(ii) Improvement of Optimization Linear Rate and Spectral Contraction.** Under hard constraints, updates are performed on the *tangent space* $\mathcal{T} = \text{Null}(J_V)$. Let $P$ denote the orthogonal projection onto $\mathcal{T}$. In a neighborhood of the optimum, the linearization of the TD mapping is denoted $J$, and the restricted one-step update is approximated as

$$\delta V^+ \approx PJ\,\delta V \approx (I - \eta\,P\nabla^2 L)\,\delta V.$$

By restricted smoothness/strong convexity (or PL) together with the restricted linear convergence theorem (Theorem D.4), there exist constants $L_T, \mu_T > 0$ (restricted constants) such that, for suitable $\eta$,

$$\|\delta V_t\| \leq \left(1 - \mu_T/L_T\right)^t \|\delta V_0\|.$$

In contrast, the unconstrained rate is $1 - \mu/L$. Thus, the ratio of iteration complexity (the number of steps to reach a fixed accuracy) is approximately

$$\boxed{\frac{\kappa}{\kappa_T} = \frac{L/\mu}{L_T/\mu_T} \geq 1}.$$

On the other hand, the spectral radius satisfies $\rho(PJ) \leq \rho(J)$, and is strictly smaller whenever $J$ has nonzero components in the normal direction:

$$\rho(PJ) < \rho(J).$$

This provides a *dynamical* perspective of "stronger contraction."

**(iii) Multiplicative Reduction of End-to-End Budget.** Multiplying the *statistical* gains of (i) with the *optimization* gains of (ii): if we take the "sample size $\times$ iteration count" as the wall-clock budget metric, then

$$\text{Budget}_{\text{oracle}} \lesssim \underbrace{\frac{1}{1 + (|\mathcal{G}| - 1)\rho}}_{\text{symmetry sharing}} \times \underbrace{\frac{W(D^\star)}{N_{\text{batch}}}}_{\text{partial-order collapse}} \times \underbrace{\frac{\kappa_T}{\kappa}}_{\text{restricted condition number}} \times \text{Budget}_{\text{base}}.$$

Equivalently,

$$\boxed{\left(1 + (|\mathcal{G}| - 1)\rho\right) \cdot \frac{N_{\text{batch}}}{W(D^\star)} \cdot \frac{\kappa}{\kappa_T}}^{-1}$$ 
times the budget (or equivalently, taking the reciprocal gives the *speedup factor*).

Thus, written as a speedup factor, the theorem states:

$$\boxed{\left(1 + (|\mathcal{G}| - 1)\rho\right) \cdot \frac{N_{\text{batch}}}{W(D^\star)} \cdot \frac{\kappa}{\kappa_T}}.$$

When $\rho \approx 1$, we recover an orbit-level amplification of order $|\mathcal{G}|$; when $W(D^\star) \ll N_{\text{batch}}$, we approach "full-chain collapse."

---

**Algorithm 4** One training step with learned symmetry & logic-order regularization

---

1. Sample minibatch $\mathcal{B} = \{(s_i, a_i, r_i, s_i')\}_{i=1}^{B}$. Compute TD targets $y_i$ and nudges $\Delta_i$ via (7).

2. Build weighted candidate comparisons from $\{\Delta_i\}$ and construct an acyclic preference skeleton $D$ by greedy add-unless-cycle (Section 3.2).

3. Run $T_{\mathrm{iso}}$ inner gradient steps on the auxiliary variables $\hat{V}$ to minimize the isotonic surrogate (Section 3.3), yielding $\hat{V}^{\star}$.

4. Compute $\mathcal{L}_{\mathrm{total}}$ and backpropagate through the value network and both regularizer branches.

---

**Remark on the Constant $C_T$.** The constant $C_T$ collects the (restricted) Lipschitz constants, noise variance, and restricted complexity factors (such as the constant term in (189) and possible logarithmic terms). Since we compare *ratios* (multiplicative factors), the ratio $C_T/C$ between oracle and baseline can be treated as of the same order, and does not affect the dominant three-fold multiplicative structure.

**Conclusion.** We have thus established statements (i)–(iii): statistical sample complexity scales as $W/(1 + (|\mathcal{G}| - 1)\rho)$; the optimization linear rate improves from $\kappa$ to $\kappa_T$; and the end-to-end budget contracts multiplicatively by these three factors. This completes the proof.

## H. Algorithm

Algorithm 4 summarizes a single training step; it is fully differentiable except for the greedy DAGification step, which only selects the edge set $D$ used by the differentiable isotonic surrogate.

Geometrically, the symmetry loss keeps updates close to $\mathsf{Eq}(\mathcal{G})$, while the isotonic surrogate keeps values close to a batchwise approximation of $\mathsf{Mono}(D)$. We next state concise guarantees on residual reduction, stability, and large-sample consistency.

