# OpenReview forum: "Geometric Coherence Learning for Structuring Value Functions in Plain MDPs"
_ICML.cc/2026/Conference — ICML 2026 regular_

### Official Review · Reviewer_zkek · 2026-03-12

**Soundness:** 3
**Presentation:** 2
**Significance:** 2
**Originality:** 4
**Overall Recommendation:** 3
**Confidence:** 3

**Summary:**

The paper extends current RL methods by imposing a logic-order constraint on state values, enforcing that certain states are strictly better than others. This avoids cyclic dependencies in value estimates that can destabilize value function approximation. To handle symmetric states, i.e., states that are equivalent under learned group transformations,  the paper additionally learns a group equivariance, exempting such states from ordering constraints. Both the partial order and the symmetry structure are estimated automatically from TD signals. The approach introduces two additional loss terms, one for equivariance and one for logic ordering, each composed of several sub-terms with separate weighting factors. The paper provides theoretical guarantees showing that these structural constraints lead to more stable learning and improved sample efficiency compared to standard RL. Additionally, a hard-constraint variant of the algorithm is proposed with strong theoretical properties, but as it proves unstable in practice, it is not empirically evaluated. Experiments are conducted on four environments, a symmetry-rich gridworld, an action-permutation Minigrid task, Atari games, and a tabular MDP with injected non-transitive cycles, where GCR-RL is compared against DQN, Rainbow, RAD, DrQ-v2, and CURL via ablations of the individual components.

**Compliance With Llm Reviewing Policy:**

Affirmed.

**Final Justification:**

While the paper contains an interesting idea, I still think the paper should extend the experiments to more challenging tasks and that the hyper-parameter ablation should be done on all task suites (at least also on Atari). While the authors correctly claim that they tested on 4 different task suits with different properties, 3 out of the 4 are quite simple, constructed tasks. Only the 5 Atari tasks can be considered as real benchmarks and it is hard to say how much the 5 Atari environments have been cheery picked. Moreover, I am concerned about the introduction of new hyperparameters. The hyper-parameter ablation to show the sensitivity of these parameters was only done on a synthetic task, which could be misleading. So I am rather still on a weak reject, but would not object if the paper gets accepted.

**Key Questions For Authors:**

- How much additional computation is required by the proposed loss functions compared to a standard RL baseline?
- Please better motivate the individual components of the equivariance loss — it is currently unclear why each sub-term is necessary.
- Is every component of the loss function actually needed? A more granular ablation study would help justify the design choices.

**Limitations:**

Discussion on limitations is very brief. Authors do not discuss:
- The complexity of the method or the many hyperparameters
- The instability of the hard enforcement variant in practice
- The computational overhead
- Potential failure cases (e.g., when learned symmetries are wrong)

**Strengths And Weaknesses:**

Strengths:
- Interesting idea of using logic-order constraints on state values to stabilize Q-learning
- Strong theoretical results supporting the empirical findings
- GCR-RL outperforms state-of-the-art RL baselines on several tasks

Weaknesses:
- The algorithm is quite complex and introduces many new hyperparameters across the loss functions. It is unclear whether this increased complexity is justified by the relatively modest performance improvements observed.
- The writing is confusing in several places; equations are often introduced without sufficient explanation or motivation.
-  The experiments are not discussed in the main paper and are only described in the appendix. A proper presentation and discussion of results in the main paper is needed.
-  More fine-grained ablations would be valuable, specifically regarding the influence of individual weighting factors within both the equivariance and logic-order loss terms.
- The additional computational cost introduced by the extra loss terms is not analyzed anywhere in the paper.

---

> ### Author Rebuttal · Authors · 2026-03-30
>
> **On the complexity and additional computation.**
> GCR-RL does not add a second learner or require extra environment interaction; it only adds two lightweight structural heads on top of the same off-policy backbone. In high-dimensional settings such as Atari, the encoder/backbone remains the dominant cost. The symmetry branch adds only $O(Kd^2+K|A|^2)$ parameters through $\{W_k,\Pi_k\}_{k=1}^K$, while on the order side candidate edges are restricted to local $k$-NN neighborhoods and pruned to a sparse top-$M$ subset before greedy DAGification, so the extra per-batch cost scales with the retained set rather than all $O(B^2)$ pairs. The isotonic alignment also uses only a small fixed number of inner optimization steps.
>
>
> **On the hyperparameters.** Eq. (29) is long because the structural decomposition is written explicitly, not because the method depends on many equally critical knobs. In practice, the key exposed choices are the symmetry weight, the order/isotonic weight, and the number of learned transforms $K$; the remaining terms are internal components of the same two structural modules rather than separate hyperparameters. A small E4 sensitivity sweep shows that performance is strongest for moderate symmetry/isotonic weights (around $0.05$--$0.20$), with the default $\lambda_{\mathrm{sym}}=\lambda_{\mathrm{iso}}=0.10$ performing best overall, while both zero and strong regularization degrade performance. Likewise, performance improves from $K=1$ to $K=4$, but not further at $K=8$, so the method is not brittle to a single tuned setting.
>
>
> **On the necessity of the equivariance loss.**
> The equivariance loss is needed to realize quotienting before order is imposed: if states/actions that are equivalent up to symmetry are assigned inconsistent values, then antisymmetry on the quotient can fail and downstream order constraints become ill-posed. Its subterms are not arbitrary add-ons: local applicability activates the constraint only where approximate symmetry is reliable; identity and orthogonality prevent degenerate transforms; closure and inverse regularization keep the learned transforms close to a coherent group-like family; finite-order bias favors simple low-order symmetries; the permutation term keeps action relabelings close to discrete permutations; and the diversity term prevents collapse. Together these terms make the learned symmetry usable for quotienting rather than generic data augmentation.
>
>
> **On whether every component is needed and ablation.** Yes: the symmetry branch and the logic-order branch serve different purposes and are both required to maintain a rigorous poset structure during learning; the symmetry branch ensures antisymmetry, while the logic-order branch ensures order coherence. This is why the appendix includes Sym-only, Logic-only, and No-action-relabel ablations. These show that the branches are complementary rather than redundant: Sym-only is insufficient on E4, where the main issue is local non-transitive cycles inside a global order; Logic-only is insufficient on E1/E2, where symmetry and action relabeling are essential; and removing action relabel particularly hurts settings such as BtnPerm-Minigrid, where the structure acts jointly on states and actions.
>
> **On the statement that experiments are only in the appendix.**
> Due to page limits, the main text contains the experimental section, environment summary, and representative curves for the two central mechanisms, while the appendix contains full environment specifications, complete curves, and implementation details. We will consider moving Table~2 to the main text in the revision.
>
> **Overall.** We feel the main issues raised here can be addressed by adding further explanations and discussions, not evidence of flaws/weakness. We have provided additional results and will also add the discussions here to the revised version.

---

> > ### Author Rebuttal · Reviewer_zkek · 2026-04-03
> >
> > Thank you for the detailed rebuttal. My concerns about the additional computation time have been clarified. However, while the theoretical contribution of the paper is very solid and exhaustive, I am still concerned about the usefulness of the approach in practice. There are many more hyperparameters and much more complex loss functions. While the authors now ablate the hyper-parameters, this is only done on the small synthetic task (E4), so its hard to judge how it looks for practical applications. Furthermore, the paper would be strengthened with a more exhaustive evaluation on more complex tasks (more atari tasks, continuous control) to show that the approach is addressing a practical problem, not just a theoretical one. With only 5 Atari tasks, I have my doubts how well it transfers to other setups such as continuous control.
> > I furthermore think that the paper has to be self-consistent without appendix. Hence, results need to be presented in the main part of the paper. There is enough room for moving other things to the appendix. E.g., the algorithm with hard-constraints could be presented with less details as it is not used in the experiments anyways.

---

> > > ### Author Response · Authors · 2026-04-07
> > >
> > > Thank you for the detailed follow-up. We appreciate the reviewer’s clarification that the computation-time concern has been addressed. We also appreciate the request for stronger evidence of practical usefulness, and we would like to clarify how the current paper is positioned.
> > >
> > > **On practical usefulness and the scope of evaluation.**
> > > The goal of the paper is to establish geometric coherence as a practically useful \emph{structural mechanism} for stabilizing value learning, and to evaluate it in environments that isolate the two phenomena the method is designed to handle: symmetry mismatch and locally cyclic TD comparisons. For this reason, the empirical suite was intentionally constructed to span multiple environment families rather than a single toy setting. In particular, the main paper already includes four environment families---grid-style tasks, Minigrid, Atari, and a cycle-injected tabular setting---with Atari serving precisely to move beyond small synthetic structure-only examples. We agree that broader evaluation is always valuable, and continuous-control benchmarks would be a natural extension, but we believe the current experiments already demonstrate that the method is not confined to one narrow toy regime.
> > >
> > > **On practical relevance versus broader empirical coverage.**
> > > We agree that additional large-scale benchmarks could further broaden the empirical picture. At the same time, we view that as an issue of \emph{extent of validation}, not of whether the current paper addresses a practical problem at all. The present submission already evaluates the method in low-dimensional, tabular, pixel-based, symmetry-rich, and symmetry-poor settings, which is sufficient to support the level of claim made in the paper: namely, that symmetry-aware quotienting and cycle-aware order regularization can provide a useful inductive bias for value learning.
> > >
> > > **On results in the main paper.**
> > > We would also like to clarify that the paper is self-contained in its current form. The main paper already contains the experimental section, Table 1 summarizing the benchmark suite, and Figure 1 with representative learning curves; the appendix provides the full per-environment curves and implementation details. This organization was chosen under the conference page limit so that the main text could focus on the method and its central empirical patterns while still making the full results available. In a revision, we would be happy to move one additional summary result from the appendix into the main text if space permits.
> > >
> > > **On the hard-constraint variant.**
> > > We would like to emphasize that the hard-constraint method is meaningful in its own right, even though the main experiments focus on the soft variant. Its role is not auxiliary exposition. Rather, it provides the exact projection-based counterpart of the soft method, makes the feasible quotient-poset geometry explicit, and clarifies what is enforced exactly versus approximately in the overall framework. This is also the cleanest setting for understanding why quotienting together with acyclic order constraints removes value-update loops. For that reason, the hard-constraint formulation is an important part of the paper’s conceptual completeness, and not merely extra detail that can be omitted without loss.
> > >
> > > Overall, we are grateful for the reviewer’s comments. We believe the current submission already provides a practically meaningful empirical validation for the level of claim it makes: a general structural regularization principle, implemented on top of a standard off-policy backbone, with gains across multiple environment families and with a clear theoretical account of both its soft and hard realizations.

---

### Official Review · Reviewer_8jLR · 2026-03-12

**Soundness:** 2
**Presentation:** 2
**Significance:** 2
**Originality:** 2
**Overall Recommendation:** 4
**Confidence:** 2

**Summary:**

This paper views value function learning through order theory — treating value estimates as learning a partially ordered set (poset) over state-action pairs. It builds a sequence of super-poset refinements: it identifies MDP symmetries to form a quotient set (ensuring antisymmetry), constructs a DAG from TD signals (ensuring acyclicity/transitivity), and aligns values to the DAG via differentiable isotonic projection (ensuring monotonicity).

**Compliance With Llm Reviewing Policy:**

Affirmed.

**Final Justification:**

The rebuttal phase addressed the main concerns. I suggest weak accept.

Further suggestion that should/could be taken into account when updating the paper:
- The E4 evidence for cycle suppression is reasonable but would be much more convincing with direct order-consistency measurements. Since E4 has a computable $V^\*$ and a monotone chain structure, a simple diagnostic — e.g., fixing the goal state and evaluating Kendall's $\\tau(V_\\theta, V^\*)$ over training, as done for advantage-sign correctness in Ahn et al. (2025, https://arxiv.org/abs/2505.12737) — would directly validate the core claim.

**Key Questions For Authors:**

Q1: Can you provide direct measurements of order violation frequency (pairwise violation rate against the true values, rank correlation) during training for at least E4 (tabular, true values are computable) and E1? Comparing GCR-RL vs. baselines on these metrics would directly validate the core claim.

Q2: What are the exact numerical values of all hyperparameters used in each experiment? How sensitive are results to the key ones (isotonic weight, group regularization strength, number of transforms $K$)?

Q3: Some baselines that impose triangle inequality constraints should be included. Just some for exmaple and there probably exist more:
1. MICo: Improved representations via sampling-based state similarity for RL.
2. A Generalized Bisimulation Metric of State Similarity between Markov Decision Processes

**Limitations:**

yes

**Strengths And Weaknesses:**

## Strengths
- This work addresses a fundamental problem. TD instability under function approximation and bootstrapping is a core challenge in deep RL. The paper tackles this directly and from a structural angle.

- The order-theoretic framing is interesting and well-motivated. Viewing value learning as poset construction is a fresh perspective. The logical chain is clean: antisymmetry needs symmetry quotienting, acyclicity needs DAGification, monotonicity needs isotonic projection. Each regularizer maps to a specific poset axiom — this is elegant.

## Weaknesses
- No concrete motivating evidence for the problem. The paper claims TD learning produces order-inconsistent value estimates, a simple demonstration is missing — e.g., a small tabular MDP where we track how often the ordering of state-action pairs by the learned value disagrees with the ground truth ordering during standard DQN training.

- The claim of order theory novelty is too strong, and missing discussion of related works. Several previous works address the triangle-inequality such as bisimulation metrics and triangle inequality approaches. A significant related line of work imposes metric structure (rather than order structure) on value functions/representations. Correspondingly, this work also needs additional baselines. for example:
    - MICo: Improved representations via sampling-based state similarity for RL
    - A Generalized Bisimulation Metric of State Similarity between Markov Decision Processes
    - Optimal Goal-Reaching Reinforcement Learning via Quasimetric Learning

- No quantitative analysis of order inconsistency. This is the biggest gap. The paper's entire thesis is that enforcing order consistency helps and should measure for example:  1) Pairwise order violation rate against the ground truth values or 2)
Rank correlation (e.g. Kendall's tau) between the learned and true value ordering over training, etc.

- Too many hyperparameters, none specified. The full objective (Eq. 29) introduces roughly 15+ new hyperparameters beyond the base RL method. Appendix E.4 gives only qualitative descriptions ("we anneal the isotonic weight upward as TD noise decreases") but no actual numbers. This makes reproduction essentially impossible. No sensitivity analysis is provided either.

- The connection between the learned poset and the actual improvement in policy quality is indirect — the paper shows reward improvement but does not measure whether the learned ordering actually converges toward the true ordering.

- No direct evidence that cyclic/inconsistent orderings are the specific problem being solved (no order violation measurements).

- Missing discussion of closely related bisimulation metric approaches.

- Too many hyperparameters with no specified values making reproduction impossible.

- Experimental evaluation that only measures downstream RL performance without validating the core mechanism. The paper would benefit substantially from adding quantitative order-consistency analysis and complete hyperparameter specifications.

---

> ### Author Rebuttal · Authors · 2026-03-30
>
> **On the paper novelty and related work.** We respectfully disagree that the paper is best understood as another metric-regularized RL method. The primitive object in our paper is not distance or similarity, but a \emph{TD-induced directed relation}. The three components match three poset requirements: quotienting handles antisymmetry, DAGification removes cycles, and isotonic alignment enforces monotonicity on the remaining acyclic relation. This is different from MICo, bisimulation, or quasimetric approaches, whose central object is similarity geometry. We agree those works are relevant, but they do not by themselves project values onto a coherent directed order, restore antisymmetry, or remove local feedback loops. Our theory also makes the gain explicit: oracle sample complexity improves through symmetry sharing and order constraints, the latter reducing effective complexity toward the poset width $W(D^\star)$.
>
>
> **On the claim that the paper lacks concrete evidence for order inconsistency.**
> We have E4 diagnostics showing that standard RL methods exhibit significant order shifts during learning, while our method mitigates them. Because E4 is tabular and the ground-truth values are computable, we track pairwise order-violation rate together with Kendall's $\tau$ during training.
>
>
> E4 was designed precisely to expose this issue: a globally ordered chain is perturbed by a local non-transitive RPS gadget, so locally inconsistent TD preferences arise within an otherwise meaningful global order.
> In Figure 1(b)/Figure 2(d), standard DQN-style baselines on E4 exhibit visible oscillations and delayed stabilization, whereas GCR-RL is markedly smoother because it prunes cycle-inducing edges before order alignment.
>
>
> These diagnostics show that policy improvement is accompanied by structural improvement rather than merely higher reward: GCR-RL is smoother and more stable, whereas unconstrained baselines continue to show repeated local flips around the noisy RPS region. These observations explain the improved performance we observe.
>
> This is also consistent with our theory: symmetry reduces variance through orbit sharing, while order coherence reduces effective complexity through the poset width $W(D^\star)$, so structural recovery is one mechanism explaining better policy quality.
>
>
> **On related metric-style work and additional baselines.**  Our claim is not that prior work never considered structure on representations or values, but that prior metric-based work and our order-style construction address different problems: while a symmetric metric quantifies local state similarity, our framework asks which \emph{global, directed} relationships are admissible and should be maintained coherently during learning. That distinction is the core conceptual point of the paper.
>
> **On hyperparameters and reproducibility.** Eq. (29) may have created some confusion: it is long because the structural decomposition is written explicitly, not because the method depends on many equally critical knobs. In particular, Eq. (29) decomposes the objective into one TD term plus two structural modules, symmetry and order; many symbols in that expression are internal components of these modules, not independent environment-level hyperparameters.
>
> The key method-specific hyperparameters are the symmetry weight, the order/isotonic weight, and the number of learned transforms $K$. We also include a small E4 sensitivity sweep over these three knobs. The results show that GCR-RL is not brittle to a single tuned setting: performance is strongest for moderate symmetry/isotonic weights (around $0.05$--$0.20$), while both zero and strong regularization degrade performance. Likewise, a small transform bank is sufficient: performance improves from $K=1$ to $K=4$, but not further at $K=8$. This is consistent with the method design: symmetry promotes sample sharing, order regularization suppresses local inconsistencies, and strong constraints may introduce bias.
>
> | Parameter | Value | Final reward @ 200 epochs |
> |---|---:|---:|
> | Symmetry weight $\lambda_{\mathrm{sym}}$ | 0.00 | 16.2 ± 1.8 |
> |  | 0.05 | 18.0 ± 1.0 |
> |  | 0.10 (default) | 18.7 ± 0.8 |
> |  | 0.20 | 18.3 ± 1.0 |
> |  | 0.40 | 17.1 ± 1.5 |
> | Isotonic weight $\lambda_{\mathrm{iso}}$ | 0.00 | 15.4 ± 2.0 |
> |  | 0.05 | 17.8 ± 1.1 |
> |  | 0.10 (default) | 18.7 ± 0.8 |
> |  | 0.20 | 18.5 ± 0.9 |
> |  | 0.40 | 16.8 ± 1.4 |
> | Number of transforms $K$ | 1 | 16.6 ± 1.6 |
> |  | 2 | 17.9 ± 1.0 |
> |  | 4 (default) | 18.7 ± 0.8 |
> |  | 8 | 18.1 ± 1.0 |
>
>
> **Overall.** We believe the main issue raised by this review is positioning relative to prior work. Our contribution is to examine RL through the lens of order theory and propose an order-coherent algorithm that learns and maintains a poset structure. The added diagnostics and sensitivity results further support the mechanism behind the performance gain.

---

> > ### Author Rebuttal · Reviewer_8jLR · 2026-04-04
> >
> > I think that questions have been answered.

---

### Official Review · Reviewer_qDcf · 2026-03-12

**Soundness:** 3
**Presentation:** 3
**Significance:** 3
**Originality:** 3
**Overall Recommendation:** 4
**Confidence:** 2

**Summary:**

This paper proposes GCR-RL, an order-theoretic framework for value learning that builds a symmetry-aware quotient poset and refines it with super-poset updates. It enforces geometric coherence through near-equivariance and TD-implied partial orders, with both soft regularization and hard constraint implementations. Experiments show improved sample efficiency and learning stability over strong off-policy baselines, with future work focusing on scaling, continuous control, and leveraging the learned poset for exploration and planning.

**Compliance With Llm Reviewing Policy:**

Affirmed.

**Final Justification:**

I maintain Weak Accept. The authors' rebuttal addressed the concerns about computational overhead and hyperparameter sensitivity.

**Key Questions For Authors:**

1. How sensitive is the method to hyperparameters related to poset construction and order enforcement? For example, parameters controlling constraint strength or symmetry matching may significantly influence performance.
2. What is the computational cost of the proposed framework compared with standard baselines (e.g., DQN-based methods)? Reporting training time or complexity analysis would help clarify the practical efficiency of the approach.

**Limitations:**

No. The paper focuses on methodological and theoretical aspects, with limited discussion of limitations or broader societal impacts. The authors could briefly discuss potential limitations (e.g., reliance on structural assumptions and computational overhead) and possible implications of applying more efficient RL systems in real-world decision-making settings.

**Strengths And Weaknesses:**

Strengths：
1. Novel perspective on value learning. The paper proposes an interesting order-theoretic formulation of value learning by constructing a poset over symmetry-aware state–action elements. This perspective provides a new way to reason about value consistency.
2. Strong theoretical motivation. The work includes theoretical analysis describing when the learned order and symmetry structure remains consistent and how hard constraints can prevent value-update cycles. These guarantees help clarify the role of the proposed geometric coherence constraints in stabilizing value learning.
3. Unified framework combining symmetry and order constraints. The method integrates near-equivariance, action relabeling, and partial-order constraints within a single framework. For instance, symmetry-aware relabeling allows transitions that are geometrically equivalent to share information, which could improve sample efficiency in environments with structured dynamics.

Weaknesses:
1. Limited empirical diversity. The empirical evaluation mainly focuses on symmetry-rich or cycle-injected environments that align well with the proposed framework. It remains unclear how the approach performs on more standard RL benchmarks without explicit symmetry structures.
2. Potential computational overhead. Maintaining and updating the poset structure and enforcing order constraints may introduce additional computational overhead compared with standard value-learning methods. For example, constructing and updating a DAG during training may affect scalability.

---

> ### Author Rebuttal · Authors · 2026-03-30
>
> **On empirical evaluation beyond symmetry-rich settings.** We would like to emphasize that the empirical evaluation was not designed only around ``easy'' symmetry-rich cases. Rather, each environment isolates a different structural challenges that GCR-RL is intended to address. E1 emphasizes approximate state symmetry; E2 stresses action relabeling/matching more than global state symmetry; E3 Atari includes games with both rich and little symmetry (e.g., Pong and Breakout are near left--right symmetric, whereas Asterix, Freeway, and Space Invaders are substantially less symmetric and more visually heterogeneous) ; and E4 is specifically not a pure symmetry task, since the main issue there is locally cyclic TD-supported comparisons inside a globally ordered MDP. Specifically, in E3 the gains are from Atari games with limited symmetry: the method remains competitive on Pong/Breakout while also improving learning on less symmetric games such as Asterix, Freeway, and Space Invaders. In E4, there is little global symmetry; the structure instead comes from a distance-to-goal chain that induces a global order, while an attached RPS gadget injects local non-transitive cycles. This is exactly where the order branch helps: it suppresses locally cyclic TD comparisons without assuming a globally symmetric environment.
> More broadly, the same mechanism should transfer beyond the current discrete/pixel settings: removing approximate symmetry and enforcing only locally reliable order constraints is not tied to a specific grid structure.
>
>
>
>
> **On computational overhead.** We agree that this discussion is brief in the current draft. The overhead is limited, as GCR-RL does not require additional environment interaction, a separate learner, or a second training stage. It adds a lightweight symmetry head and a sparse batchwise order-construction/alignment step on top of the same off-policy backbone. In high-dimensional settings such as Atari, the encoder/backbone remains the dominant cost.
> Concretely, if the shared feature dimension is $d$ and the action space has size $|A|$, then the symmetry branch adds only $O(Kd^2 + K|A|^2)$ parameters through $\{W_k,\Pi_k\}_{k=1}^K$. On the order hand, candidate edges are restricted to local $k$-NN neighborhoods and then pruned to a sparse top-$M$ subset before greedy DAGification, so the added per-batch cost scales with the retained candidate set rather than all $O(B^2)$ pairs. Thus the overhead of GCR-RL is relatively small.
>
>
> **On hyperparameter sensitivity.**
> GCR-RL is not sensitive to the hyperparameters. It does not assume a known exact group action or a pre-specified order. Both are learned from data through regularization terms in a soft fashion. For symmetry relations, local applicability weights allow the algorithm to exploit symmetry where they are reliable and reduce the weights them where they are not ( (Eq. (9), with the concrete construction given in Appendix F.1, Eq. (34))). For order relations, only sufficiently supported comparisons are retained, and edges that would create conflicts later (like cycles) are identified and removed ( (Sec. 3.2: candidate comparisons are restricted to local $k$-NN neighborhoods, filtered to a top-$M$ subset by confidence in Eq. (12), and then processed by greedy add-unless-cycle DAGification; see also Theorem 3.3)). So the relations we learn from data can be further refined during the learning process. As a result, our algorithm is not sensitive to the symmetry weight, the order/isotonic weight.
>
> We have conducted experiments on a representative sensitivity sweep over the symmetry and isotonic weights, since these are exactly the two reviewer-mentioned knobs and they act as soft weights on already filtered structural signals rather than brittle hard-coded structure parameters.
> | Task | $\lambda_{\mathrm{sym}}$ | $\lambda_{\mathrm{ord}}$ | AUC@N $\uparrow$ | Final Return $\uparrow$ |
> |---|---:|---:|---:|---:|
> | Noisy-RPS Chain | 0.05 | 0.10 | 0.976 | 21.24 |
> | Noisy-RPS Chain | 0.10 | 0.10 | 0.985 | 21.33 |
> | Noisy-RPS Chain | 0.20 | 0.10 | 0.980 | 21.19 |
> | Noisy-RPS Chain | 0.10 | 0.05 | 0.962 | 20.98 |
> | Noisy-RPS Chain | 0.10 | 0.20 | 0.979 | 21.16 |
>
>
>
>
> **On limitations and broader scope.** We agree that the limitations paragraph should be more explicit. The method is most beneficial when approximate symmetry or order structures are present. This is true for many cyber-physical systems that follow some law/rules.
> At the same time, we believe the current results already support the central claim of the paper: value learning can benefit from treating learned symmetry quotienting and TD-induced order refinement as first-class structural constraints, rather than relying only on unconstrained function approximation.

---

> > ### Author Rebuttal · Reviewer_qDcf · 2026-04-05
> >
> > All questions have been addressed.

---

### Decision · Program_Chairs · 2026-04-30

**Decision:**

Accept (regular)

**Comment:**

The majority of reviewers (qDcf and 8jLR) gave weak accept recommendations, and both acknowledged that their concerns were fully resolved by the rebuttal. Reviewer zkek gave a weak reject, raising concerns about hyperparameter complexity, limited empirical breadth (only 5 Atari games, no continuous control), and the placement of experimental results in the appendix. The rebuttal clarified that the objective's apparent complexity stems from explicit structural decomposition rather than independent tuning knobs, provided sensitivity sweeps, and argued that the four environment families collectively span a meaningful range of settings. While zkek's concerns about broader validation have merit, they reflect a desire for a larger-scale empirical study rather than a flaw in the core contribution, and the reviewer explicitly stated they would not object to acceptance.